# HARNESSING SHALLOW FEATURES IN PRE-TRAINED TRANSFORMERS FOR OOD DETECTION

## ABSTRACT

Recognizing out-of-distribution (OOD) samples is essential for deploying robust machine learning systems in the open-world environments. Conventional OOD detection approaches rely on feature representations from the final layer of neuron networks, often neglecting the rich information encapsulated in shallow layers. Leveraging the strengths of transformer-based architectures, we introduce an attention-based fusion module, which dynamically assigns importance weights to representations learned by each Transformer layer and detects OOD samples using the Mahalanobis distance. Compared to existing approaches, our method enables a lightweight fine-tuning of pre-trained models, and retains all feature representations that are beneficial to the OOD detection. We also thoroughly study various parameter-efficient fine-tuning strategies. Our experiments show the benefit of using shallow features, and demonstrate the influence of different Transformer layers. We fine-tune pre-trained models in both class-balanced and long-tailed in-distribution classification tasks, and show that our method achieves state-of-the-art OOD detection performance averaged across nine OOD datasets. The source code is provided in the supplementary material.

## 1 INTRODUCTION

In recent years, deep learning models have made significant progress in various domains (Ramesh et al., 2021; Jumper et al., 2021). However, a critical issue with these models is their tendency to be overly confident in their predictions, even when the input deviates greatly from the data distribution seen during training. This issue underscores the need for effective out-of-distribution (OOD) detection when training deep neural networks (DNNs). The detection of OODs is crucial to ensure the safety of the model in many applications such as medical diagnostics (Schlegl et al., 2017), industrial inspection (Bergmann et al., 2019), and autonomous driving (Kitt et al., 2010). For example, in the field of medical imaging, DNNs may fail to provide an accurate diagnosis when presented with data that fall outside the training data distribution, such as images from an unknown scanner. Therefore, it is imperative for a reliable model not only to recognize in-distribution (ID) samples, but also to flag any OOD input as "unknown".

Existing OOD detection methods design various scoring functions to assign an input sample a likelihood to be OOD, using 1) *predicted probabilities* Hendrycks et al. (2019b); Meinke & Hein (2020); Liu et al. (2020); Fort et al. (2021a); Liu et al. (2023), 2) *output logits* Wang et al. (2022a); Ammar et al. (2024), and 3) *learned features* Kamoi & Kobayashi (2020); Fort et al. (2021a); Ming et al. (2022) by the model. However, these approaches neglect the rich information in the features learned by the layers of shallow neural networks. Our motivation stems from the observation that while the final features of a neural network are nonlinear transformations of shallow features and inherently retain some information from earlier layers, features extracted from different layers provide diverse representations of the data. Given that certain features may be particularly effective for distinguishing between ID and OOD samples, it is crucial to comprehensively leverage the information from all layers to enhance OOD detection performance. While the motivation is appealing, a core challenge remains: *how to effectively utilize shallow layer features for OOD detection?*

To address the above issue, we propose a new OOD detection approach by leveraging features from all layers with an attention-based fusion module. We draw inspiration from the geometric properties of "neural collapse" (Papyan et al., 2020), which states that the convergence of within-class

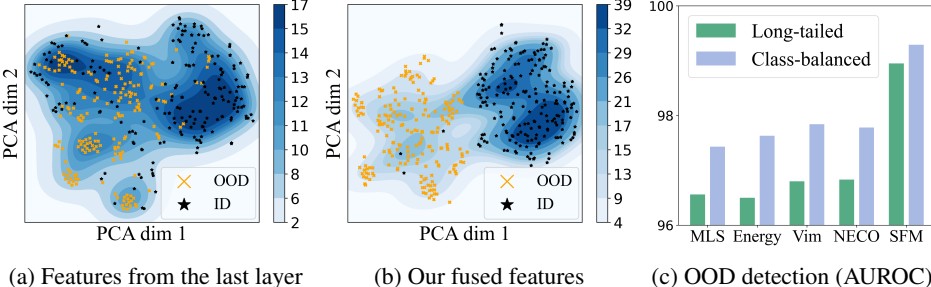

(a) Features from the last layer  (b) Our fused features  (c) OOD detection (AUROC)

Figure 1: A two-dimensional PCA projection of features from the last layer of the feature extractor and all Transformer layers fused by our approach, with examples of in-distribution (from CIFAR-100) and out-of-distribution data (from Texture). The color coding shows the Mahalanobis outlier score. The left panel shows that using the final learned features leads to overlapping clusters of ID and OOD embeddings. The shallow-feature fusion (middle panel) is able to distinguish ID and OOD data from each other well. By properly fusing shallow features, SFM achieves strong OOD detection performance (right panel).

covariance approaches zero in the terminal phase of training as each activation collapses toward its respective class mean. Therefore, we propose to measure the total variance of features across different layers of the neural network to describe their importance weights for OOD detection. Layers with larger total variance have more influence, while the contribution of layers with smaller total variance is down-weighted. The advantage of this method is that the weights of each layer are computed adaptively based on the data, without the need for manual parameter tuning. Using the weighted fused features, we calculate the Mahalanobis distance between the test sample and the data distribution of each ID class to calculate its OOD score. Figure 1 visualizes two-dimensional PCA projections of representations from the last layer of the feature extractor and all layers fused by our approach; we can observe that representations obtained by fusing transformer layers are better suited to OOD detection than representations from the last layer.

Additionally, we fine-tune the pre-trained visual models, including Vision Transformer (ViT) (Dosovitskiy et al., 2020) and CLIP (Radford et al., 2021), based on in-distribution data to achieve better representation learning. We find that parameter-efficient fine-tuning strategies can outperform full parameter fine-tuning (Ma et al., 2021; Long et al., 2022; Tian et al., 2022) and are more robust to hyperparameter choice; specifically, by freezing the pre-trained model and adding a small number of learnable parameters. Based on this finding, we develop a general fine-tuning framework and implement all comparison methods within this framework in our experiments. We also conducted an in-depth analysis of various fine-tuning strategies.

To evaluate our approach, we focus on both class-balanced ID datasets, which are commonly used in existing OOD detection literature Liu et al. (2020); Wang et al. (2022a); Ammar et al. (2024), and long-tailed ID datasets because the distribution of real-world data is often imbalanced and highly skewed per class basis, with a majority of classes containing a small number of samples. Long-tailed OOD detection has been studied in several recent works by improving 1) *representation learning* (Wang et al., 2022b; Wei et al., 2024; 2022; Choi et al., 2023), and 2) *probabilistic calibration* (Jiang et al., 2023; Miao et al., 2024). However, these methods often require the use of OOD data to train the model. In contrast, our approach only requires fine-tuning the model using ID data, and more importantly, with no changes needed for the shallow feature fusion module. Our contributions are summarized as follows:

1. We propose a new OOD detection method that exploits features from shallow layers of pre-trained Transformers to enhance OOD separation.

2. We propose a simple yet powerful way to fuse the shallow layer features with the importance weights by measuring the covariance of features in each layer.

3. Our method achieves state-of-the-art results on both class-balanced and long-tailed benchmark datasets. We achieve an absolute improvement of AUROC by $1.07\%$ and a reduction of FPR95 by $4.03\%$ on average across benchmark ID and OOD datasets.

## 2 RELATED WORKS

**Out-of-distribution detection.** In recent years, the field of OOD detection has gained considerable attention. The Maximum Softmax Probability (MSP) method (Hendrycks & Gimpel, 2016) serves as a foundational baseline, utilizing softmax predictions as OOD scores. Building on this, ODIN (Liang et al., 2017) improves the softmax score by perturbing input data and rescaling logits, enhancing its effectiveness in distinguishing OOD samples. Further advancements explore alternative scoring mechanisms, such as the energy score (Liu et al., 2020), which is further refined through feature clipping in ReAct (Sun et al., 2021). Additionally, gradient-based approaches have been explored to differentiate between ID and OOD data (Huang et al., 2021; Agarwal et al., 2022). Among previous studies, the use of the Mahalanobis distance has shown significant promise. Lee et al. (2018) propose a method where an auxiliary OOD validation dataset is employed to determine the optimal weighting for each layer in calculating the Mahalanobis distance score. Trusted (Colombo et al., 2022) introduces a novel approach that combines feature fusion during training with the Mahalanobis distance during testing, guided by the optimal transport principle. On top of the CLIP model, CLIPN Wang et al. (2023) learns a "no" prompt to capture the negation-semantic with images using an auxiliary dataset, and performs OOD detection depending on the similarity between the input image and the "no" prompt. Similarly, NegLabel Jiang et al. (2024) extracts potential negative labels from a corpus database and employs zero-shot CLIP for OOD detection by combining ID classes and negative labels.

**Long-tailed out-of-distribution detection.** In the context of long-tailed OOD detection, prior research has examined several strategies to mitigate the challenges posed by class imbalance, including the use of oversampling techniques and threshold adjustments to improve performance (Li et al., 2022). Open Sampling (Wei et al., 2022) incorporates OOD data to address the class imbalance in long-tailed datasets. PASCL (Wang et al., 2022b) focuses on enhancing representation learning for tail classes by leveraging a contrastive learning method, helping to improve the separation between minority classes and OODs. Jiang et al. (2023) identify several common scenarios where the OOD-to-ID probabilities should be the ID-class-prior distribution and propose two strategies to modify existing inference-time detection methods. EAT (Wei et al., 2024) proposes expanding the class space of ID classes with virtual classes to tackle OOD data. COCL (Miao et al., 2024) introduces a calibrated learning approach aimed at improving outlier class detection in long-tailed tasks.

**Parameter-efficient fine-tuning.** PEFT methods freeze the pre-trained model and introduce only a few learnable parameters for adaptation, which can effectively reduce overfitting and accelerate convergence. Adapter (Dosovitskiy et al., 2020) introduces a bottleneck module to optimize only a small subset of parameters. BitFit (Zaken et al., 2021) focuses on fine-tuning only the bias terms of the model, significantly reducing the number of parameters that need to be updated during training. VPT (Jia et al., 2022) prepends learnable prompts at each layer, offering two versions: VPT-Shallow, which uses prompts at shallow layers, and VPT-Deep, which applies them across deeper layers. LoRA (Hu, 2022) further optimizes efficiency by applying low-rank adaptations, minimizing the overall parameter count while retaining performance. AdaptFormer (Chen et al., 2022) builds on the Adapter method by shifting from a sequential to a parallel design. LIFT (Shi et al., 2024) provides an empirical analysis showing that the commonly used full fine-tuning strategy is prone to overfitting, especially on long-tailed datasets.

## 3 METHOD

In this section, we present a simple yet effective OOD detection method by fusing features from shallow Transformer layers based on importance weight.

### 3.1 PRELIMINARY

We first introduce the problem setting and notations used throughout this paper.

1. We denote the training set as $\mathcal{D}_{train} = \{(\boldsymbol{x}_i, y_i)\}_{i=1}^{N}$, where $\boldsymbol{x}_i \in \mathbb{R}^d$ represents an input image, $y_i \in [C]$ denotes its ground-truth class label, and $C$ denotes the total number of classes in the training set. At test time, our goal is to flag images that do not belong to any of the training classes using our OOD detector.

2. Without loss of generality, let the network be $F = f \circ g$, where $f(\cdot)$ is known as the feature exactor and $g(\cdot)$ is the classifier following $f$. For each layer $\phi(\cdot)$ in $f$, we define the transformation learned by the $l$-th layer as $\phi_l(\cdot)$. For an instance $\boldsymbol{x}_i$, its output from the $l$-th layer is denoted as $\boldsymbol{x}^l = \phi_l(\boldsymbol{x})$. Particularly, we have the final feature learned by the model $\boldsymbol{x}^L = \phi_L(\boldsymbol{x})$, where $L$ denotes the number of Transformer layers.

3. In this paper, we build our OOD detector based on the Mahalanobis distance. For any test image $\boldsymbol{x}$, we calculate the negative distance between the image feature $f(\boldsymbol{x})$ and feature distribution of each class as the scoring function:

$$M(\boldsymbol{x}; \boldsymbol{\mu}_c, \boldsymbol{\Sigma}) = -\left(f(\boldsymbol{x}) - \boldsymbol{\mu}_c\right)^\top \boldsymbol{\Sigma}^{-1} \left(f(\boldsymbol{x}) - \boldsymbol{\mu}_c\right) \tag{1}$$

where $\boldsymbol{\mu}_c$ is the mean feature vector of class $c$ and $\boldsymbol{\Sigma}$ is the covariance matrix of ID data.

4. To measure the Mahalanobis distance, we calculate the empirical class mean and covariance matrix of training samples as follows:

$$\boldsymbol{\mu}_c = \frac{1}{N_c} \sum_{i:y_i=c} f(\boldsymbol{x}_i), \ \boldsymbol{\Sigma} = \frac{1}{N} \sum_{c=1}^{C} \sum_{i:y_i=c} \left(f(\boldsymbol{x}_i) - \boldsymbol{\mu}_c\right)\left(f(\boldsymbol{x}_i) - \boldsymbol{\mu}_c\right)^\top, \tag{2}$$

where $N_c$ is the number of training samples with class $c$. This is equivalent to fitting the class-conditional Gaussian distribution with a tied covariance to the training samples under the maximum likelihood estimator (Lee et al., 2018).

## 3.2 SFM: Shallow Feature Matters for OOD Detection

By default, the Mahalanobis distance in Eq. (1) uses the final output of the feature extractor, i.e., $f(\boldsymbol{x})$, neglecting rich information in shallow layer features. Therefore, we now proceed to present our approach to demonstrate that shallow features can help improve OOD detection performance.

For any test image $\boldsymbol{x}$ and a fine-tuned model, we first obtain its hidden representations $\boldsymbol{x}_i^l$ of the $l$-th layer, $\forall 1 \leq l \leq L$. Notably, we may use "features" and "representations" interchangeably throughout the paper. We then integrate features from all layers by different importance weights. Formally, we compute the fused feature representation of $\boldsymbol{x}$ by:

$$\Phi(\boldsymbol{x}) = \sum_{l=1}^{L} \alpha^l \boldsymbol{x}^l, \tag{3}$$

where $\alpha^l$ is the weight of the $l$-th layer. To measure the Mahalanobis distance, we also calculate the class mean feature vectors and global covariance matrix in the fused feature space. We reformulate Eq. (2) by fusing shallow features as follows:

$$M_{\text{SFM}}(\boldsymbol{x}; \widetilde{\boldsymbol{\mu}}_c, \widetilde{\boldsymbol{\Sigma}}) = -\left(\Phi(\boldsymbol{x}) - \widetilde{\boldsymbol{\mu}}_c\right)^\top \widetilde{\boldsymbol{\Sigma}}^{-1} \left(\Phi(\boldsymbol{x}) - \widetilde{\boldsymbol{\mu}}_c\right), \tag{4}$$

where $\widetilde{\boldsymbol{\Sigma}} = \frac{1}{N} \sum_{c=1}^{C} \sum_{i:y_i=c} \left(f(\boldsymbol{x}_i) - \widetilde{\boldsymbol{\mu}}_c\right)\left(f(\boldsymbol{x}_i) - \widetilde{\boldsymbol{\mu}}_c\right)^\top$ and $\widetilde{\boldsymbol{\mu}}_c = \frac{1}{N_c} \sum_{i:y_i=c} \Phi(\boldsymbol{x}_i)$.

To reflect the importance of each layer, we propose to calculate the weights by measuring the discriminative capacity or variability of the features of each layer.

**Definition 3.1** (Measure of Variablitiy). Given a collection of $\boldsymbol{x}_l$, we calculate the mean feature by $\boldsymbol{\mu}^l = \frac{1}{N} \sum_{i=1}^{N} \boldsymbol{x}_i^l$, then and measure the feature variability of the $l$-th layer by:

$$\alpha^l = \text{Tr}((\boldsymbol{A}^l)^\top \boldsymbol{A}^l), \tag{5}$$

where $\boldsymbol{A}^l = (\boldsymbol{x}_1^l - \boldsymbol{\mu}^l, \boldsymbol{x}_2^l - \boldsymbol{\mu}^l, \cdots, \boldsymbol{x}_N^l - \boldsymbol{\mu}^l)^\top$ is the centralized feature matrix of the $l$-th layer, and $\text{Tr}(\cdot)$ denotes the trance of a matrix, which is the sum of its diagonal elements. We normalize the weights so that the sum of the weights across all layers is equal to $1$.

**Interpretation of the Trace (Trace as a Measure of Variability).** The trace of the matrix, $\text{Tr}((A^l)^\top A^l)$, is the sum of the variances along each principal direction of the feature space in the $l$-th layer. Essentially, it quantifies the overall variability or spread of the features across different samples. In the context of feature fusion, a higher trace value indicates that the features of the $l$-th layer exhibit substantial variability among the samples, suggesting that these features have

a strong discriminative capacity. Therefore, such layers should be assigned higher weights during the fusion process. By using the trace as a measure of the variability of features, $\alpha^l$ reflects the importance of the features of the $l$ th layer based on their ability to distinguish between different classes. A higher trace value implies that the features can capture more discriminative information, which is crucial for tasks such as ID classification and OOD detection. Intuitively, the last layer of the feature extractor has the highest weight because the features learned from it exhibit the best inter-class discriminative ability; however, the weight of the first layer is often smaller.

**Remark.** Our work differs from *Mahalanobis* (Lee et al., 2018) and *Trusted* (Colombo et al., 2022) which also use features from multiple layers. 1) *Mahalanobis* calculates the OOD score using each layer's feature individually and weights them together by training a logistic regression model using the validation set. Our approach computes importance weights from training data and does not require any validation set. 2) *Trusted* treats every layer equally with the same importance and averages the features. It is clear that certain layer features may be more effective for detecting OODs, whereas others may bring noise. Our approach can prevent the degradation of the overall OOD detection performance even in the case when the features from some layers are not effective: the weights would be nearly zero for those ineffective layers.

### 3.3 NEW INSIGHTS ON FINE-TUNING PRE-TRAINED MODELS FOR OOD DETECTION

**Parameter-efficient fine-tuning is more robust than fully fine-tuning.** To adapt the pre-trained models to downstream classification and OOD detection tasks, we learn a linear classifier and fine-tune the feature extractor using ID training data. In this paper, we adopt the logit adjustment loss (Menon et al., 2020) as the optimization objective for its simplicity and good generalization ability. The key advantage of this choice is that, for class-balanced ID datasets, it simplifies to the conventional cross-entropy loss; however, for long-tailed ID datasets, it allows the model to balance predictive confidence across classes. Formally, the logit adjustment loss is defined as:

$$\mathcal{L}_{\text{LA}}(\boldsymbol{x}, y = j) = -\log \frac{\exp(z_j + \log \mathrm{P}(y = j))}{\sum_{k \in [C]} \exp(z_k + \log \mathrm{P}(y = k))} \tag{6}$$

where $y = j$ denotes the ground-truth label of the input $\boldsymbol{x}$, and $z_j$ is the logit (pre-softmax activation) for class $j$. The class-prior probability $\mathrm{P}(y = j)$ is estimated from the training distribution.

However, when choosing the fine-tuning strategy, we observe that full parameter fine-tuning (FFT) is significantly more sensitive to hyperparameters, such as learning rate, compared to parameter-efficient fine-tuning (PEFT), especially when the ID data follows a long-tailed label distribution. Figure 2 highlights the impact of learning rates on both fine-tuning strategies in CIFAR-100 (ID) classification accuracy and OOD detection AUROC, averaged on six OOD

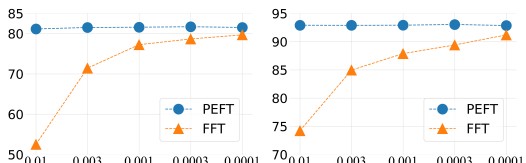

(a) ImageNet-LT ACC    (b) ImageNet-LT AUROC

Figure 2: Comparison of the sensitivity of FFT and PEFT to learning rates.

datasets. The x-axis denotes the learning rate. The results indicate that FFT requires careful tuning of learning rates to achieve optimal performance, while PEFT demonstrates more robust performance across a wider range of hyperparameters. Moreover, FFT necessitates tuning hyperparameters like the learning rate individually for each dataset, whereas PEFT allows for consistent hyperparameter settings across multiple datasets, reducing the burden of hyperparameter search.

**Extension of Our Approach to Vision-Language Models.** Our empirical analysis indicates that models pre-trained on large-scale supervised classification datasets (e.g., ViT pre-trained on ImageNet-21k) tend to capture more discriminative deep and shallow features compared to models pre-trained through self-supervised tasks (e.g., CLIP). To address this disparity, we extend SFM by leveraging the alignment between image and text embeddings learned in the feature space of vision-language models. Specifically, we calculate the cosine similarity between the image embedding and ID class text prompt embeddings with minimal computational overhead. This similarity score is integrated into SFM to enhance the effectiveness of OOD detection. Formally, the revised scoring function is defined as:

$$G(\boldsymbol{x}) = \max_{c \in [C]} M_{\text{SFM}}(\boldsymbol{x}; \widetilde{\boldsymbol{\mu}}_c, \widetilde{\boldsymbol{\Sigma}}) + \lambda \cdot \text{sim}(\boldsymbol{v}, \boldsymbol{t}_c) \tag{7}$$

where $\boldsymbol{v}$ denotes the image embedding of $\boldsymbol{x}$ extracted by the pre-trained image encoder, and $\boldsymbol{t}_c$ represents the text prompt embedding of class $c$, i.e., both image and text embeddings are obtained from pre-trained CLIP. The similarity measure $\text{sim}(\boldsymbol{v}, \boldsymbol{t}_c)$ is defined as: $\text{sim}(\boldsymbol{v}, \boldsymbol{t}_c) = \frac{e^{\boldsymbol{v}^\top \boldsymbol{t}_c}}{\sum_k e^{\boldsymbol{v}^\top \boldsymbol{t}_k}}$, where we use the default prompt template "a photo of a {classname}" to obtain text embedding $\boldsymbol{t}_c$ in our experiments. The hyperparameter $\lambda$ controls the relative influence of the predicted similarity scores of the vision-language model. *Notably, we set $\lambda = 0$ when using vision-only models.* A test image is classified as OOD if $G(\boldsymbol{x}) \geq \rho$, where $\rho$ is selected such that a high proportion of ID data exceeds this threshold. For samples classified as ID, the class label is determined as $\hat{y} = \arg\max_{c \in [C]} p_c$, where $\boldsymbol{p} = F(\boldsymbol{x})$ denotes the predicted class probabilities from the classifier.

## 4 EXPERIMENTS

### 4.1 EXPERIMENT SETUP

In this section, we compare our approach with the latest algorithms across both small- and large-scale OOD detection benchmarks. In line with prior research, we utilize CIFAR-100 and ImageNet as the in-distribution (ID) datasets. Additionally, we incorporate the more challenging long-tailed variants, CIFAR-100-LT and ImageNet-LT, as ID training sets to further demonstrate the effectiveness of our proposed method in OOD detection scenarios. The imbalance ratio for CIFAR-100-LT is set to 100, reflecting a highly imbalanced class distribution.

**OOD datasets.** When CIFAR-100 or CIFAR-100-LT is used as the ID dataset, we evaluate OOD detection performance on a range of diverse datasets, including Textures (Cimpoi et al., 2014), SVHN (Yuval, 2011), CIFAR-10, Tiny ImageNet (Le & Yang, 2015), LSUN (Yu et al., 2015), and Places365 (Zhou et al., 2016). For experiments with ImageNet and ImageNet-LT as the ID datasets, we assess the model's OOD detection capability using five benchmark OOD datasets: Textures (Cimpoi et al., 2014), Places365 (Zhou et al., 2016), iNaturalist (Van Horn et al., 2018), ImageNet-O (Hendrycks et al., 2021), and SUN (Xiao et al., 2010).

**Evaluation metrics.** We present our results using two widely adopted OOD evaluation metrics (Yang et al., 2021). The first metric is the area under the receiver operating characteristic curve (AUROC), which is threshold-independent and evaluates the trade-off between true positive rate (TPR) and false positive rate (FPR). Higher AUROC values indicate better OOD detection performance. The second metric is FPR95, which measures the false positive rate at a 95% TPR. Lower FPR95 values signify superior OOD detection capabilities. Both metrics are presented as percentages. The highest AUROC ($\uparrow$) and lowest FPR95 ($\downarrow$) scores averaged on OOD datasets are highlighted in bold, while the second-best results are underlined.

**Implementation details.** We implement our approach and *all competing methods* in the same framework on top of the ImageNet-21k pre-trained Vision Transformer (ViT) (Dosovitskiy et al., 2020) and the official pre-trained CLIP model[1]. We fine-tune the pre-trained models using in-distribution data for downstream tasks. We employ a batch size of 64 for all experiments. For CIFAR-100 and CIFAR-100-LT, we set the initial learning rate to 0.01 with a cosine annealing scheduler and fine-tune for 10 epochs. For ImageNet and ImageNet-LT, the initial learning rate is set to 0.1, with a cosine annealing scheduler, and the models are fine-tuned for 5 and 20 epochs, respectively. We set $\lambda = 1$ on ImageNet and $\lambda = 0.1$ on CIFAR-100 for the CLIP model to calculate the scoring function as defined in Eq. (7). For the Adapterformer module, we set the dimension to $\frac{C}{2L}$, where $C$ is the number of classes, and $L$ is the number of blocks in the ViT model. Other hyperparameters include a momentum of 0.9, and a weight decay of $5 \times 10^{-4}$, following LIFT (Shi et al., 2024). For all baseline methods, we ensure a fair comparison by using the same hyperparameter settings. All experiments are conducted on a single NVIDIA RTX 3090 GPU.

**Baselines.** We compare our method with eight baselines, including MSP (Hendrycks & Gimpel, 2016), MLS (Hendrycks et al., 2019a), Energy (Liu et al., 2020), Mahalanobis (Lee et al., 2018), Residual and Vim (Wang et al., 2022a), NECO (Ammar et al., 2024), MCM (Ming et al., 2022), Trusted (Colombo et al., 2022), and KL-matching (Hendrycks et al., 2019a). For Mahalanobis, we follow the setting in (Fort et al., 2021b), which uses only the final feature instead of an ensemble of

---

[1]https://github.com/openai/CLIP

Table 1: OOD detection performance on CIFAR-100 (ID) and six OOD datasets.

| Method | Texture | | SVHN | | CIFAR10 | | Tiny ImageNet | | LSUN | | Places365 | | Average | |
|---|---|---|---|---|---|---|---|---|---|---|---|---|---|---|
| | AUROC | FPR95 | AUROC | FPR95 | AUROC | FPR95 | AUROC | FPR95 | AUROC | FPR95 | AUROC | FPR95 | AUROC | FPR95 |
| IMAGENET-21K PRE-TRAINED VIT | | | | | | | | | | | | | | |
| MSP | 97.65 | 11.81 | 94.91 | 28.17 | 94.92 | 26.32 | 88.58 | 44.50 | 86.75 | 64.21 | 92.23 | 41.41 | 92.51 | 36.07 |
| MLS | 99.79 | 0.83 | 97.38 | 10.31 | 97.07 | 13.42 | 93.28 | 25.16 | 98.09 | 10.93 | 98.98 | 5.39 | 97.43 | 11.01 |
| Energy | 99.86 | 0.57 | 97.48 | 9.47 | **97.09** | **12.88** | 93.51 | 23.61 | 98.59 | 7.58 | 99.26 | 3.78 | 97.63 | 9.65 |
| Mahalanobis | 99.97 | 0.12 | 99.16 | 3.92 | 97.09 | 16.49 | 97.99 | 8.96 | 99.61 | 1.07 | 99.67 | 1.33 | 98.92 | 5.32 |
| Residual | 99.99 | 0.02 | 97.66 | 12.81 | 92.08 | 41.38 | 99.10 | 3.68 | 99.93 | 0.00 | 99.92 | 0.08 | 98.12 | 9.66 |
| Vim | 99.89 | 0.44 | 97.68 | 8.63 | 97.13 | 12.73 | 94.09 | 21.96 | 98.85 | 5.72 | 99.39 | 2.94 | 97.84 | 8.74 |
| NECO | 99.83 | 0.83 | 97.95 | 8.70 | 97.31 | 13.98 | 94.25 | 21.93 | 98.29 | 10.77 | 99.08 | 5.35 | 97.78 | 10.26 |
| Trusted | 100.0 | 0.00 | 98.78 | 5.77 | 93.35 | 33.51 | 98.09 | 9.76 | 100.0 | 0.00 | 100.0 | 0.00 | 98.37 | 8.17 |
| KL-matching | 98.60 | 6.10 | 96.66 | 14.93 | 96.34 | 17.12 | 90.05 | 34.17 | 88.15 | 49.34 | 93.67 | 28.21 | 93.91 | 24.98 |
| SFM (ours) | **100.0** | **0.00** | **99.50** | **1.91** | 96.47 | 19.52 | **99.80** | **1.11** | **100.0** | **0.00** | **100.0** | **0.00** | **99.29** | **3.76** |
| CLIP-VIT-B/16 | | | | | | | | | | | | | | |
| MSP | 91.14 | 41.33 | 86.22 | 57.75 | 87.35 | 53.18 | 82.11 | 62.50 | 74.83 | 80.64 | 84.02 | 60.61 | 84.28 | 59.33 |
| MLS | 96.11 | 20.73 | 91.58 | 41.81 | 93.32 | 30.69 | 88.58 | 45.86 | 88.49 | 51.20 | 93.15 | 33.12 | 91.87 | 37.23 |
| Energy | 96.56 | 18.03 | 91.85 | 41.92 | **93.77** | **28.89** | 89.06 | 44.49 | 89.66 | 45.66 | 93.92 | 29.21 | 92.47 | 34.70 |
| Mahalanobis | 99.23 | 1.68 | 96.89 | 23.27 | 89.01 | 52.26 | 93.75 | 32.28 | 98.81 | 6.44 | 99.29 | 3.13 | 96.16 | 19.84 |
| Residual | 99.05 | 1.86 | 95.61 | 31.96 | 82.22 | 67.74 | 94.48 | 31.92 | 99.19 | 3.05 | 99.36 | 2.03 | 94.98 | 23.09 |
| Vim | 97.23 | 14.33 | 92.88 | 36.41 | 93.82 | 28.66 | 88.94 | 41.40 | 91.58 | 38.73 | 95.13 | 23.97 | 93.43 | 30.58 |
| NECO | 97.67 | 12.20 | 94.04 | 33.31 | 93.57 | 31.58 | 90.25 | 41.08 | 92.65 | 34.50 | 95.90 | 21.27 | 94.02 | 28.99 |
| MCM | 72.98 | 92.09 | 90.75 | 63.39 | 75.53 | 88.66 | 65.54 | 93.36 | 50.79 | 99.11 | 60.97 | 97.79 | 69.43 | 89.06 |
| Trusted | 99.98 | 0.04 | 97.21 | 17.80 | 86.32 | 61.45 | 97.13 | 15.68 | 99.95 | 0.03 | 99.96 | 0.08 | 96.76 | 15.85 |
| KL-matching | 94.32 | 25.12 | 90.69 | 38.25 | 90.69 | 38.52 | 84.16 | 52.94 | 77.85 | 70.96 | 86.99 | 47.80 | 87.45 | 45.60 |
| SFM (ours) | **99.95** | **0.02** | **98.31** | **8.62** | 88.56 | 53.97 | **97.54** | **12.91** | **99.93** | **0.06** | **99.95** | **0.02** | **97.37** | **12.60** |

Table 2: OOD detection performance on CIFAR-100-LT (ID) and six OOD datasets.

| Method | Texture | | SVHN | | CIFAR10 | | Tiny ImageNet | | LSUN | | Places365 | | Average | |
|---|---|---|---|---|---|---|---|---|---|---|---|---|---|---|
| | AUROC | FPR95 | AUROC | FPR95 | AUROC | FPR95 | AUROC | FPR95 | AUROC | FPR95 | AUROC | FPR95 | AUROC | FPR95 |
| IMAGENET-21K PRE-TRAINED VIT | | | | | | | | | | | | | | |
| MSP | 97.21 | 13.12 | 95.52 | 24.13 | 91.92 | 38.50 | 85.27 | 48.02 | 84.06 | 64.81 | 90.47 | 43.10 | 90.75 | 38.61 |
| MLS | 99.83 | 0.62 | 96.38 | 18.35 | 94.94 | 25.58 | 90.36 | 34.08 | 98.58 | 7.52 | 99.26 | 3.06 | 96.56 | 14.87 |
| Energy | 99.89 | 0.43 | 95.65 | 24.00 | 94.49 | 29.42 | 90.38 | 34.32 | 99.09 | 4.00 | 99.52 | 1.62 | 96.50 | 15.63 |
| Mahalanobis | 99.96 | 0.20 | 99.33 | 2.51 | **95.09** | 25.98 | 97.63 | 9.26 | 99.48 | 2.26 | 99.57 | 1.71 | 98.51 | 6.99 |
| Residual | 99.98 | 0.05 | 97.33 | 17.74 | 86.41 | 62.76 | 98.52 | 6.90 | 99.83 | 0.47 | 99.80 | 0.45 | 96.98 | 14.73 |
| Vim | 99.91 | 0.28 | 96.18 | 20.72 | 94.56 | 29.01 | 91.27 | 31.59 | 99.25 | 3.20 | 99.60 | 1.23 | 96.80 | 14.34 |
| NECO | 99.86 | 0.64 | 97.37 | 13.58 | 94.91 | **24.62** | 91.22 | 29.21 | 98.39 | 10.21 | 99.22 | 3.78 | 96.83 | 13.67 |
| Trusted | 100.0 | 0.00 | 99.12 | 3.60 | 87.34 | 52.84 | 97.67 | 10.37 | 99.97 | 0.00 | 99.98 | 0.00 | 97.35 | 11.13 |
| KL-matching | 98.48 | 6.40 | 97.44 | 12.11 | 94.00 | 26.88 | 87.56 | 38.91 | 86.65 | 52.78 | 92.94 | 31.01 | 92.84 | 28.02 |
| SFM (ours) | **100.0** | **0.00** | **99.75** | **0.43** | 94.22 | 29.86 | **99.75** | **1.12** | **99.99** | **0.00** | **99.99** | **0.01** | **98.95** | **5.24** |
| CLIP-VIT-B/16 | | | | | | | | | | | | | | |
| MSP | 91.05 | 39.34 | 86.13 | 48.73 | 85.33 | 55.47 | 78.22 | 68.10 | 73.52 | 76.50 | 83.16 | 57.92 | 82.90 | 57.68 |
| MLS | 96.76 | 16.95 | 88.44 | 49.78 | 91.85 | 36.84 | 87.05 | 47.53 | 90.35 | 36.77 | 94.29 | 25.52 | 91.46 | 35.57 |
| Energy | 97.31 | 13.09 | 86.40 | 59.64 | **92.37** | **34.15** | 88.01 | 43.79 | 92.25 | 28.45 | 95.49 | 19.49 | 91.97 | 33.10 |
| Mahalanobis | 99.11 | 1.03 | 95.92 | 29.87 | 84.76 | 60.58 | 90.97 | 43.83 | 99.08 | 4.07 | 99.28 | 1.99 | 94.85 | 23.56 |
| Residual | 98.90 | 1.42 | 94.83 | 33.99 | 77.19 | 73.51 | 91.24 | 48.57 | 99.28 | 1.94 | 99.34 | 0.87 | 93.46 | 26.72 |
| Vim | 98.12 | 9.17 | 88.61 | 52.92 | 92.19 | 35.68 | 88.97 | 41.63 | 94.26 | 21.87 | 96.76 | 14.76 | 93.15 | 29.34 |
| NECO | 98.00 | 9.57 | 91.32 | 41.13 | 91.11 | 40.21 | 87.51 | 46.54 | 93.99 | 23.37 | 96.71 | 16.22 | 93.11 | 29.51 |
| Trusted | 99.97 | 0.11 | 93.57 | 43.80 | 80.76 | 70.36 | 95.46 | 25.58 | 99.95 | 0.10 | 99.95 | 0.08 | 94.94 | 23.34 |
| KL-matching | 95.01 | 21.76 | 90.76 | 31.69 | 88.87 | 44.17 | 81.68 | 57.93 | 79.31 | 63.65 | 87.64 | 43.21 | 87.21 | 43.73 |
| SFM (ours) | **99.92** | **0.02** | **97.50** | **15.65** | 85.20 | 60.41 | **95.05** | **26.05** | **99.92** | **0.04** | **99.93** | **0.01** | **96.25** | **17.03** |

multiple layers (Huang & Li, 2021; Lee et al., 2018). It is worth noting that all these baselines are reimplemented based on our fine-tuned models, except that MCM uses zero-shot CLIP.

## 4.2 MAIN RESULTS

**Result on CIFAR-100 and CIFAR-100-LT.** As shown in Table 1, our proposed method, SFM, outperforms state-of-the-art approaches across multiple OOD datasets. In particular, the average performance of SFM on both the CLIP model and the ImageNet-21k pre-trained ViT significantly surpasses previous methods. SFM achieves perfect separation of ID and OOD data on Texture, LSUN, and Places365 datasets. However, we observe a decrease in the performance when using CIFAR-10 as the OOD data. This reduction can be attributed to the high similarity between CIFAR-10 and CIFAR-100 in terms of characteristics, resolution, and visual style—both datasets consist of low-resolution, $32 \times 32$ images with somewhat blurred features, making certain samples challenging to differentiate, even for human observers. This resemblance leads to overlapping feature representations in the shallow layers, resulting in relatively diminished performance. Notably, MCM (Ming et al., 2022) is a zero-shot CLIP-based OOD detection method, and its performance is significantly inferior to other methods, highlighting the necessity of fine-tuning for downstream tasks.

Additionally, Table 2 presents the results on the long-tailed in-distribution dataset. It can be seen that our method consistently outperforms previous approaches. When using the CLIP model, our method effectively reduces the FPR95 by an average of 6.53% (from 23.56% to 17.03%).

Table 3: OOD detection performance on ImageNet (ID) and five OOD datasets. † indicates the results are taken from their papers, except that results for MCM on ImageNet-O are reproduced using official codebase.

| Method | Texture | | Places | | SUN | | iNaturalist | | ImageNet-O | | Average | |
|---|---|---|---|---|---|---|---|---|---|---|---|---|
| | AUROC | FPR95 | AUROC | FPR95 | AUROC | FPR95 | AUROC | FPR95 | AUROC | FPR95 | AUROC | FPR95 |
| IMAGENET-21K PRE-TRAINED VIT | | | | | | | | | | | | |
| MSP | 84.89 | 51.88 | 84.52 | 59.44 | 85.31 | 56.52 | 95.86 | 18.73 | 82.24 | 60.00 | 86.56 | 49.31 |
| MLS | 90.12 | 37.80 | 88.01 | 51.67 | 89.72 | 47.21 | 97.98 | 8.75 | 89.79 | 44.65 | 91.12 | 38.02 |
| Energy | 90.72 | 34.65 | 88.15 | 50.40 | 90.06 | 45.31 | 98.23 | 7.41 | 90.73 | 41.00 | 91.58 | 35.75 |
| Mahalanobis | 92.93 | 26.31 | 89.27 | 47.56 | 91.53 | 39.82 | 99.33 | 2.72 | 92.12 | 37.50 | 93.03 | 30.78 |
| Residual | 92.84 | 30.66 | 84.80 | 61.14 | 88.34 | 50.14 | 98.02 | 9.51 | 87.11 | 52.50 | 90.22 | 40.79 |
| Vim | 91.04 | 33.33 | 88.37 | 49.82 | 90.30 | 44.34 | 98.37 | 6.86 | 90.92 | 40.20 | 91.80 | 34.91 |
| NECO | 92.13 | 30.16 | 89.92 | 46.49 | 91.95 | 40.11 | 98.99 | 4.12 | 91.45 | 39.80 | 92.89 | 32.14 |
| NECO† | 92.86 | 32.44 | 90.38 | 42.66 | 93.15 | 33.98 | 99.34 | 3.26 | 94.53 | 25.20 | 94.05 | 27.51 |
| Trusted | 43.56 | 86.45 | 46.82 | 96.95 | 50.95 | 94.75 | 49.36 | 91.48 | 39.15 | 95.45 | 45.97 | 93.02 |
| KL-matching | 87.85 | 40.92 | 86.76 | 53.02 | 87.89 | 49.19 | 97.84 | 8.84 | 86.25 | 49.20 | 89.32 | 40.23 |
| SFM (ours) | 96.65 | 11.70 | 89.64 | 46.00 | 92.04 | 37.78 | 99.40 | 2.26 | 93.76 | 29.80 | 94.30 | 25.51 |
| CLIP-VIT-B/16 | | | | | | | | | | | | |
| MSP | 83.05 | 57.59 | 79.83 | 68.39 | 79.33 | 70.29 | 89.74 | 41.95 | 78.60 | 71.00 | 82.11 | 61.84 |
| MLS | 88.76 | 45.43 | 86.02 | 57.05 | 86.39 | 58.28 | 95.57 | 23.45 | 86.53 | 61.15 | 88.65 | 49.07 |
| Energy | 89.26 | 44.01 | 86.59 | 54.39 | 87.12 | 54.85 | 96.38 | 17.67 | 87.32 | 58.30 | 89.33 | 45.84 |
| Mahalanobis | 85.05 | 66.49 | 84.34 | 72.06 | 85.15 | 75.37 | 90.35 | 65.00 | 80.71 | 79.00 | 85.12 | 71.58 |
| Residual | 76.25 | 80.05 | 75.64 | 88.95 | 75.40 | 91.87 | 71.20 | 94.15 | 67.87 | 88.10 | 73.27 | 88.62 |
| Vim | 89.30 | 44.20 | 86.70 | 54.49 | 87.22 | 55.21 | 96.17 | 18.83 | 87.17 | 59.25 | 89.31 | 46.40 |
| NECO | 88.77 | 47.02 | 87.86 | 52.40 | 88.61 | 53.92 | 95.24 | 25.30 | 85.29 | 64.00 | 89.15 | 48.53 |
| MCM† | 86.11 | 57.77 | 89.77 | 44.69 | 92.57 | 37.59 | 94.61 | 30.91 | 79.51 | 75.70 | 88.51 | 49.33 |
| Trusted | 95.87 | 19.80 | 74.59 | 78.06 | 76.71 | 76.42 | 84.61 | 72.77 | 84.12 | 62.40 | 83.18 | 61.89 |
| KL-matching | 86.64 | 46.45 | 83.28 | 59.25 | 83.21 | 61.23 | 94.18 | 24.99 | 83.19 | 62.45 | 86.10 | 50.87 |
| SFM (ours) | 89.16 | 48.44 | 91.88 | 36.46 | 93.24 | 34.87 | 95.47 | 23.49 | 82.27 | 70.35 | 90.41 | 42.72 |

Table 4: OOD detection performance on ImageNet-LT (ID) and five OOD datasets.

| Method | Texture | | Places | | SUN | | iNaturalist | | ImageNet-O | | Average | |
|---|---|---|---|---|---|---|---|---|---|---|---|---|
| | AUROC | FPR95 | AUROC | FPR95 | AUROC | FPR95 | AUROC | FPR95 | AUROC | FPR95 | AUROC | FPR95 |
| IMAGENET-21K PRE-TRAINED VIT | | | | | | | | | | | | |
| MSP | 86.04 | 47.50 | 85.20 | 56.52 | 86.36 | 53.07 | 97.17 | 11.97 | 83.68 | 57.40 | 87.69 | 45.29 |
| MLS | 90.18 | 38.71 | 88.76 | 49.34 | 90.39 | 45.44 | 98.47 | 6.71 | 88.91 | 47.90 | 91.34 | 37.62 |
| Energy | 90.87 | 35.51 | 89.29 | 45.97 | 91.05 | 41.04 | 98.78 | 5.06 | 89.83 | 42.80 | 91.96 | 34.08 |
| Mahalanobis | 92.99 | 26.95 | 89.48 | 46.34 | 91.71 | 38.35 | 99.28 | 2.84 | 91.66 | 38.85 | 93.02 | 30.67 |
| Residual | 91.60 | 35.74 | 82.23 | 65.71 | 86.58 | 55.54 | 97.44 | 12.67 | 84.05 | 59.05 | 88.38 | 45.74 |
| Vim | 91.23 | 34.10 | 89.47 | 45.23 | 91.27 | 39.83 | 98.88 | 4.77 | 90.05 | 41.70 | 92.18 | 33.13 |
| NECO | 91.66 | 31.44 | 89.21 | 43.71 | 91.44 | 37.07 | 98.93 | 4.09 | 89.64 | 42.70 | 92.18 | 31.80 |
| Trusted | 91.98 | 32.36 | 82.11 | 66.31 | 85.72 | 58.34 | 98.09 | 9.29 | 90.91 | 40.15 | 89.76 | 41.29 |
| KL-matching | 88.72 | 38.71 | 87.41 | 50.03 | 89.14 | 45.83 | 98.44 | 6.19 | 87.24 | 47.50 | 90.19 | 37.65 |
| SFM (ours) | 96.92 | 11.79 | 89.82 | 45.36 | 92.18 | 36.16 | 99.33 | 2.51 | 93.46 | 31.10 | 94.34 | 25.38 |
| CLIP-VIT-B/16 | | | | | | | | | | | | |
| MSP | 81.55 | 60.34 | 79.32 | 65.16 | 78.44 | 66.53 | 90.60 | 38.49 | 78.37 | 71.60 | 81.66 | 60.42 |
| MLS | 87.00 | 52.27 | 85.31 | 56.20 | 85.47 | 57.19 | 95.03 | 25.21 | 84.33 | 65.10 | 87.43 | 51.19 |
| Energy | 87.81 | 50.07 | 86.37 | 51.85 | 86.76 | 53.08 | 95.94 | 19.61 | 85.12 | 63.65 | 88.40 | 47.65 |
| Mahalanobis | 83.81 | 67.64 | 84.44 | 66.85 | 85.50 | 69.58 | 87.49 | 72.57 | 78.82 | 80.20 | 84.01 | 71.37 |
| Residual | 74.81 | 80.71 | 75.62 | 86.49 | 76.56 | 87.93 | 63.27 | 96.67 | 64.43 | 89.30 | 70.94 | 88.22 |
| Vim | 87.90 | 49.72 | 86.52 | 51.32 | 86.96 | 52.47 | 95.55 | 21.06 | 84.96 | 63.90 | 88.38 | 47.69 |
| NECO | 86.67 | 53.67 | 86.71 | 53.11 | 87.17 | 54.63 | 94.08 | 29.95 | 82.90 | 67.60 | 87.51 | 51.79 |
| Trusted | 71.96 | 70.46 | 44.51 | 97.89 | 49.78 | 97.77 | 49.44 | 98.59 | 48.79 | 89.05 | 52.90 | 90.75 |
| KL-matching | 85.35 | 51.56 | 82.84 | 57.00 | 82.51 | 57.56 | 94.54 | 23.36 | 82.52 | 64.00 | 85.55 | 50.70 |
| SFM (ours) | 90.95 | 43.10 | 92.43 | 34.80 | 92.62 | 39.07 | 94.59 | 28.54 | 83.62 | 68.90 | 90.84 | 42.88 |

**Result on ImageNet and ImageNet-LT.** Table 3 summarizes the performance of our proposed method, SFM, on the ImageNet dataset. Across both pre-trained models, namely, the ImageNet-21k pre-trained ViT and CLIP-ViT-B/16, SFM consistently outperforms existing methods. Specifically, when using the ImageNet-21k pre-trained ViT, SFM improves the FPR95 by more than 5% on average compared to the second-best method Mahalanobis (Lee et al., 2018). Notably, while MCM Ming et al. (2022) does not require fine-tuning, it achieves competitive performance across four OOD datasets, except ImageNet-O. Its overall average performance is on par with the Vim (Wang et al., 2022a) and NECO (Ammar et al., 2024) methods. However, SFM still outperforms MCM by ∼ 2% in AUROC and ∼ 7% in FPR95.

Additionally, Table 4 presents the results on the long-tailed in-distribution dataset. It can be seen that our method consistently outperforms previous approaches. On average, our method reduces FPR95 by 5.29% and 4.77% for ImageNet-21k pre-trained ViT and CLIP, respectively. The AUROC also improves by 2.44% when using the CLIP model.

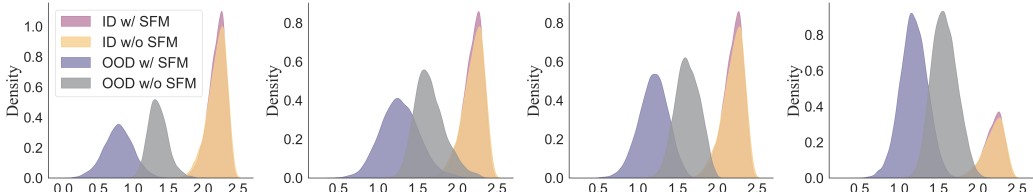

Figure 3: Comparisons of OOD score distribution before and after applying our SFM method. CIFAR-100 is used as the ID dataset and the OOD dataset from left to right is Texture, Tiny ImageNet, LSUN, and Places365. The horizontal axis represents the OOD score (small values indicate a high likelihood of being OOD samples).

### 4.3 ABLATION STUDIES

**Why SFM works?** Unless otherwise specified, in this subsection, we use the ImageNet-21k pretrained ViT as the default base model. Figure 3 presents a comparison of OOD score distributions with and without the application of our proposed SFM method. When SFM is not applied, only the final layer features are used to compute the Mahalanobis distance as a scoring function. It can be seen that the score distributions for ID samples remain largely consistent, whether or not the SFM method is applied. However, the use of SFM causes a significant leftward shift in the score distribution for OOD samples. This shift occurs because the features in the final layer of unseen OOD samples are not effectively captured. Furthermore, re-weighted information from the shallow layers amplifies this shift, resulting in better discrimination. As a result, the SFM method enhances the separation between ID and OOD samples in the embedding space. This improvement is critical for more accurate identification and differentiation of ID and OOD samples, thus boosting the overall performance and reliability of the detection process.

**Importance weights of each layer.** As depicted in Figure 4, our proposed method can adaptively assign importance weights to different layers. Overall, the first 4 layers are assigned relatively lower weights compared to the rest of the Transformer layers. Notably, the final layer's weight is particularly prominent. This is because the last layer of the feature extractor learns the most discriminative features for in-distribution classes and is important for OOD detection. As shown in the figure, rather than relying solely on the last layer's features, our method effectively utilizes shallow layer features as well.

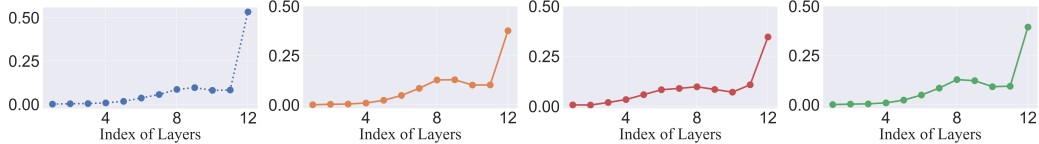

Figure 4: Distribution of layer-specific weights for CIFAR-100, ImageNet, ImageNet (CLIP), and ImageNet-LT where the y-axis denotes AUROC (%).

**Impact of features from shallow layers.** Figure 5 illustrates the effect of fusing features from varying numbers of layers. The x-axis represents the number of layers counted from the last layer towards the first, while the y-axis indicates the average OOD detection AUROC. As shown in the figures, using only the last layer's features yields decent results, but fusing the last 6 layers of the Transformer achieves the best performance, highlighting the importance of shallow features. For features from the sixth layer and beyond, their impact on the results is minimal. As discussed in the previous analysis, our method assigns lower weights to these layers accordingly.

**Way to fuse shallow features.** Certain methods of feature fusion have been implemented by modifying neural networks, like (Dai et al., 2021), (Li et al., 2020), and (Wei et al., 2021). (Xu et al., 2020) proposed to use different enhancements for feature fusion We compare our proposed attention-based feature fusion method with other fusion strategies including 1) Trusted (Colombo et al., 2022) which directly employs the arithmetic mean to fuse features from each layer during both the training and test phases; 2) Score Aggregation (SA) (Lee et al., 2018) which calculates the OOD score via Mahalanobis distance using features from each layer separately and weighted them together. Since SA

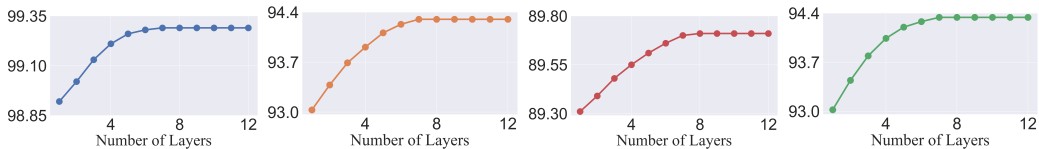

Figure 5: Impact of the number of layers used for feature fusion on OOD detection performance. The ID dataset from left to right is CIFAR-100, ImageNet, ImageNet (CLIP), and ImageNet-LT, where the vertical axis represents AUROC.

Table 5: Comparisons of different feature fusion strategies. 'In21k' denotes ViT pre-trained on ImageNet-21k.

| Method | CIFAR-100 | | | | ImageNet | | | | | |
| | CLIP | | In21k | | CLIP | | In21k | | Average | |
| | AUROC | FPR95 | AUROC | FPR95 | AUROC | FPR95 | AUROC | FPR95 | AUROC | FPR95 |
| --- | --- | --- | --- | --- | --- | --- | --- | --- | --- | --- |
| Trusted | 96.76 | 15.85 | 98.37 | 8.17 | 83.18 | 61.89 | 45.97 | 93.02 | 81.49 | 44.73 |
| SA | 96.53 | 13.19 | 98.77 | 6.58 | 82.68 | 64.59 | 94.01 | 27.62 | 93.00 | 28.00 |
| PM | 96.15 | 18.13 | 98.16 | 10.05 | 91.03 | 77.59 | 81.86 | 27.47 | 91.80 | 33.31 |
| Flatten12 | 42.10 | 89.67 | 29.00 | 90.93 | - | - | - | - | 34.05 | 90.15 |
| Flatten6 | 93.31 | 15.99 | 81.75 | 49.33 | - | - | - | - | 87.53 | 32.66 |
| Ours | **97.28** | **13.34** | **99.29** | **3.76** | **89.79** | **45.30** | **94.30** | **25.51** | **95.17** | **21.98** |

requires a validation set containing both ID and OOD data, we use the weights derived from our method to calculate the weighted sum of scores; 3) Power Mean (PM) (Rücklé et al., 2018) proposes to reweight each layer's feature based on feature norms; 4) Flatten12 concatenates all layers' features into a single vector, while Flatten6 concatenates the last six layers' features. The results are presented in Table 5. It can be seen that our proposed attention-based fusion method achieves a significant advantage in aggregating shallow features, further confirming its effectiveness.

**Additional time consumption analysis.** Unlike the direct Mahalanobis distance, which considers only the final layer of features, our approach necessitates the integration of features across all layers. This inevitably leads to additional time consumption. Table 6 presents the time consumption at different stages of the test phase, measured in seconds, on the ImageNet-LT dataset (ID) and the fine-tuned ViT model. "Pre-process" represents the process of pre-processing the ID training set, including the calculation of the mean and covariance matrix required for Mahalanobis distance, with additional importance weights $\alpha$ for SFM. Each subsequent column represents the time required to process each dataset including the ID test set and OOD datasets, and the last column represents the total time consumed. From the results, we observe that our approach only brings about an additional $10\%$ total time consumption, but results in an improvement of AUROC by $2.39\%$ and a reduction of FPR95 by $7.66\%$ on average, demonstrating the efficacy of our approach.

Table 6: Time consumption comparison between Mahalanobis and SFM.

| Dataset | Pre-process | ID test set | Texture | Places | SUN | iNaturalist | Imagenet-O | Total |
| --- | --- | --- | --- | --- | --- | --- | --- | --- |
| Mahalanobis | 685 | 238 | 36 | 61 | 56 | 59 | 14 | 1149 |
| SFM | 748 | 291 | 38 | 62 | 60 | 61 | 15 | 1275 |

## 5 CONCLUSION

In this paper, we propose a simple, yet effective attention-based feature fusion module for out-of-distribution detection. Our method calculates the importance of features of each layer from data, and weights them together accordingly. The OOD detection is achieved by calculating our proposed scoring function based on the Mahalanobis distance in the new feature space. To boost the features learned from pre-trained models, we also present a parameter-efficient fine-tuning framework and implement various OOD detection methods on top of it. Additionally, we extend our method to the commonly used vision-language model CLIP by incorporating image-text similarity score in the aligned space. The effectiveness of our method is shown by the fact that it achieves state-of-the-art results in both class-balanced and long-tailed out-of-distribution detection tasks.

**Ethics Statement.** This study introduces shallow feature matters (SFM) as an efficient tool for facilitating out-ofdistribution (OOD) solutions. By improving OOD detection, SFM has the potential to significantly enhance the dependability and safety of modern machine learning models. Thus, the social impact of this research can be far-reaching, spanning consumer and business applications in digital content understanding, transportation systems including driver assistance and autonomous vehicles, as well as healthcare applications such as identifying unseen diseases. Moreover, by openly sharing our code, we strive to offer machine learning practitioners a readily available resource for responsible AI development, ultimately benefiting society as a whole.

**Reproducibility Statement.** We have made significant efforts to ensure the reproducibility of our results. The source code used in our experiments is included in the supplementary materials, along with a detailed README file that provides step-by-step instructions and the necessary commands to reproduce the experiments. All the hyperparameters and experimental settings are clearly specified to facilitate replication.

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

## A  ADDITIONAL EXPERIMENT RESULTS

**In-distribution classification accuracy.**  Our fine-tuned model also shows strong ID classification performance, as detailed in Table 7. In terms of overall accuracy, both CIFAR-100 and ImageNet-1k perform better with balanced data compared to long-tailed data. This indicates that data balance positively impacts model performance, facilitating more accurate classification tasks.

When comparing different models, the pre-trained ViT consistently outperform CLIP-ViT-B/16 in most scenarios. This indicates that the pre-trained ViT has specific advantages for these data sets and tasks, suggesting that its pre-training approach is more suitable for these classification tasks, thereby also enhancing its efficacy in OOD detection tasks.

Table 7: Top 1% accuracy on ID data for the original classification task, for the models.

| ID dataset | Label distribution | Model | Accuracy (%) |
|---|---|---|---|
| CIFAR-100 | Zero-shot | CLIP-ViT-B/16 | 66.69 |
| | Long-tailed | CLIP-ViT-B/16 | 82.87 |
| | | Pre-trained ViT | 89.99 |
| | Balanced | CLIP-ViT-B/16 | 88.59 |
| | | Pre-trained ViT | 93.47 |
| ImageNet-1k | Zero-shot | CLIP-ViT-B/16 | 67.12 |
| | Long-tailed | CLIP-ViT-B/16 | 75.82 |
| | | Pre-trained ViT | 81.79 |
| | Balanced | CLIP-ViT-B/16 | 79.08 |
| | | Pre-trained ViT | 83.50 |

**Impact of hyper-parameter $\lambda$ in Eq. (7).**  To demonstrate the effect of the hyper-parameter $\lambda$, Table 8 showcases the OOD detection performance (AUROC) on the CLIP model with ImageNet-LT as the ID dataset. From the results, we observe that the performance is relatively poor when the scoring function does not integrate the zero-shot CLIP similarity score, i.e., $\lambda = 0$, decreasing from 90.84 to 86.75. Conversely, when $\lambda \neq 0$, our method demonstrates robustness to different values of $\lambda$. In our experiments, we directly set $\lambda = 1$ on the ImageNet dataset for simplicity.

Table 8: Sensitivity analysis of hyperparameter $\lambda$.

| $\lambda$ | 0 | 0.2 | 0.5 | 0.8 | 1 | 1.2 | 1.5 | 2 |
|---|---|---|---|---|---|---|---|---|
| AUROC | 86.75 | 89.31 | 90.52 | 90.81 | 90.84 | 90.81 | 90.72 | 90.52 |

**Fair comparsion with MCM.**  The MCM method is naturally better suited for zero-shot OOD tasks compared to fine-tuning tasks. The prevalent fine-tuning approach, which mainly targets the visual encoder, tends to disrupt the initial alignment between the visual and text components after fine-tuning, resulting in less effective outcomes. Our goal in including the MCM method in our experiment was not to make a direct comparison but to empirically showcase that our proposed method enhances OOD detection performance. Conversely, methods like ViM and NECO are methodologically and conceptually more similar to our approach and, therefore, require a more thorough comparison. Moreover, we present the results of MCM on the fine-tuned model (i.e. MCM-tuned) in the table below for comparison.

**Ablation studies on weights of different layers.**  To further emphasize the importance of differentiated layer weighting, we provide experimental tables (i.e., Table 10, 11, 12). In these table, we test different scenarios where the final layer is given weights of 0.083 (i.e., uniform), 0.5, 0.75, and 1 (which are represented by $W_{0.083}, W_{0.5}, W_{0.75}, W_{1.0}$), while the other layers receive the remaining weights evenly. These experiments clearly highlight the vital role different layer weightings play in boosting OOD detection performance and the potency of our proposed method. The findings demonstrate that the SFM approach consistently achieves either the top (bold) or the second-best

Table 9: Fair comparsion with MCM on CIFAR-100 and CIFAR-100-LT (ID).

| Method | Texture | | SVHN | | CIFAR10 | | Tiny ImageNet | | LSUN | | Places365 | | Average | |
| | AUROC | FPR95 | AUROC | FPR95 | AUROC | FPR95 | AUROC | FPR95 | AUROC | FPR95 | AUROC | FPR95 | AUROC | FPR95 |
|---|---|---|---|---|---|---|---|---|---|---|---|---|---|---|
| **CIFAR-100** | | | | | | | | | | | | | | |
| MCM-untuned | 72.98 | 92.09 | 90.75 | 63.39 | 75.53 | 88.66 | 65.54 | 93.36 | 50.79 | 99.11 | 60.97 | 97.79 | 69.43 | 89.06 |
| MCM-tuned | 75.33 | 91.38 | 91.55 | 60.96 | 75.60 | 91.03 | 64.07 | 95.40 | 55.14 | 98.93 | 63.71 | 97.67 | 70.90 | 89.23 |
| SFM (ours) | **99.95** | **0.02** | **98.31** | **8.62** | **88.56** | **53.97** | **97.54** | **12.91** | **99.93** | **0.06** | **99.95** | **0.02** | **97.37** | **12.60** |
| **CIFAR-100-LT** | | | | | | | | | | | | | | |
| MCM-untuned | 72.98 | 92.09 | 90.75 | 63.39 | 75.53 | 88.66 | 65.54 | 93.36 | 50.79 | 99.11 | 60.97 | 97.79 | 69.43 | 89.06 |
| MCM-tuned | 75.33 | 91.38 | 91.55 | 60.96 | 75.60 | 91.03 | 64.07 | 95.40 | 55.14 | 98.93 | 63.71 | 97.67 | 70.90 | 89.23 |
| SFM (ours) | **99.92** | **0.02** | **97.50** | **15.65** | **85.20** | **60.41** | **95.05** | **26.05** | **99.92** | **0.04** | **99.93** | **0.01** | **96.25** | **17.03** |

| Method | Texture | | Places | | SUN | | iNaturalist | | ImageNet-O | | Average | |
| | AUROC | FPR95 | AUROC | FPR95 | AUROC | FPR95 | AUROC | FPR95 | AUROC | FPR95 | AUROC | FPR95 |
|---|---|---|---|---|---|---|---|---|---|---|---|---|
| **IMAGENET-1K-LT** | | | | | | | | | | | | |
| MCM-untuned | 86.11 | 57.77 | 89.77 | 44.69 | 92.57 | 37.59 | **94.61** | 30.91 | 79.51 | 75.70 | 88.51 | 49.33 |
| MCM-tuned | 85.64 | 60.11 | 89.82 | 44.32 | **92.92** | **36.25** | 94.26 | 32.01 | 79.26 | 76.10 | 88.38 | 49.76 |
| SFM (ours) | **90.95** | **43.10** | **92.43** | **34.80** | 92.62 | 39.07 | 94.59 | **28.54** | **83.62** | **68.90** | **90.84** | **42.88** |

Table 10: Ablation studies on weights of different layers on CIFAR-100 (ID).

| Method | Texture | | SVHN | | CIFAR10 | | Tiny ImageNet | | LSUN | | Places365 | | Average | |
| | AUROC | FPR95 | AUROC | FPR95 | AUROC | FPR95 | AUROC | FPR95 | AUROC | FPR95 | AUROC | FPR95 | AUROC | FPR95 |
|---|---|---|---|---|---|---|---|---|---|---|---|---|---|---|
| **IMAGENET-21K PRE-TRAINED VIT** | | | | | | | | | | | | | | |
| 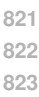 $W_{0.083}$ | 100.0 | 0.00 | **99.61** | **0.28** | 90.02 | 51.14 | **100.0** | **0.00** | 100.0 | 0.00 | 100.0 | 0.00 | 98.27 | 8.57 |
| $W_{0.5}$ | 100.0 | 0.00 | 99.43 | 2.36 | 96.76 | 18.26 | 99.62 | 1.81 | 100.0 | 0.00 | 99.99 | 0.01 | **99.30** | **3.74** |
| $W_{0.75}$ | 99.99 | 0.04 | 99.27 | 3.33 | 96.99 | 16.82 | 98.83 | 5.22 | 99.89 | 0.06 | 99.89 | 0.26 | 99.14 | 4.29 |
| $W_{1.0}$ | 99.97 | 0.12 | 99.16 | 3.92 | **97.09** | **16.49** | 97.99 | 8.96 | 99.61 | 1.07 | 99.67 | 1.33 | 98.92 | 5.32 |
| SFM (ours) | **100.0** | **0.00** | 99.50 | 1.91 | 96.47 | 19.52 | 99.80 | 1.11 | **100.0** | **0.00** | **100.0** | **0.00** | 99.29 | 3.76 |
| **CLIP-VIT-B/16** | | | | | | | | | | | | | | |
| $W_{0.083}$ | **100.0** | **0.00** | **99.05** | **2.82** | 83.92 | 65.74 | **99.94** | **0.08** | 100.0 | 0.00 | 100.0 | 0.00 | 97.15 | 11.44 |
| $W_{0.5}$ | 99.92 | 0.07 | 98.00 | 10.27 | 89.15 | 51.09 | 96.49 | 17.36 | 99.88 | 0.21 | 99.91 | 0.12 | 97.22 | 13.19 |
| $W_{0.75}$ | 99.61 | 0.85 | 97.84 | 12.34 | **89.36** | **50.66** | 94.38 | 26.56 | 99.24 | 3.85 | 99.55 | 1.68 | 96.66 | 15.99 |
| $W_{1.0}$ | 99.23 | 1.68 | 96.89 | 23.27 | 89.01 | 52.26 | 93.75 | 32.28 | 98.81 | 6.44 | 99.29 | 3.13 | 96.16 | 19.84 |
| SFM (ours) | 99.95 | 0.02 | 98.31 | 8.62 | 88.56 | 53.97 | 97.54 | 12.91 | 99.93 | 0.06 | 99.95 | 0.02 | **97.37** | 12.60 |

(underlined) position across all contexts. Remarkably, when ranked second, our method's performance closely rivals that of the first-place finisher. Specifically, our approach attains 96.17 and 17.82, whereas the uniform weight variant $W_{0.083}$ records 95.52 and 19.71, and $W_{0.05}$ achieves 95.82 and 19.04, on AUROC and FPR95 respectively.

**Ablation studies on smaller pre-trained transformers.** As depicted in Table 13, 14, and 15, we have included models like vit_tiny_patch16_224 and vit_small_patch16_224, shown in the upper and lower sections of each table. The outcomes from these smaller models provide further confirmation that our OOD score remains robust and effective across various model scales, thereby enhancing the generalizability and reliability of our proposed approach.

**Ablation studies on OpenOOD v1.5 benchmark.** We conducted our experiment again using the Openood v1.5 (Zhang et al., 2023) benchmark and chose Imagenet-1K-LT as the ID dataset, as shown in Table 16. From our experience, this approach is comparable to using ImageNet-1k while being more time-efficient. Our results surpassed those of all other methods by a significant margin on average, highlighting the success of our SFM strategy.

**Ablation studies on varying parameter-efficient fine-tuning methods.** SFM is a general framework in which many lightweight fine-tuning methods can be integrated. In addition to Adaptformer

Table 11: Ablation studies on weights of different layers on CIFAR-100-LT (ID).

| Method | Texture | | SVHN | | CIFAR10 | | Tiny ImageNet | | LSUN | | Places365 | | Average | |
| | AUROC | FPR95 | AUROC | FPR95 | AUROC | FPR95 | AUROC | FPR95 | AUROC | FPR95 | AUROC | FPR95 | AUROC | FPR95 |
|---|---|---|---|---|---|---|---|---|---|---|---|---|---|---|
| **IMAGENET-21K PRE-TRAINED VIT** | | | | | | | | | | | | | | |
| 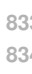 $W_{0.083}$ | 100.0 | 0.00 | **99.85** | **0.01** | 85.54 | 68.56 | **100.0** | **0.00** | 100.0 | 0.00 | 100.0 | 0.00 | 97.57 | 11.43 |
| $W_{0.5}$ | 100.0 | 0.00 | 99.68 | 0.68 | 94.56 | 27.90 | 99.58 | 1.68 | 99.99 | 0.00 | 99.98 | 0.01 | **98.97** | **5.05** |
| $W_{0.75}$ | 99.99 | 0.05 | 99.49 | 1.66 | 94.93 | 26.42 | 98.67 | 5.57 | 99.81 | 0.39 | 99.83 | 0.53 | 98.79 | 5.77 |
| $W_{1.0}$ | 99.96 | 0.20 | 99.33 | 2.51 | **95.09** | **25.98** | 97.63 | 9.26 | 99.48 | 2.26 | 99.57 | 1.71 | 98.51 | 6.99 |
| SFM (ours) | **100.0** | **0.00** | 99.75 | 0.43 | 94.22 | 29.86 | 99.75 | 1.12 | 99.99 | 0.00 | 99.99 | 0.01 | 98.95 | 5.24 |
| **CLIP-VIT-B/16** | | | | | | | | | | | | | | |
| 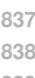 $W_{0.083}$ | **100.0** | **0.00** | **98.76** | **5.09** | 80.77 | 68.39 | **99.89** | **0.20** | 100.0 | 0.00 | 100.0 | 0.00 | 96.57 | 12.28 |
| $W_{0.5}$ | 99.88 | 0.04 | 97.37 | 16.90 | 85.52 | 58.97 | 94.36 | 28.65 | 99.87 | 0.11 | 99.89 | 0.05 | 96.15 | 17.45 |
| $W_{0.75}$ | 99.52 | 0.55 | 97.15 | 19.66 | **85.73** | 58.99 | 91.58 | 39.22 | 99.36 | 2.63 | 99.52 | 1.20 | 95.48 | 20.38 |
| $W_{1.0}$ | 99.11 | 1.03 | 95.92 | 29.87 | 84.76 | 60.58 | 90.97 | 43.83 | 99.08 | 4.07 | 99.28 | 1.99 | 94.85 | 23.5 |
| SFM (ours) | 99.92 | 0.02 | 97.50 | 15.65 | 85.20 | 60.41 | 95.05 | 26.05 | 99.92 | 0.04 | 99.93 | 0.01 | **96.25** | **17.03** |

Table 12: Ablation studies on weights of different layers on ImageNet-1k-LT (ID).

| Method | Texture | | Places | | SUN | | iNaturalist | | ImageNet-O | | Average | |
|---|---|---|---|---|---|---|---|---|---|---|---|---|
| | AUROC | FPR95 | AUROC | FPR95 | AUROC | FPR95 | AUROC | FPR95 | AUROC | FPR95 | AUROC | FPR95 |
| **IMAGENET-21K PRE-TRAINED VIT** | | | | | | | | | | | | |
| $W_{0.083}$ | **98.55** | **6.45** | 86.32 | 60.57 | 88.92 | 49.25 | 98.02 | 9.42 | 91.72 | 37.05 | 92.71 | 32.55 |
| $W_{0.5}$ | 95.02 | 17.96 | 89.76 | **44.79** | 92.08 | 36.60 | **99.36** | 2.63 | 92.68 | 34.30 | 93.78 | 27.26 |
| $W_{0.75}$ | 93.78 | 23.39 | 89.62 | 45.54 | 91.88 | 37.22 | 99.32 | 2.74 | 92.06 | 36.75 | 93.33 | 29.13 |
| $W_{1.0}$ | 92.99 | 26.95 | 89.48 | 46.34 | 91.71 | 38.35 | 99.28 | 2.84 | 91.66 | 38.85 | 93.02 | 30.67 |
| SFM (ours) | 96.92 | 11.79 | **89.82** | 45.36 | **92.18** | **36.16** | 99.33 | **2.51** | **93.46** | **31.10** | **94.34** | **25.38** |
| **CLIP-VIT-B/16** | | | | | | | | | | | | |
| $W_{0.083}$ | **92.23** | **36.76** | 91.11 | 39.65 | **93.02** | **36.38** | **94.62** | 29.54 | 83.30 | 67.45 | **90.86** | **41.96** |
| $W_{0.5}$ | 88.52 | 52.23 | 89.87 | 45.47 | 92.21 | 40.58 | 94.54 | **28.45** | 82.56 | 71.00 | 89.54 | 47.55 |
| $W_{0.75}$ | 87.68 | 55.39 | 89.69 | 46.37 | 92.08 | 41.23 | 94.41 | 29.30 | 82.23 | 72.40 | 89.22 | 48.94 |
| $W_{1.0}$ | 83.81 | 67.64 | 84.44 | 66.85 | 85.50 | 69.58 | 87.49 | 72.57 | 78.82 | 80.20 | 84.01 | 71.37 |
| SFM (ours) | 90.95 | 43.10 | **92.43** | **34.80** | 92.62 | 39.07 | 94.59 | 28.54 | **83.62** | 68.90 | 90.84 | 42.88 |

Table 13: OOD detection performance on CIFAR-100 (ID) on smaller transformers.

| Method | Texture | | SVHN | | CIFAR10 | | Tiny ImageNet | | LSUN | | Places365 | | Average | |
|---|---|---|---|---|---|---|---|---|---|---|---|---|---|---|
| | AUROC | FPR95 | AUROC | FPR95 | AUROC | FPR95 | AUROC | FPR95 | AUROC | FPR95 | AUROC | FPR95 | AUROC | FPR95 |
| **VIT_TINY_PATCH16_224** | | | | | | | | | | | | | | |
| MSP | 92.09 | 35.34 | 83.28 | 61.28 | 83.30 | 63.73 | 79.89 | 69.73 | 72.86 | 84.07 | 82.13 | 65.55 | 82.26 | 63.28 |
| MLS | 98.62 | 6.13 | 92.09 | 35.85 | 87.39 | 54.92 | 87.71 | 52.79 | 88.92 | 57.57 | 94.27 | 30.32 | 91.50 | 39.60 |
| Energy | 99.03 | 4.26 | 92.78 | 32.41 | 87.28 | 55.96 | 88.18 | 51.27 | 90.32 | 51.04 | 95.25 | 24.94 | 92.14 | 36.65 |
| Mahalanobis | 99.90 | 0.35 | 96.28 | 15.78 | 87.78 | 56.67 | 92.48 | 33.81 | 98.20 | 9.10 | 98.77 | 6.27 | 95.57 | 20.33 |
| Residual | 99.71 | 0.85 | 86.24 | 52.70 | 76.86 | 72.72 | 70.89 | 42.55 | 97.46 | 14.02 | 97.25 | 13.66 | 91.40 | 32.75 |
| Vim | 99.19 | 3.62 | 92.99 | 31.18 | 87.49 | 54.90 | 88.70 | 48.96 | 91.14 | 47.63 | 95.67 | 22.63 | 92.53 | 34.82 |
| NECO | 99.17 | 3.83 | 92.34 | 34.24 | **87.85** | **53.47** | 89.47 | 46.45 | 92.38 | 43.05 | 96.06 | 21.11 | 92.88 | 33.69 |
| KL-matching | 95.41 | 18.40 | 87.58 | 45.71 | 86.32 | 53.75 | 82.47 | 60.45 | 75.28 | 79.24 | 85.65 | 52.82 | 85.45 | 51.73 |
| SFM (ours) | **100.0** | **0.02** | **96.85** | **14.13** | 86.48 | 60.56 | **97.00** | **15.34** | **99.98** | **0.01** | **99.96** | **0.10** | **96.71** | **15.03** |
| **VIT_SMALL_PATCH16_224** | | | | | | | | | | | | | | |
| MSP | 95.98 | 19.17 | 92.29 | 38.18 | 90.82 | 39.01 | 85.95 | 52.36 | 82.84 | 68.92 | 89.31 | 47.87 | 89.53 | 44.25 |
| MLS | 99.28 | 3.16 | 96.35 | 18.16 | 95.22 | 24.90 | 92.18 | 32.72 | 96.21 | 25.40 | 97.71 | 13.44 | 96.16 | 19.63 |
| Energy | 99.48 | 2.29 | 96.54 | 16.55 | 95.42 | 23.24 | 92.59 | 29.99 | 97.12 | 18.57 | 98.25 | 10.09 | 96.57 | 16.79 |
| Mahalanobis | 99.91 | 0.59 | 99.05 | 4.72 | 94.65 | 28.65 | 97.53 | 11.36 | 99.60 | 1.78 | 99.54 | 2.52 | 98.38 | 8.27 |
| Residual | 99.96 | 0.11 | 98.60 | 7.06 | 88.66 | 52.27 | 98.09 | 9.68 | 99.65 | 0.75 | 99.67 | 1.14 | 97.44 | 11.83 |
| Vim | 99.56 | 1.99 | 96.88 | 14.63 | **95.46** | **23.06** | 93.17 | 27.68 | 97.52 | 16.28 | 98.47 | 8.82 | 96.84 | 15.41 |
| NECO | 99.50 | 2.16 | 96.76 | 15.91 | 95.33 | 24.33 | 93.49 | 26.62 | 97.26 | 17.04 | 98.29 | 9.75 | 96.77 | 15.97 |
| KL-matching | 97.66 | 9.24 | 94.75 | 22.21 | 93.18 | 28.02 | 88.04 | 40.26 | 85.27 | 55.47 | 91.73 | 33.41 | 91.77 | 31.43 |
| SFM (ours) | **100.0** | **0.00** | **99.36** | **3.16** | 94.09 | 31.35 | **99.47** | **2.69** | **99.99** | **0.01** | **99.99** | **0.01** | **98.82** | **6.20** |

(Chen et al., 2022) which is used in our experiments by default, we test SFM with another 5 parameter-efficient fine-tuning (PEFT) methods as well as full fine-tuning. Specifically, we combine SFM with *Bias-tuning* (Zaken et al., 2021), *VPT-shallow* (Jia et al., 2022), *VPT-deep* (Jia et al., 2022), *LoRA* (Hu et al., 2022), and *Adapter* (Houlsby et al., 2019). We report the empirical results for CIFAR-100 in Table 17, CIFAR-100-LT in Table 18, and ImageNet-LT in Table 19. From the results, we observe that SFM consistently improves the baselines by a large margin, showing its robustness to the PEFT methods.

Table 14: OOD detection performance on CIFAR-100-LT (ID) on smaller transformers.

| Method | Texture | | SVHN | | CIFAR10 | | Tiny ImageNet | | LSUN | | Places365 | | Average | |
|---|---|---|---|---|---|---|---|---|---|---|---|---|---|---|
| | AUROC | FPR95 | AUROC | FPR95 | AUROC | FPR95 | AUROC | FPR95 | AUROC | FPR95 | AUROC | FPR95 | AUROC | FPR95 |
| **VIT_TINY_PATCH16_224** | | | | | | | | | | | | | | |
| MSP | 90.08 | 43.37 | 81.70 | 67.65 | 79.48 | 71.62 | 75.96 | 75.24 | 71.36 | 84.26 | 79.41 | 71.40 | 79.66 | 68.92 |
| MLS | 99.12 | 3.60 | 93.33 | 33.93 | 79.87 | 74.86 | 85.81 | 56.68 | 93.82 | 33.03 | 96.58 | 18.97 | 91.42 | 36.85 |
| Energy | 99.38 | 2.13 | 93.83 | 31.25 | 78.33 | 78.86 | 86.24 | 56.10 | 95.32 | 24.89 | 97.50 | 13.26 | 91.77 | 34.42 |
| Mahalanobis | 99.85 | 0.53 | 97.29 | 12.80 | **85.08** | **63.78** | 91.26 | 35.45 | 98.44 | 8.10 | 98.67 | 6.67 | 95.10 | 21.22 |
| Residual | 99.36 | 2.70 | 85.09 | 63.99 | 63.92 | 86.08 | 86.92 | 56.50 | 95.55 | 23.38 | 95.84 | 23.68 | 88.68 | 42.72 |
| Vim | 99.48 | 1.86 | 94.02 | 30.06 | 78.59 | 78.29 | 86.75 | 53.95 | 95.67 | 22.96 | 97.70 | 12.17 | 92.03 | 33.21 |
| NECO | 99.43 | 2.16 | 93.71 | 31.78 | 80.42 | 73.09 | 87.26 | 50.52 | 95.19 | 24.22 | 97.42 | 14.01 | 92.24 | 32.63 |
| KL-matching | 94.50 | 23.48 | 86.54 | 55.21 | 82.74 | 62.81 | 79.16 | 66.66 | 74.56 | 80.30 | 83.53 | 60.59 | 83.51 | 58.18 |
| SFM (ours) | **99.99** | **0.04** | **97.77** | **10.81** | 83.67 | 66.99 | **96.31** | **16.67** | **99.97** | **0.01** | **99.95** | **0.14** | **96.28** | **15.78** |
| **VIT_SMALL_PATCH16_224** | | | | | | | | | | | | | | |
| MSP | 96.39 | 16.72 | 92.72 | 37.39 | 87.58 | 49.60 | 82.39 | 57.10 | 80.54 | 68.09 | 87.52 | 49.64 | 87.85 | 46.42 |
| MLS | 99.69 | 1.44 | 95.97 | 21.84 | 91.96 | 41.67 | 92.62 | 29.45 | 97.66 | 14.61 | 98.77 | 6.74 | 96.11 | 19.29 |
| Energy | 99.80 | 1.13 | 95.29 | 27.28 | 91.55 | 45.20 | 93.37 | 25.75 | 98.58 | 8.86 | 99.28 | 3.78 | 96.31 | 18.67 |
| Mahalanobis | 99.91 | 0.53 | 99.43 | 2.35 | **93.02** | **35.98** | 97.15 | 12.93 | 99.59 | 2.35 | 99.65 | 1.67 | 98.12 | 9.30 |
| Residual | 99.93 | 0.25 | 96.22 | 24.49 | 83.28 | 64.13 | 95.96 | 21.20 | 99.26 | 3.78 | 99.37 | 2.78 | 95.67 | 19.44 |
| Vim | 99.84 | 0.96 | 95.74 | 25.12 | 91.69 | 44.68 | 93.78 | 26.50 | 98.77 | 7.65 | 99.38 | 3.29 | 96.53 | 17.72 |
| NECO | 99.77 | 1.13 | 96.30 | 20.92 | 91.64 | 41.49 | 92.86 | 27.03 | 97.67 | 14.01 | 98.92 | 5.90 | 96.19 | 18.41 |
| KL-matching | 98.12 | 7.73 | 95.63 | 20.27 | 90.16 | 37.62 | 85.25 | 46.40 | 83.94 | 56.00 | 90.77 | 36.06 | 90.65 | 34.01 |
| SFM (ours) | **100.0** | **0.00** | **99.68** | **1.05** | 92.24 | 39.11 | **99.48** | **2.44** | **100.0** | **0.00** | **100.0** | **0.00** | **98.57** | **7.10** |

Table 15: OOD detection performance on ImageNet-LT (ID) on smaller transformers.

| Method | Texture | | Places | | SUN | | iNaturalist | | ImageNet-O | | Average | |
|---|---|---|---|---|---|---|---|---|---|---|---|---|
| | AUROC | FPR95 | AUROC | FPR95 | AUROC | FPR95 | AUROC | FPR95 | AUROC | FPR95 | AUROC | FPR95 |
| **VIT_TINY_PATCH16_224** | | | | | | | | | | | | |
| MSP | 78.01 | 73.35 | 75.50 | 78.45 | 75.30 | 79.07 | 87.21 | 54.52 | 67.95 | 87.70 | 76.80 | 74.62 |
| MLS | 84.44 | 65.73 | 78.50 | 75.84 | 79.44 | 75.44 | 91.83 | 46.49 | 76.83 | 84.55 | 82.21 | 69.61 |
| Energy | 85.85 | 60.04 | 78.76 | 74.90 | 80.02 | 74.11 | 92.72 | 42.10 | 78.73 | 81.55 | 83.22 | 66.54 |
| Mahalanobis | 89.61 | 41.86 | **79.27** | **67.26** | **82.44** | **63.88** | 97.62 | 11.83 | 80.09 | 77.30 | 85.81 | 52.43 |
| Residual | 84.86 | 56.21 | 68.21 | 86.03 | 69.96 | 84.55 | 88.63 | 49.79 | 73.94 | 77.95 | 77.12 | 70.91 |
| Vim | 86.49 | 57.06 | 78.97 | 74.35 | 80.27 | 73.02 | 93.25 | 38.66 | 79.22 | 80.85 | 83.64 | 64.79 |
| NECO | 86.84 | 56.44 | 79.03 | 73.47 | 80.61 | 72.26 | 94.78 | 30.01 | 79.54 | 79.75 | 84.16 | 52.39 |
| KL-matching | 81.97 | 67.22 | 77.80 | 74.27 | 78.06 | 73.72 | 91.59 | 41.18 | 72.78 | 84.35 | 80.44 | 68.15 |
| SFM (ours) | **92.21** | **29.84** | 78.40 | 68.87 | 81.25 | 66.86 | **97.72** | **11.30** | **82.43** | **69.90** | **86.40** | **49.35** |
| **VIT_SMALL_PATCH16_224** | | | | | | | | | | | | |
| MSP | 82.60 | 60.11 | 81.41 | 66.59 | 81.97 | 63.84 | 94.31 | 25.69 | 77.74 | 73.60 | 83.61 | 57.97 |
| MLS | 87.94 | 50.51 | 84.97 | 60.73 | 86.46 | 56.44 | 96.63 | 17.04 | 84.64 | 65.40 | 88.13 | 50.02 |
| Energy | 88.96 | 46.03 | 85.45 | 58.10 | 87.20 | 53.27 | 97.09 | 14.06 | 85.91 | 60.85 | 88.92 | 46.46 |
| Mahalanobis | 91.13 | 36.06 | **86.30** | **54.87** | 89.54 | 46.81 | 99.03 | 4.49 | 87.74 | 55.45 | 90.75 | 39.54 |
| Residual | 88.66 | 45.11 | 79.43 | 70.30 | 84.29 | 60.89 | 96.18 | 20.30 | 82.07 | 65.85 | 86.12 | 52.49 |
| Vim | 89.38 | 44.08 | 85.72 | 56.97 | 87.57 | 52.06 | 97.39 | 12.38 | 86.29 | 59.25 | 89.27 | 44.95 |
| NECO | 89.72 | 43.40 | 85.86 | 56.63 | 88.18 | 51.50 | 98.07 | 9.36 | 87.00 | 58.40 | 89.77 | 43.86 |
| KL-matching | 86.01 | 50.67 | 83.63 | 60.68 | 84.80 | 56.71 | 96.68 | 14.62 | 81.90 | 65.35 | 86.60 | 49.61 |
| SFM (ours) | **93.35** | **25.94** | 86.18 | 55.21 | 89.52 | **46.74** | **99.13** | **3.99** | **89.13** | **50.20** | **91.46** | **36.42** |

Table 16: OOD detection performance on ImageNet-LT (ID) on OpenOOD v1.5.

| Method | Ninco | | Openimage-o | | Ssb-hard | | iImageNet-c | | ImageNet-es | | iImageNet-r | | ImageNet-v2 | | Average | |
|---|---|---|---|---|---|---|---|---|---|---|---|---|---|---|---|---|
| | AUROC | FPR95 | AUROC | FPR95 | AUROC | FPR95 | AUROC | FPR95 | AUROC | FPR95 | AUROC | FPR95 | AUROC | FPR95 | AUROC | FPR95 |
| **IMAGENET-21K PRE-TRAINED VIT** | | | | | | | | | | | | | | | | |
| MSP | 87.81 | 50.02 | 93.72 | 27.51 | 76.72 | 68.43 | 67.91 | 78.58 | 69.35 | 69.26 | 79.73 | 59.15 | 57.57 | 89.92 | 76.12 | 63.27 |
| MLS | 91.59 | 42.80 | 96.28 | 18.65 | 81.25 | 63.86 | 70.54 | 76.77 | 72.11 | 66.79 | 83.65 | 53.22 | 57.86 | **90.09** | 79.04 | 58.88 |
| Energy | 92.12 | 39.62 | 96.80 | 16.02 | 81.87 | 61.53 | 70.80 | 76.02 | 72.44 | 66.06 | 84.25 | 50.52 | 57.79 | 90.19 | 79.44 | 57.14 |
| Mahalanobis | 94.00 | 32.51 | 97.58 | 12.61 | 85.01 | 52.17 | 73.93 | 72.64 | 73.04 | 67.08 | 85.32 | 48.95 | 58.02 | 90.81 | 80.99 | 53.83 |
| Residual | 83.87 | 62.45 | 92.41 | 33.88 | 84.87 | 56.19 | 74.96 | 78.03 | 65.25 | 82.87 | 75.05 | 76.46 | 53.03 | 94.38 | 75.63 | 69.18 |
| Vim | 92.29 | 38.65 | 96.94 | 15.32 | 82.36 | 60.40 | 71.19 | 75.39 | 72.47 | 65.98 | 84.37 | 50.20 | 57.79 | 90.15 | 79.63 | 56.58 |
| NECO | 91.97 | 38.09 | 96.90 | 15.19 | 84.81 | 54.96 | 70.55 | 75.48 | 72.01 | 67.61 | 82.43 | 53.86 | 56.86 | 90.44 | 79.36 | 56.52 |
| KL-matching | 90.53 | 41.63 | 95.95 | 18.15 | 79.52 | 63.02 | 70.03 | 75.91 | 71.54 | 66.35 | 82.56 | 52.60 | 58.33 | 89.85 | 78.35 | 58.22 |
| SFM (ours) | **94.98** | **26.74** | **98.21** | **9.72** | **86.34** | 49.46 | **83.96** | 53.57 | **76.78** | 63.45 | **88.49** | 42.85 | 58.36 | 91.36 | **83.88** | 48.16 |
| **CLIP-VIT-B/16** | | | | | | | | | | | | | | | | |
| MSP | 80.11 | 68.94 | 88.22 | 46.72 | 68.06 | 83.66 | 73.44 | 70.44 | 70.31 | 68.46 | 77.27 | 64.00 | 57.12 | 90.78 | 73.50 | 70.43 |
| MLS | **84.17** | 67.11 | 92.93 | 35.17 | 71.99 | 82.44 | 77.66 | 67.65 | 75.66 | 64.21 | 84.61 | 55.96 | **58.24** | 90.28 | 77.89 | 66.13 |
| Energy | 84.15 | 67.78 | **93.73** | 30.59 | 72.25 | 82.43 | 78.02 | 67.13 | 76.51 | 62.51 | 85.87 | 52.26 | 58.22 | 90.28 | 78.39 | 64.71 |
| Mahalanobis | 75.13 | 83.28 | 86.95 | 63.82 | 66.11 | 89.49 | 82.68 | 62.64 | 84.27 | 52.46 | **90.02** | 47.33 | 58.18 | 90.31 | 77.62 | 69.90 |
| Residual | 61.56 | 91.54 | 70.43 | 81.35 | 61.00 | 92.63 | 82.86 | 68.28 | 86.30 | 54.65 | 86.28 | 57.41 | 56.47 | 92.06 | 72.13 | 76.85 |
| Vim | 83.91 | 68.20 | 93.57 | 31.15 | **72.28** | 82.57 | 78.85 | 65.06 | 77.52 | 60.34 | 86.73 | **49.79** | 58.35 | 89.88 | 78.74 | 63.86 |
| NECO | 880.96 | 71.90 | 92.48 | 37.13 | 69.22 | 85.10 | 77.84 | 68.17 | 78.69 | 61.76 | 86.00 | 54.92 | 58.18 | 89.83 | 77.63 | 66.97 |
| KL-matching | 83.21 | **66.88** | 92.13 | 34.08 | 70.63 | 81.75 | 76.09 | 67.07 | 72.92 | 64.25 | 81.70 | 55.46 | 57.83 | 90.51 | 76.36 | 65.71 |
| SFM (ours) | 8.30 | 74.67 | 92.90 | 36.93 | 71.64 | **81.79** | **85.49** | **50.94** | **87.84** | **43.49** | 88.76 | 50.31 | 58.22 | **89.50** | **80.45** | **61.09** |

Table 17: OOD detection performance in terms of AUROC (↑) and FPR95 (↓) for different PEFT methods, and full fine-tuning on CIFAR-100 dataset.

| Method | Texture AUROC | Texture FPR95 | SVHN AUROC | SVHN FPR95 | CIFAR10 AUROC | CIFAR10 FPR95 | Tiny ImageNet AUROC | Tiny ImageNet FPR95 | LSUN AUROC | LSUN FPR95 | Places AUROC | Places FPR95 | Average AUROC | Average FPR95 |
|---|---|---|---|---|---|---|---|---|---|---|---|---|---|---|
| **Bias-tuning** | | | | | | | | | | | | | | |
| + MSP | 97.46 | 12.77 | 94.72 | 27.13 | 94.19 | 29.44 | 88.42 | 44.58 | 86.25 | 64.76 | 91.93 | 41.80 | 92.16 | 36.75 |
| + MLS | 99.71 | 1.24 | 96.60 | 13.31 | 96.96 | 14.23 | 94.69 | 22.00 | 98.07 | 10.86 | 98.73 | 6.84 | 97.46 | 11.41 |
| + Energy | 99.82 | 0.87 | 96.62 | 12.60 | 97.07 | 13.58 | 95.10 | 20.07 | 98.72 | 6.86 | 99.11 | 4.74 | 97.74 | 9.79 |
| + Mahalanobis | 99.93 | 0.34 | 98.80 | 5.32 | 96.18 | 20.60 | 97.31 | 10.71 | 99.58 | 1.32 | 99.45 | 2.68 | 98.54 | 6.83 |
| + Residual | 99.98 | 0.04 | 97.20 | 16.92 | 90.96 | 49.13 | 98.46 | 6.69 | 99.90 | 0.14 | 99.81 | 0.59 | 97.72 | 12.25 |
| + Vim | 99.85 | 0.74 | 96.84 | 11.84 | 97.09 | 13.58 | 95.48 | 18.94 | 98.96 | 5.21 | 99.24 | 3.90 | 97.91 | 9.04 |
| + NECO | 99.77 | 1.21 | 97.03 | 12.42 | 96.95 | 15.77 | 94.97 | 20.55 | 98.34 | 10.19 | 98.84 | 6.58 | 97.65 | 11.12 |
| + SFM (ours) | 100.0 | 0.00 | 99.44 | 2.29 | 95.32 | 25.35 | 99.68 | 1.52 | 99.99 | 0.01 | 99.99 | 0.02 | **99.07** | **4.87** |
| **VPT-shallow** | | | | | | | | | | | | | | |
| + MSP | 95.84 | 18.09 | 93.78 | 34.50 | 92.09 | 37.35 | 85.90 | 49.15 | 79.15 | 78.05 | 87.17 | 54.51 | 88.99 | 45.27 |
| + MLS | 98.77 | 5.28 | 96.55 | 18.36 | 94.42 | 25.10 | 86.29 | 47.31 | 88.68 | 59.55 | 92.94 | 35.22 | 92.24 | 31.81 |
| + Energy | 99.04 | 4.57 | 96.58 | 15.87 | 94.42 | 24.75 | 85.83 | 51.00 | 89.64 | 55.53 | 93.47 | 32.64 | 93.16 | 30.73 |
| + Mahalanobis | 99.97 | 0.18 | 92.41 | 44.63 | 93.84 | 32.15 | 98.04 | 9.23 | 99.77 | 0.88 | | | 97.31 | 14.54 |
| + Residual | 99.98 | 0.05 | 80.46 | 67.16 | 86.92 | 55.54 | 99.02 | 5.17 | 99.95 | 0.10 | 99.89 | 0.37 | 94.37 | 21.40 |
| + Vim | 99.29 | 3.62 | 96.71 | 15.16 | 94.57 | 24.64 | 87.34 | 45.83 | 96.71 | 46.32 | 94.70 | 26.97 | 94.04 | 27.09 |
| + NECO | 99.30 | 3.56 | 95.99 | 25.70 | 95.02 | 24.55 | 90.59 | 34.88 | 94.24 | 34.05 | 96.17 | 20.60 | 95.22 | 23.89 |
| + SFM (ours) | 100.0 | 0.00 | 94.28 | 36.41 | 92.37 | 38.48 | 99.78 | 1.15 | 99.99 | 0.01 | 99.98 | 0.06 | **97.73** | **12.68** |
| **VPT-deep** | | | | | | | | | | | | | | |
| + MSP | 97.43 | 13.49 | 91.72 | 44.53 | 94.33 | 30.07 | 86.93 | 48.16 | 84.23 | 69.02 | 91.10 | 47.98 | 90.79 | 42.21 |
| + MLS | 99.69 | 12.49 | 96.55 | 15.98 | 91.21 | 30.65 | 95.81 | 25.68 | 97.51 | 13.26 | 97.51 | 13.26 | 96.34 | 16.59 |
| + Energy | 99.79 | 1.12 | 97.59 | 10.49 | 96.53 | 15.82 | 91.43 | 29.60 | 97.95 | 11.21 | | | 96.64 | 14.91 |
| + Mahalanobis | 99.94 | 0.30 | 94.27 | 39.67 | 96.08 | 22.59 | 97.10 | 162.99 | 99.08 | 4.88 | 99.16 | 4.47 | 97.60 | 14.15 |
| + Residual | 99.97 | 0.04 | 91.25 | 53.35 | 89.88 | 50.67 | 98.07 | 10.16 | 99.58 | 0.74 | 99.58 | 1.64 | 96.41 | 19.43 |
| + Vim | 99.83 | 0.83 | 97.68 | 10.13 | 96.57 | 15.91 | 92.09 | 27.75 | 97.05 | 18.26 | 98.22 | 9.96 | 96.91 | 13.81 |
| + NECO | 99.72 | 1.44 | 96.53 | 17.02 | 96.71 | 17.06 | 92.73 | 26.10 | 96.80 | 20.68 | 98.03 | 11.12 | 96.75 | 15.57 |
| + SFM (ours) | 99.99 | 0.02 | 96.36 | 25.31 | 95.33 | 26.39 | 99.59 | 2.03 | 99.95 | 0.00 | 99.93 | 0.18 | **98.52** | **8.99** |
| **LoRA** | | | | | | | | | | | | | | |
| + MSP | 97.36 | 12.77 | 94.85 | 29.23 | 94.36 | 29.49 | 87.26 | 46.89 | 84.76 | 68.83 | 90.95 | 45.35 | 91.59 | 38.76 |
| + MLS | 99.57 | 1.91 | 97.88 | 8.89 | 96.98 | 14.76 | 89.51 | 34.34 | 95.70 | 27.11 | 97.68 | 12.53 | 96.22 | 16.59 |
| + Energy | 99.68 | 1.38 | 98.09 | 7.79 | 97.09 | 14.28 | 89.57 | 34.75 | 96.37 | 23.26 | 98.09 | 10.67 | 96.48 | 15.36 |
| + Mahalanobis | 99.96 | 0.11 | 99.33 | 2.69 | 96.65 | 17.98 | 97.72 | 9.47 | 99.47 | 2.07 | 99.47 | 2.35 | 98.76 | 5.78 |
| + Residual | 99.99 | 0.02 | 98.15 | 9.65 | 91.25 | 44.12 | 98.85 | 4.83 | 99.84 | 0.14 | 99.80 | 0.45 | 97.98 | 9.87 |
| + Vim | 99.75 | 1.13 | 98.29 | 6.91 | 97.121 | 14.25 | 90.50 | 32.38 | 96.96 | 20.00 | 98.38 | 9.27 | 96.83 | 13.99 |
| + NECO | 99.69 | 1.67 | 98.43 | 6.15 | 96.98 | 15.96 | 91.95 | 27.74 | 96.66 | 21.84 | 98.17 | 9.97 | 96.98 | 13.89 |
| + SFM (ours) | 100.0 | 0.00 | 99.78 | 0.88 | 95.99 | 21.52 | 99.82 | 0.96 | 99.99 | 0.00 | 99.99 | 0.01 | **99.26** | **3.89** |
| **Adapter** | | | | | | | | | | | | | | |
| + MSP | 97.34 | 12.54 | 95.56 | 23.93 | 91.73 | 38.80 | 85.30 | 48.04 | 84.70 | 62.47 | 90.66 | 42.81 | 90.88 | 38.10 |
| + MLS | 99.90 | 0.32 | 98.31 | 8.01 | 94.26 | 29.06 | 92.18 | 29.58 | 99.10 | 3.69 | 99.50 | 1.51 | 97.21 | 12.03 |
| + Energy | 99.93 | 0.18 | 98.28 | 7.56 | 94.67 | 34.00 | 92.43 | 28.42 | 99.71 | 0.61 | 99.70 | 1.43 | 97.25 | 12.03 |
| + Mahalanobis | 99.97 | 0.12 | 99.44 | 1.82 | 95.04 | 26.43 | 97.58 | 9.60 | 99.55 | 1.76 | 99.63 | 1.36 | 98.53 | 6.85 |
| + Residual | 99.98 | 0.02 | 97.78 | 14.80 | 86.21 | 64.68 | 98.51 | 6.91 | 99.85 | 0.47 | 99.84 | 0.36 | 97.03 | 14.54 |
| + Vim | 99.95 | 0.16 | 98.48 | 6.36 | 93.77 | 33.18 | 93.09 | 26.10 | 99.57 | 0.98 | 99.75 | 0.45 | 97.44 | 11.20 |
| + NECO | 99.90 | 0.30 | 98.60 | 6.92 | 94.41 | 27.76 | 92.31 | 26.44 | 98.91 | 6.70 | 99.45 | 2.68 | 97.26 | 11.82 |
| + SFM (ours) | 100.0 | 0.00 | 99.50 | 2.54 | 96.79 | 18.07 | 99.72 | 1.50 | 100.0 | 0.00 | 100.0 | 0.00 | **99.33** | **3.68** |
| **Full fine-tuning** | | | | | | | | | | | | | | |
| + MSP | 97.24 | 15.39 | 91.45 | 46.78 | 93.64 | 33.62 | 87.79 | 48.74 | 85.44 | 72.87 | 91.58 | 48.41 | 91.19 | 44.30 |
| + MLS | 99.72 | 1.12 | 90.65 | 36.84 | 96.55 | 16.03 | 90.43 | 30.61 | 97.84 | 11.13 | 98.97 | 3.63 | 95.69 | 15.56 |
| + Energy | 99.76 | 0.89 | 90.44 | 38.61 | 96.57 | 15.89 | 90.47 | 30.36 | 98.11 | 9.60 | 99.13 | 3.08 | 95.75 | 16.40 |
| + Mahalanobis | 99.87 | 0.55 | 96.80 | 16.06 | 96.87 | 15.38 | 97.46 | 13.26 | 97.69 | 16.25 | 98.96 | 6.61 | 97.94 | 11.35 |
| + Residual | 99.98 | 0.12 | 98.13 | 9.62 | 95.11 | 26.57 | 99.13 | 4.86 | 99.70 | 1.04 | 99.86 | 0.54 | **98.65** | **7.13** |
| + Vim | 99.82 | 0.57 | 91.57 | 34.06 | 96.64 | 15.50 | 91.95 | 25.90 | 98.39 | 7.19 | 99.28 | 2.27 | 96.27 | 14.25 |
| + NECO | 99.71 | 1.33 | 93.01 | 31.88 | 96.96 | 15.53 | 92.41 | 26.21 | 97.35 | 17.86 | 98.78 | 6.72 | 96.37 | 16.59 |
| + SFM (ours) | 99.93 | 0.32 | 97.12 | 14.56 | 96.91 | 15.18 | 97.92 | 11.72 | 98.64 | 8.87 | 99.40 | 3.41 | 98.32 | 9.01 |

Table 18: OOD detection performance in terms of AUROC (↑) and FPR95 (↓) for different PEFT methods, and full fine-tuning on CIFAR-100-LT dataset.

| Method | Texture | | SVHN | | CIFAR10 | | Tiny ImageNet | | LSUN | | Places | | Average | |
|---|---|---|---|---|---|---|---|---|---|---|---|---|---|---|
| | AUROC | FPR95 | AUROC | FPR95 | AUROC | FPR95 | AUROC | FPR95 | AUROC | FPR95 | AUROC | FPR95 | AUROC | FPR95 |
| **Bias-tuning** | | | | | | | | | | | | | | |
| + MSP | 97.23 | 12.91 | 95.68 | 23.01 | 91.66 | 38.32 | 85.10 | 49.19 | 83.91 | 65.44 | 89.82 | 45.37 | 90.56 | 39.04 |
| + MLS | 99.89 | 0.37 | 97.73 | 10.36 | 94.29 | 28.15 | 93.67 | 24.36 | 98.87 | 5.06 | 99.29 | 2.91 | 97.29 | 11.87 |
| + Energy | 99.93 | 0.25 | 97.42 | 13.18 | 93.78 | 33.19 | 94.14 | 22.64 | 99.38 | 2.08 | 99.59 | 1.38 | 97.37 | 12.12 |
| + Mahalanobis | 99.96 | 0.20 | 99.58 | 1.27 | 94.59 | 28.56 | 97.26 | 10.39 | 99.54 | 2.14 | 99.55 | 1.84 | 98.40 | 7.40 |
| + Residual | 99.97 | 0.05 | 97.98 | 11.87 | 85.50 | 67.10 | 98.00 | 9.44 | 99.77 | 0.53 | 99.74 | 0.71 | 96.83 | 14.95 |
| + Vim | 99.95 | 0.16 | 97.74 | 10.84 | 93.87 | 32.51 | 94.67 | 20.93 | 99.50 | 1.49 | 99.66 | 0.99 | 97.57 | 11.15 |
| + NECO | 99.89 | 0.51 | 98.15 | 8.82 | 94.32 | 27.12 | 93.33 | 22.52 | 98.59 | 9.34 | 99.23 | 4.02 | 97.25 | 12.06 |
| + SFM (ours) | 100.0 | 0.00 | 99.91 | 0.08 | 93.33 | 35.86 | 99.76 | 1.18 | 100.0 | 0.00 | 100.0 | 0.00 | **98.83** | **6.19** |
| **VPT-shallow** | | | | | | | | | | | | | | |
| + MSP | 94.99 | 22.66 | 94.31 | 32.06 | 88.65 | 52.40 | 82.64 | 58.58 | 78.45 | 82.38 | 85.96 | 59.76 | 87.50 | 51.31 |
| + MLS | 99.43 | 2.85 | 96.76 | 18.52 | 87.77 | 54.41 | 81.14 | 64.50 | 93.33 | 40.85 | 95.01 | 27.59 | 92.24 | 34.79 |
| + Energy | 99.61 | 1.72 | 96.05 | 24.39 | 86.05 | 62.46 | 79.32 | 73.40 | 94.82 | 31.44 | 95.72 | 23.16 | 91.93 | 36.09 |
| + Mahalanobis | 99.92 | 0.37 | 93.09 | 38.88 | 91.12 | 42.84 | 96.48 | 14.27 | 99.64 | 1.34 | 99.57 | 1.82 | 96.64 | 16.59 |
| + Residual | 99.92 | 0.28 | 84.78 | 53.96 | 80.83 | 70.84 | 97.42 | 13.56 | 99.75 | 0.76 | 99.61 | 1.35 | 93.72 | 23.46 |
| + Vim | 99.72 | 1.40 | 96.26 | 22.38 | 86.39 | 60.61 | 81.39 | 67.38 | 95.27 | 24.05 | 96.55 | 18.72 | 92.72 | 32.42 |
| + NECO | 99.70 | 1.37 | 96.04 | 24.00 | 89.82 | 43.79 | 86.69 | 43.97 | 95.27 | 24.19 | 97.75 | 16.80 | 94.05 | 25.69 |
| + SFM (ours) | 100.0 | 0.02 | 95.65 | 25.96 | 88.82 | 50.14 | 99.72 | 1.36 | 99.98 | 0.00 | 99.97 | 0.04 | **97.36** | **12.92** |
| **VPT-deep** | | | | | | | | | | | | | | |
| + MSP | 96.78 | 14.73 | 92.13 | 38.71 | 90.87 | 42.17 | 83.57 | 53.66 | 81.08 | 72.16 | 87.71 | 52.15 | 88.69 | 45.60 |
| + MLS | 99.78 | 0.87 | 97.63 | 11.72 | 90.42 | 44.26 | 87.38 | 45.51 | 96.43 | 22.13 | 97.87 | 12.18 | 94.92 | 22.78 |
| + Energy | 99.86 | 0.55 | 97.75 | 10.55 | 88.92 | 54.12 | 87.04 | 49.12 | 97.25 | 16.66 | 98.35 | 9.25 | 94.86 | 23.37 |
| + Mahalanobis | 99.88 | 0.39 | 98.22 | 10.49 | 92.63 | 40.93 | 95.85 | 16.72 | 98.81 | 6.74 | 98.94 | 5.10 | 97.39 | 13.39 |
| + Residual | 99.90 | 0.30 | 95.61 | 23.77 | 82.05 | 73.28 | 96.25 | 18.13 | 99.23 | 3.43 | 99.11 | 3.77 | 95.36 | 20.45 |
| + Vim | 99.89 | 0.50 | 98.00 | 9.27 | 89.13 | 53.34 | 88.05 | 45.00 | 97.68 | 14.16 | 98.58 | 7.80 | 95.22 | 21.68 |
| + NECO | 99.82 | 0.78 | 97.78 | 11.74 | 91.39 | 39.08 | 89.00 | 35.93 | 96.60 | 18.40 | 98.14 | 10.10 | 95.45 | 19.34 |
| + SFM (ours) | 99.99 | 0.00 | 99.32 | 2.97 | 91.05 | 46.61 | 99.56 | 2.10 | 99.94 | 0.00 | 99.92 | 0.07 | **98.30** | **8.62** |
| **LoRA** | | | | | | | | | | | | | | |
| + MSP | 96.77 | 15.05 | 94.10 | 32.79 | 91.25 | 41.07 | 84.06 | 51.62 | 81.70 | 71.24 | 88.80 | 49.24 | 89.45 | 43.50 |
| + MLS | 99.78 | 0.85 | 96.95 | 16.78 | 93.06 | 32.74 | 87.54 | 43.19 | 97.66 | 14.45 | 98.75 | 6.61 | 95.62 | 19.10 |
| + Energy | 99.84 | 0.44 | 96.51 | 20.22 | 92.28 | 39.13 | 87.19 | 47.63 | 98.31 | 9.71 | 99.09 | 4.23 | 95.54 | 20.23 |
| + Mahalanobis | 99.97 | 0.09 | 99.59 | 1.12 | 94.16 | 30.32 | 97.26 | 10.76 | 99.47 | 2.15 | 99.59 | 1.55 | 98.34 | 7.66 |
| + Residual | 99.98 | 0.07 | 98.04 | 11.86 | 84.77 | 66.35 | 98.08 | 8.99 | 99.74 | 0.77 | 99.72 | 0.70 | 96.72 | 14.79 |
| + Vim | 99.89 | 0.39 | 97.00 | 16.42 | 92.41 | 38.74 | 88.48 | 42.91 | 98.63 | 7.75 | 99.25 | 3.28 | 95.95 | 18.25 |
| + NECO | 99.84 | 0.69 | 97.89 | 11.25 | 93.25 | 31.42 | 89.67 | 33.65 | 97.47 | 15.13 | 98.84 | 6.11 | 96.16 | 16.38 |
| + SFM (ours) | 100.0 | 0.00 | 99.96 | 0.01 | 92.62 | 38.01 | 99.87 | 0.59 | 100.0 | 0.00 | 100.0 | 0.00 | **98.74** | **6.43** |
| **Adapter** | | | | | | | | | | | | | | |
| + MSP | 97.34 | 12.54 | 95.56 | 23.93 | 91.73 | 38.80 | 85.30 | 48.04 | 84.70 | 62.47 | 90.66 | 42.81 | 90.88 | 38.10 |
| + MLS | 99.90 | 0.32 | 98.31 | 8.01 | 94.26 | 29.06 | 92.18 | 29.58 | 99.10 | 3.69 | 99.50 | 1.51 | 97.21 | 12.03 |
| + Energy | 99.93 | 0.18 | 98.28 | 7.56 | 93.67 | 34.00 | 92.43 | 28.42 | 99.48 | 1.43 | 99.71 | 0.61 | 97.25 | 12.03 |
| + Mahalanobis | 99.97 | 0.12 | 99.44 | 1.82 | 95.04 | 26.43 | 97.58 | 9.60 | 99.55 | 1.76 | 99.63 | 1.36 | 98.53 | 6.85 |
| + Residual | 99.98 | 0.02 | 97.78 | 14.80 | 86.21 | 64.68 | 98.51 | 6.91 | 99.85 | 0.47 | 99.84 | 0.36 | 97.03 | 14.54 |
| + Vim | 99.95 | 0.16 | 98.48 | 6.36 | 93.77 | 33.18 | 93.09 | 26.10 | 99.57 | 0.98 | 99.75 | 0.45 | 97.44 | 11.20 |
| + NECO | 99.90 | 0.30 | 98.60 | 6.92 | 94.41 | 27.76 | 92.31 | 26.44 | 98.91 | 6.70 | 99.45 | 2.68 | 97.26 | 11.82 |
| + SFM (ours) | 100.0 | 0.00 | 99.92 | 0.09 | 94.12 | 30.70 | 99.80 | 0.92 | 100.0 | 0.00 | 100.0 | 0.00 | **98.97** | **5.28** |
| **Full fine-tuning** | | | | | | | | | | | | | | |
| + MSP | 96.93 | 14.29 | 93.98 | 32.33 | 90.46 | 46.33 | 83.91 | 53.60 | 85.16 | 66.75 | 90.43 | 46.44 | 90.14 | 43.29 |
| + MLS | 99.86 | 0.50 | 94.01 | 33.58 | 93.96 | 28.83 | 88.43 | 37.80 | 99.16 | 1.93 | 99.41 | 1.80 | 95.81 | 17.41 |
| + Energy | 99.91 | 0.34 | 92.59 | 47.61 | 93.84 | 29.66 | 88.46 | 37.68 | 99.53 | 0.71 | 99.64 | 0.97 | 95.66 | 19.49 |
| + Mahalanobis | 99.95 | 0.27 | 97.08 | 16.06 | 95.14 | 23.82 | 97.04 | 13.62 | 99.18 | 5.22 | 99.52 | 2.62 | 97.99 | 10.27 |
| + Residual | 99.99 | 0.02 | 97.71 | 13.20 | 90.08 | 49.81 | 98.89 | 5.16 | 99.86 | 0.35 | 99.91 | 0.26 | 97.74 | 11.47 |
| + Vim | 99.94 | 0.25 | 93.63 | 40.38 | 93.98 | 29.20 | 90.18 | 32.65 | 99.65 | 0.41 | 99.73 | 0.65 | 96.18 | 17.26 |
| + NECO | 99.87 | 0.57 | 94.73 | 29.96 | 94.34 | 25.66 | 90.05 | 31.81 | 98.75 | 8.39 | 99.27 | 3.89 | 96.17 | 16.71 |
| + SFM (ours) | 99.99 | 0.02 | 97.86 | 10.72 | 95.20 | 23.66 | 98.50 | 38.37 | 99.81 | 80.77 | 99.87 | 0.58 | **98.54** | **7.35** |

Table 19: OOD detection performance in terms of AUROC (↑) and FPR95 (↓) for different PEFT methods, and full fine-tuning on ImageNet-LT dataset.

| Method | Texture | | Places | | SUN | | iNaturalist | | ImageNet-O | | Average | |
|---|---|---|---|---|---|---|---|---|---|---|---|---|
| | AUROC | FPR95 | AUROC | FPR95 | AUROC | FPR95 | AUROC | FPR95 | AUROC | FPR95 | AUROC | FPR95 |
| **Bias-tuning** | | | | | | | | | | | | |
| + MSP | 83.92 | 58.30 | 80.95 | 67.86 | 81.16 | 66.33 | 94.31 | 26.99 | 75.83 | 77.60 | 83.23 | 59.42 |
| + MLS | 88.71 | 49.75 | 84.46 | 61.47 | 85.84 | 58.57 | 96.93 | 16.31 | 82.68 | 71.40 | 87.72 | 51.50 |
| + Energy | 89.93 | 43.48 | 85.15 | 57.30 | 86.97 | 52.85 | 97.84 | 10.11 | 84.22 | 66.95 | 88.82 | 46.14 |
| + Mahalanobis | 87.55 | 59.63 | 82.72 | 61.63 | 86.38 | 51.55 | 97.89 | 10.89 | 85.03 | 62.65 | 87.92 | 49.27 |
| + Residual | 73.74 | 81.88 | 68.43 | 82.87 | 75.65 | 73.07 | 88.27 | 48.56 | 72.40 | 80.00 | 75.70 | 73.28 |
| + Vim | 90.06 | 42.68 | 85.22 | 57.05 | 87.16 | 52.08 | 97.92 | 9.83 | 84.39 | 66.60 | 88.95 | 45.65 |
| + NECO | 88.38 | 50.51 | 84.00 | 60.84 | 86.25 | 57.23 | 97.65 | 12.16 | 83.85 | 67.40 | 88.03 | 49.63 |
| + SFM (ours) | 90.95 | 41.45 | 81.67 | 63.98 | 85.48 | 54.88 | 97.75 | 11.44 | 86.58 | 56.70 | 88.48 | 45.69 |
| **VPT-shallow** | | | | | | | | | | | | |
| + MSP | 85.58 | 49.11 | 85.44 | 55.99 | 86.38 | 52.79 | 97.53 | 10.26 | 83.79 | 57.95 | 87.74 | 45.22 |
| + MLS | 89.30 | 41.47 | 88.52 | 49.37 | 90.05 | 45.47 | 98.64 | 5.90 | 88.74 | 48.50 | 91.05 | 38.14 |
| + Energy | 90.00 | 37.94 | 89.01 | 46.13 | 90.73 | 41.65 | 98.95 | 4.31 | 89.69 | 44.20 | 91.68 | 34.85 |
| + Mahalanobis | 92.07 | 29.52 | 86.20 | 58.31 | 88.98 | 49.94 | 99.19 | 3.10 | 91.51 | 38.95 | 91.59 | 35.96 |
| + Residual | 88.37 | 49.73 | 73.68 | 80.55 | 79.27 | 72.40 | 96.66 | 16.66 | 84.03 | 60.05 | 84.40 | 55.88 |
| + Vim | 90.32 | 36.33 | 89.04 | 46.03 | 90.82 | 41.24 | 99.02 | 4.11 | 89.93 | 42.75 | 91.83 | 34.09 |
| + NECO | 91.15 | 33.32 | 87.20 | 49.53 | 89.67 | 44.33 | 99.01 | 3.86 | 88.98 | 43.80 | 91.40 | 34.97 |
| + SFM (ours) | 95.93 | 14.54 | 85.98 | 57.66 | 89.03 | 47.59 | 99.18 | 3.17 | 93.34 | 32.30 | 92.69 | 31.05 |
| **VPT-deep** | | | | | | | | | | | | |
| + MSP | 85.28 | 49.27 | 84.75 | 57.12 | 85.92 | 53.82 | 97.13 | 11.51 | 83.13 | 58.20 | 87.24 | 45.98 |
| + MLS | 89.57 | 40.30 | 88.37 | 49.82 | 89.97 | 45.76 | 98.42 | 6.61 | 88.35 | 49.70 | 90.93 | 38.44 |
| + Energy | 90.32 | 37.02 | 88.92 | 46.67 | 90.65 | 42.09 | 98.72 | 5.21 | 89.23 | 45.60 | 91.57 | 35.32 |
| + Mahalanobis | 92.06 | 29.38 | 89.41 | 46.03 | 91.53 | 39.21 | 99.20 | 3.07 | 90.76 | 41.65 | 92.59 | 31.87 |
| + Residual | 89.31 | 43.60 | 82.48 | 65.82 | 86.52 | 56.29 | 97.04 | 14.73 | 82.29 | 62.05 | 87.53 | 48.50 |
| + Vim | 90.62 | 35.25 | 89.11 | 45.87 | 90.88 | 41.07 | 98.81 | 4.86 | 89.42 | 44.15 | 91.77 | 34.24 |
| + NECO | 90.47 | 35.04 | 88.46 | 46.25 | 90.73 | 39.93 | 98.82 | 4.56 | 88.81 | 45.15 | 91.46 | 34.19 |
| + SFM (ours) | 95.52 | 16.03 | 89.27 | 46.36 | 91.57 | 38.40 | 99.25 | 3.07 | 92.55 | 35.25 | 93.63 | 27.82 |
| **LoRA** | | | | | | | | | | | | |
| + MSP | 85.99 | 47.75 | 85.29 | 56.70 | 86.36 | 53.65 | 97.14 | 11.87 | 83.59 | 58.30 | 87.67 | 45.65 |
| + MLS | 90.06 | 39.08 | 88.56 | 50.13 | 90.17 | 45.98 | 98.41 | 6.79 | 88.82 | 48.35 | 91.20 | 38.07 |
| + Energy | 90.81 | 35.80 | 89.03 | 47.35 | 90.81 | 42.37 | 98.70 | 5.17 | 89.78 | 43.35 | 91.93 | 34.81 |
| + Mahalanobis | 93.12 | 25.78 | 88.31 | 50.24 | 90.92 | 41.66 | 99.28 | 2.84 | 91.57 | 39.00 | 92.64 | 31.90 |
| + Residual | 91.25 | 37.61 | 78.95 | 71.78 | 84.13 | 61.05 | 97.08 | 14.96 | 83.92 | 59.10 | 87.07 | 48.90 |
| + Vim | 91.18 | 33.92 | 89.16 | 46.63 | 91.01 | 41.06 | 98.81 | 4.81 | 90.01 | 42.30 | 92.03 | 33.74 |
| + NECO | 91.80 | 30.76 | 88.35 | 47.11 | 90.79 | 40.31 | 98.93 | 4.07 | 89.71 | 43.80 | 91.92 | 33.21 |
| + SFM (ours) | 96.85 | 11.28 | 88.36 | 49.55 | 91.06 | 40.36 | 99.26 | 2.84 | 93.58 | 30.70 | 93.82 | 26.95 |
| **Adapter** | | | | | | | | | | | | |
| + MSP | 85.48 | 49.04 | 84.97 | 56.62 | 86.28 | 53.16 | 96.97 | 12.59 | 83.56 | 57.50 | 87.45 | 45.78 |
| + MLS | 89.75 | 40.18 | 88.51 | 49.51 | 90.28 | 44.91 | 98.34 | 6.89 | 88.88 | 48.10 | 91.15 | 37.92 |
| + Energy | 90.47 | 37.02 | 89.01 | 46.88 | 90.93 | 41.73 | 98.65 | 5.59 | 89.79 | 42.90 | 91.77 | 34.82 |
| + Mahalanobis | 92.61 | 28.32 | 89.17 | 47.15 | 91.47 | 39.20 | 99.24 | 3.00 | 91.35 | 39.95 | 92.77 | 31.52 |
| + Residual | 91.32 | 37.02 | 82.47 | 65.59 | 86.63 | 55.03 | 97.42 | 12.88 | 83.67 | 60.25 | 88.30 | 46.15 |
| + Vim | 90.83 | 35.55 | 89.20 | 45.93 | 91.15 | 40.46 | 98.76 | 5.14 | 90.00 | 42.05 | 91.99 | 33.83 |
| + NECO | 91.13 | 33.10 | 88.91 | 44.55 | 91.23 | 37.50 | 98.84 | 4.33 | 89.49 | 43.10 | 91.92 | 32.52 |
| + SFM (ours) | 96.71 | 12.48 | 89.35 | 46.41 | 91.87 | 37.28 | 99.28 | 2.77 | 93.41 | 31.90 | 94.12 | 26.17 |
| **Full fine-tuning** | | | | | | | | | | | | |
| + MSP | 82.21 | 56.24 | 81.12 | 65.98 | 81.83 | 62.82 | 93.92 | 24.60 | 78.67 | 64.85 | 83.55 | 54.90 |
| + MLS | 87.34 | 48.72 | 84.31 | 57.30 | 85.87 | 57.24 | 96.04 | 18.77 | 86.30 | 56.00 | 87.97 | 48.26 |
| + Energy | 87.32 | 49.65 | 84.07 | 61.19 | 85.70 | 58.51 | 95.57 | 22.25 | 86.55 | 55.45 | 87.84 | 49.41 |
| + Mahalanobis | 89.84 | 37.94 | 85.82 | 56.61 | 87.51 | 53.62 | 98.21 | 7.27 | 87.21 | 52.45 | 89.72 | 41.28 |
| + Residual | 81.65 | 65.37 | 71.82 | 83.97 | 75.22 | 78.32 | 92.41 | 38.93 | 70.36 | 79.90 | 78.29 | 69.30 |
| + Vim | 87.56 | 48.42 | 84.12 | 60.91 | 85.80 | 58.11 | 95.81 | 20.61 | 86.53 | 55.70 | 87.97 | 48.75 |
| + NECO | 87.56 | 43.74 | 83.30 | 57.95 | 85.99 | 53.72 | 97.11 | 11.76 | 85.76 | 52.20 | 87.95 | 43.87 |
| + SFM (ours) | 93.28 | 25.05 | 86.17 | 55.15 | 87.90 | 51.41 | 98.54 | 5.99 | 88.87 | 46.35 | 90.95 | 36.79 |