# OpenReview forum: "Harnessing Shallow Features in Pre-Trained Models for Out-of-Distribution Detection"
_ICLR.cc/2025/Conference — Submitted to ICLR 2025_

### Official Review · Reviewer_cJWd · 2024-10-30

**Soundness:** 2
**Presentation:** 3
**Contribution:** 2
**Rating:** 6
**Confidence:** 4

**Summary:**

With the favor of transformer architecture,  this work proposes an out-of-distribution (OOD) score based on the fused feature entailed from each layer. Further, it is empirically observed that parameter-efficient fine-tuning (PEFT) is more robust than fully fine-tuning when using different learning rates. Finally, it demonstrates the effectiveness of the proposed OOD score along with PEFT on the OOD benchmark and also long-tailed OOD benchmark.

**Strengths:**

- The proposed OOD score based on the fused feature from each layer is novel.
- This work aims to tackle the OOD detection on both class-balanced benchmark and long-tailed benchmark.

**Weaknesses:**

-  The title appears somewhat overstated, as the proposed method specifically requires transformer architectures. A more precise title, such as "... in pre-trained transformers for ...", would better reflect the scope of the method.
- Line 103-104: The mention of "measuring the degree of neural collapse of each layer" is unclear. Could the authors elaborate on this point and clarify how it relates to determining the importance weights for each layer.
- The proposed OOD score aligns with post-hoc OOD scores that require access to training data. To assess its effectiveness, the authors should consider evaluating it on a standard OOD benchmark, as outlined in [2].  Comparisons with other training-data-dependent OOD scores—such as KL matching, Residual, and ViM—would provide a more comprehensive evaluation (see Table 1 in [1] for reference).
- It is unclear why the authors chose to fine-tune a network pre-trained on ImageNet-21k. Why not instead use models pre-trained directly on ImageNet-1k or CIFAR-100 and then perform OOD detection on a standard benchmark[2]?
- Dependency on Fine-tuning: Could the authors discuss whether the proposed OOD score's effectiveness relies on the pre-trained models with extensive datasets? For instance, the selected models in this work is pre-trained ViT models on ImageNet-21k and  CLIP-ViT-B/16. Have you tested or considered testing with smaller pre-trained models or datasets to assess its effectiveness across different scales of pre-training.
- Comparison with MCM (Line 394): The comparison with MCM, in which the authors claim that SFM outperforms MCM by ~2%, may not be fair. The original MCM model was not fine-tuned on ImageNet-1k, which could impact the validity of this comparison.











Reference
- [1] GEN: Pushing the Limits of Softmax-Based Out-of-Distribution Detection. In CVPR, 2023.
- [2] OpenOOD v1.5: Enhanced Benchmark for Out-of-Distribution Detection.  In  NeurIPS 2023 Workshop on Distribution Shifts.

**Questions:**

See weakness.

---

> ### Author Response · Authors · 2024-11-16
>
> Dear Reviewer cJWd,
> We are extremely grateful for the comprehensive and detailed review you have provided. Your insights and suggestions are invaluable in helping us improve our work.
>
> **[W1] The title appears overstated.**
>
> **[A1]** We are truly and sincerely grateful for your astute and perceptive advice regarding the title of our work. After meticulous and in-depth consideration, we have come to recognize the validity of your observation. Indeed, a more precise and accurate title is essential to aptly reflect the core scope and nature of our proposed method. We wholeheartedly concur with your suggestion and will accordingly modify the title to "Harnessing Shallow Features in Pre-Trained Transfomers for Out-of-Distribution Detection". This revised title, we believe, will not only enhance the clarity and specificity of our research's representation but also align more closely with the technical details and contributions of our work.
> Once again, we express our profound appreciation for your invaluable feedback, which has significantly contributed to the refinement and improvement of our manuscript.
>
> **[W2] An explanation for "measuring the degree of neural collapse of each layer".**
>
> **[A2]** We are extremely grateful for your incisive query regarding the mention of "measuring the degree of neural collapse of each layer" and its relation to determining the importance weights.
> Our fundamental motivation was grounded in the convergence behavior of the last layer.  In machine learning, the typical aspiration is for the class center, classifier, and samples to converge.  However, this poses a detriment to OOD detection as **OOD samples may also converge to the same class-center**, and this is equally true for OOD detection scores based on Mahalanobis distance.  To address this, we aimed to introduce less convergent information into the sample features, thereby enabling them to escape the neural collapse condition, while still retaining a certain degree of convergent information.  Through comprehensive statistical analysis and comparisons, we devised a method to dissolve the weights of different layers.  We posit that the weights we currently employ can, to a certain extent, represent the convergence degree of each layer.  **For a more detailed understanding, please refer to the definition of alpha in Equation 5.** To illustrate, if all samples within the same layer achieve optimal convergence, known as neural collapse, the variance of feature samples in that layer should be zero, meaning that the covariance matrix is zero except along the diagonal. Hence, moving towards neural collapse requires minimizing the off-diagonal elements while maximizing the diagonal elements, or the trace. Consequently, we assert that the definition of alpha provided by Eq5 effectively quantifies the extent of neurocollapse.
> We also acknowledge that the expression in lines 103 - 104 was suboptimal and lacked clarity.  We will undertake a thorough revision of this description in the subsequent iteration to enhance the overall readability and comprehensibility of the paper.

---

> ### Author Response · Authors · 2024-11-16
>
> **[W3] To assess effectiveness.**
>
> **[A3]** We greatly appreciate your valuable feedback. We have seriously considered how to more suitably assess the proposed OOD score.
> We've carefully analyzed the dataset in OpenOOD v1.5. In terms of using CIFAR-10 as the in-distribution (ID) dataset, we found it to be relatively straightforward. As you might be aware, most methods we investigated achieve nearly perfect results, with the AUROC almost reaching 100 and the FPR95 being 0.00 when utilizing the ViT model.
> Currently, our CIFAR100 test set diverges only from Mnist. Therefore, for the sake of simplicity and also due to the time constraints during the rebuttal phase, we've opted to repeat the experiment with Imagenet-1k as the ID dataset in the OpenOOD v1.5 benchmark. Furthermore, we chose Imagenet-1K-LT as an ID as well since earlier experiments showed it is quite similar to Imagenet-1k but saves time.
> The outcomes of these experiments are shown in the following table. We sincerely hope that this strategy provides a more meaningful and challenging benchmark to evaluate our proposed OOD score while acknowledging practical constraints such as limited time.
>
> **Results for ImageNet-1k-LT in-distribution dataset on OpenOOD v1.5**
> | Method | Ninco (AUROC) | Ninco (FPR95) | Openimage-o (AUROC) | Openimage-o (FPR95) | Ssb-hard (AUROC) | Ssb-hard (FPR95) | Imagenet_c (AUROC) | Imagenet_c (FPR95) | Imagenet_es (AUROC) | Imagenet_es (FPR95) | Imagenet_r (AUROC) | Imagenet_r (FPR95) | Imagenet_v2 (AUROC) | Imagenet_v2 (FPR95) | Average (AUROC) | Average (FPR95) |
> | ----| ---- | ---- | ---- | ---- | ---- | ---- | ---- |---- | ---- | ---- | ---- | ---- | ---- | ---- | ---- | ---- |
> | MSP         | 87.81  | 50.02 |  93.72 | 27.51  |  76.72   | 68.43  |  67.91   | 78.58 |  69.35  | 69.26 |  79.73   | 59.15 | 57.57  | 89.92 |  76.12   | 63.27 |
> | MLS         | 91.59  | 42.80 |  96.28 | 18.65  |  81.25   | 63.86  |  70.54   | 76.77 |  72.11  | 66.79 |  83.65   | 53.22 | 57.86  | **90.09** |  79.04   | 58.88 |
> | Energy      | 92.12  | 39.62 |  96.80 | 16.02  |  81.87   | 61.53  |  70.80   | 76.02 |  72.44  | 66.06 |  84.25   | 50.52 | 57.79  | 90.19 |  79.44   | 57.14 |
> | Mahalanobis | 94.00  | 32.51 |  97.58 | 12.61  |  85.01   | 52.17  |  73.93   | 72.64 |  73.04  | 67.08 |  85.32   | 48.95 | 58.02  | 90.81 |  80.99   | 53.83 |
> | residual    | 83.87  | 62.45 |  92.41 | 33.88  |  84.87   | 56.19  |  74.96   | 78.03 |  65.25  | 82.87 |  75.05   | 76.46 | 53.03  | 94.38 |  75.63   | 69.18 |
> | Vim         | 92.29  | 38.65 |  96.94 | 15.32  |  82.36   | 60.40  |  71.19   | 75.39 |  72.47  | 65.98 |  84.37   | 50.20 | 57.79  | 90.15 |  79.63   | 56.58 |
> | NECO        | 91.97  | 38.09 |  96.90 | 15.19  |  84.81   | 54.96  |  70.55   | 75.48 |  72.01  | 67.61 |  82.43   | 53.86 | 56.86  | 90.44 |  79.36   | 56.52 |
> | KL-matching | 90.53  | 41.63 |  95.95 | 18.15  |  79.52   | 63.02  |  70.03   | 75.91 |  71.54  | 66.35 | 82.56   | 52.60 | 58.33  | 89.85 |  78.35   | 58.22 |
> | SFM (ours)  | **94.98**  | **26.74** |  **98.21** | **9.72**  |  **86.34**   | **49.46**  |  **83.96**   | **53.57** |  **76.78**  | **63.45** |  **88.49**   | **42.85** | **58.36**  | 91.36 |  **83.88**  | **48.16** |
> | ----| ---- | ---- | ---- | ---- | ---- | ---- | ---- |---- | ---- | ---- | ---- | ---- | ---- | ---- | ---- | ---- |
> | MSP         | 80.11  | 68.94 |  88.22 | 46.72 |  68.06   | 83.66  |  73.44   | 70.44 |  70.31  | 68.46 |  77.27   | 64.00 | 57.12  | 90.78 |  73.50   | 70.43 |
> | MLS         | **84.17**  | 67.11 |  92.93 | 35.17 |  71.99   | 82.44  |  77.66   | 67.65 |  75.66  | 64.21 |  84.61   | 55.96 | **58.24**  | 90.38 |  77.89   | 66.13 |
> | Energy      | 84.15  | 67.78 |  **93.73** | **30.59** |  72.25   | 82.43  |  78.02   | 67.13 |  76.51  | 62.51 |  85.87   | 52.26 | 58.22  | 90.28 |  78.39   | 64.71 |
> | Mahalanobis | 75.13  | 83.28 |  86.95 | 63.82 |  66.11   | 89.49  |  82.68   | 62.64 |  84.27  | 52.46 |  **90.02**   | 47.33 | 58.18  | 90.31 |  77.62   | 69.90 |
> | residual    | 61.56  | 91.54 |  70.43 | 81.35 |  61.00   | 92.63  |  82.86   | 68.28 |  86.30  | 54.65 |  86.28   | 57.41 | 56.47  | 92.06 |  72.13   | 76.85 |
> | Vim         | 83.91  | 68.20 |  93.57 | 31.15 |  **72.28**   | 82.57  |  78.85   | 65.06 |  77.52  | 60.34 |  86.73   | **49.79** | 58.35  | 89.88 |  78.74   | 63.86 |
> | NECO        | 80.96  | 71.90 |  92.48 | 37.13 |  69.22   | 85.10  |  77.84   | 68.17 |  78.69  | 61.76 |  86.00   | 54.92 | 58.18  | 89.83 |  77.63   | 66.97 |
> | KL-matching | 83.21  | **66.88** |  92.13 | 34.08 |  70.63   | 81.75  |  76.09   | 67.07 | 72.92   | 64.25 |  81.70   | 55.46 | 57.83  | 90.51 |  76.36   | 65.71 |
> | SFM (ours)  | 78.30  | 74.67 |  92.90 | 36.93 |  71.64   | **81.79**  |  **85.49**   | **50.94** |  **87.84**  | **43.49** |  88.76   | 50.31 | 58.22  | **89.50** |  **80.45**   | **61.09** |

---

> > ### Comment · Reviewer_cJWd · 2024-11-25
> > **the effectiveness of the proposed post-hoc score**
> >
> > Thank you for conducting the comparative experiments. If I understand correctly, the results presented in the first half of the block use ImageNet-1k as the ID dataset, while the second half of the block uses ImageNet-1k-LT as the ID dataset. Additionally, all post-hoc scores are calculated using the fine-tuned checkpoints obtained with the proposed loss. This indicates that the effectiveness of the proposed post-hoc method depends on the additional fine-tuning step.
> >
> > * I am curious, if you directly use the pre-trained checkpoints with ImageNet-1k as the ID dataset, what would the performance look like?
> > * If the proposed post-hoc score depends on fine-tuning, it would be beneficial to include a comparison of classification accuracy before and after fine-tuning. This would help demonstrate that the improvement in OOD detection performance does not come at the expense of classification accuracy.

---

> > > ### Author Response · Authors · 2024-11-26
> > > **Further response by the Authors**
> > >
> > > Dear Reviewer cJWd,
> > >
> > > Thank you for your thoughtful comments and for your accurate understanding of our proposed method. Below, we provide detailed responses to address your additional concerns.
> > >
> > > **[Q1] Performance for directly use the pre-trained checkpoints with ImageNet-1k as the ID dataset.**
> > >
> > > **[A1]** Thank you for your suggestion! We have included the comparison results for applying our proposed method to both the pre-trained model and the fine-tuned model in the tables below. As expected, **the fine-tuned model demonstrates superior performance** due to its ability to learn more discriminative features specific to the ID datasets. Interestingly, **even when applied directly to the pre-trained model, our method achieves competitive performance**, highlighting its effectiveness.
> > >
> > >
> > >
> > > *Table 1: Results for using ImageNet-1k as ID dataset*
> > > |  | Texture (AUROC) | Texture (FPR95) | Places (AUROC) | Places (FPR95) | SUN (AUROC) | SUN (FPR95) | iNaturalist (AUROC) | iNaturalist (FPR95) | ImageNet-O (AUROC) | ImageNet-O (FPR95) | Average (AUROC) | Average (FPR95) |
> > > | ----| ---- | ---- | ---- | ---- | ---- | ---- | ---- |---- | ---- | ---- | ---- | ---- |
> > > | Pre-trained model  | **97.19** | **9.88** | 82.72 | 65.14 | 86.41 | 54.64 | 98.80 | 4.75 | 93.40 | 31.45 | 91.70 | 33.17|
> > > | Fine-tuned model  | 96.65 | 11.70 | **89.64** | **46.00** | **92.04** | **37.78** | **99.40** | **2.26** | **93.76** | **29.80** | **94.30** | **25.51** |
> > >
> > >
> > > ---
> > > ---
> > >
> > > *Table 2: Results for using ImageNet-1k-LT as ID dataset*
> > > |  | Texture (AUROC) | Texture (FPR95) | Places (AUROC) | Places (FPR95) | SUN (AUROC) | SUN (FPR95) | iNaturalist (AUROC) | iNaturalist (FPR95) | ImageNet-O (AUROC) | ImageNet-O (FPR95) | Average (AUROC) | Average (FPR95) |
> > > | ----| ---- | ---- | ---- | ---- | ---- | ---- | ---- |---- | ---- | ---- | ---- | ---- |
> > > | Pre-trained model  | **97.15** | **10.20** | 81.51 | 67.20 | 85.49 | 56.49 | 98.69 | 5.13 | 93.15 | 32.15 | 91.20 | 34.23 |
> > > | Fine-tuned model   | 96.92 | 11.79 | **89.82** | **45.36** | **92.18** | **36.16** | **99.33** | **2.51** | **93.46** | **31.10** | **94.34** | **25.38** |
> > >
> > >
> > > **[Q2] Comparison of classification accuracy before and after fine-tuning.**
> > >
> > > **[A2]** Thanks for your suggestion again. We present the accuracy of the pre-trained model and after the fine-tuning in the table below. `More detailed performance comparisons are reported in Table 7 (Appendix A) of our paper`.
> > > The results demonstrate that fine-tuning is essential for boosting ID classification accuracy. **This justify that the improvement in OOD detection performance does not come at the expense of classification accuracy.**
> > >
> > > *Table 3: Comparison of classification accuracy for the pre-trained model and the fine-tuned model.*
> > > |  | Pre-trained model | Fine-tuned model |
> > > | ----| ---- | ---- |
> > > | ImageNet-1k      | 80.03 | 83.50 |
> > > | ImageNet-1k-LT      | 79.25 |  81.79 |
> > > | CIFAR-100 | 77.62 |  93.47 |
> > > | CIFAR-100-LT | 75.54 |  89.99 |
> > >
> > > Thank you for your valuable comments! We will include the experimental analysis in the paper and further discuss the implications.

---

> > > > ### Comment · Reviewer_cJWd · 2024-11-26
> > > >
> > > > Thanks for conducting the suggested experiments!
> > > >
> > > > * It would be beneficial to include the results for other post-hoc methods such as ViM when utilizing the pre-trained checkpoints as the backbone.
> > > > * I double-checked Table 7 in Appendix A and confirmed that the numbers presented represent the accuracy for the original classification tasks. However, it appears that the accuracy shown in Table 7 reflects the results after fine-tuning, while the accuracy before fine-tuning is missing. Could you please clarify this?

---

> ### Author Response · Authors · 2024-11-16
> **The second part of reply to [W3]**
>
> We are highly appreciative of your valuable suggestion regarding the comparison with other training-data-dependent OOD scores. We have indeed made efforts in this regard. As you can see, we have presented the detailed results of Residual and ViM within our experimental results section. Additionally, to further address your concern and provide a more comprehensive evaluation, we have also includ the KL-matching results in the table below for direct comparison. This will enable a more in-depth analysis and a better understanding of the performance and characteristics of our proposed method in relation to these benchmarks. We hope that this additional information will enhance the overall quality and persuasiveness of our research.
> The results of ImageNet-21k pre-trained ViT are displayed in the top half of each table, while the results on CLIP are displayed in the bottom half.
>
> **Results for CIFAR-100 in-distribution dataset**
> | Method | Texture (AUROC) | Texture (FPR95) | SVHN (AUROC) | SVHN (FPR95) | CIFAR10 (AUROC) | CIFAR10 (FPR95) | Tiny ImageNet (AUROC) | Tiny ImageNet (FPR95) | LSUN (AUROC) | LSUN (FPR95) | Places365 (AUROC) | Places365 (FPR95) | Average (AUROC) | Average (FPR95) |
> | ---- | ---- | ---- | ---- | ---- | ---- | ---- | ---- |---- | ---- | ---- | ---- | ---- | ---- | ---- |
> | KL-matching    | 98.60  | 6.10  |  96.66   | 14.93 |  96.34  | **17.12**  |  90.05  | 34.17  |  88.15  | 49.34  |  93.67   | 28.21  |  93.91   | 24.98  |
> | SFM (ours)  |  **100.0** | **0.00** | **99.50**    | **1.91** | **96.47**  | 19.52 | **99.80** | **1.11** | **100.0** | **0.00** | **100.0** | **0.00** | **99.29** | **3.76** |
> | ---- | ---- | ---- | ---- | ---- | ---- | ---- | ---- |---- | ---- | ---- | ---- | ---- | ---- | ---- |
> | KL-matching    | 94.32   | 25.12  |  90.69   | 38.25 |  90.69   | 38.52 |  84.16   | 52.94  |  77.85   | 70.96 |  86.99  | 47.80  |  87.45   | 45.60  |
> | SFM (ours)  |  **99.95** | **0.02** | **98.31**  | **8.62** | **88.56**  | **53.97** | **97.54** | **12.91** | **99.93** | **0.06** | **99.95** | **0.02** | **97.37** | **12.60** |
>
> **Results for CIFAR-100-LT in-distribution dataset**
> | Method | Texture (AUROC) | Texture (FPR95) | SVHN (AUROC) | SVHN (FPR95) | CIFAR10 (AUROC) | CIFAR10 (FPR95) | Tiny ImageNet (AUROC) | Tiny ImageNet (FPR95) | LSUN (AUROC) | LSUN (FPR95) | Places365 (AUROC) | Places365 (FPR95) | Average (AUROC) | Average (FPR95) |
> | ---- | ---- | ---- | ---- | ---- | ---- | ---- | ---- |---- | ---- | ---- | ---- | ---- | ---- | ---- |
> | KL-matching   | 98.48   | 6.40  |  97.44   | 12.11  |  94.00   | **26.88**  |  87.56  | 38.91  |  86.65   | 52.78  |  92.94   | 31.01  |  92.84  | 28.02  |
> | SFM (ours)  |  **100.0** | **0.00** | **99.75**  | **0.43** | **94.22**  | 29.86 | **99.75** | **1.12** | **99.99** | **0.00** | **99.99** | **0.01** | **98.95** | **5.24** |
> | ---- | ---- | ---- | ---- | ---- | ---- | ---- | ---- |---- | ---- | ---- | ---- | ---- | ---- | ---- |
> | KL-matching   | 95.01   | 21.76  |  90.76   | 31.69  |  **88.87**    | **44.17**   |  81.68   | 57.93  |  79.31   | 63.65  |  87.64   | 43.21  |  87.21  | 43.73  |
> | SFM (ours)  |  **99.92** | **0.02** | **97.50**  | **15.65** | 85.20 | 60.41| **95.05** | **26.05** | **99.92** | **0.04** | **99.93** | **0.01** | **96.25** | **17.03** |

---

> > ### Author Response · Authors · 2024-11-18
> > **The third part of reply to [W3]**
> >
> > **Results for ImageNet-1k in-distribution dataset**
> > | Method | Texture (AUROC) | Texture (FPR95) | Places (AUROC) | Places (FPR95) | SUN (AUROC) | SUN (FPR95) | iNaturalist (AUROC) | iNaturalist (FPR95) | ImageNet-O (AUROC) | ImageNet-O (FPR95) | Average (AUROC) | Average (FPR95) |
> > | ----| ---- | ---- | ---- | ---- | ---- | ---- | ---- |---- | ---- | ---- | ---- | ---- |
> > | KL-matching   | 87.85  | 40.92  |  86.76  | 53.02 |  87.89   | 49.19  |  97.84   | 8.84 |  86.25   | 49.20 |  89.32   | 40.23 |
> > | SFM (ours) |  **96.65** | **11.70** | **89.64** | **46.00** | **92.04** | **37.78** | **99.40** | **2.26** | **93.76** | **29.80** | **94.30** | **25.51** |
> > | ---- | ---- | ---- | ---- | ---- | ---- | ---- | ---- |---- | ---- | ---- | ---- | ---- |
> > | KL-matching   | 86.64  | **46.45** |  83.28  | 59.25 |  83.21   | 61.23  |  94.18   | 24.99 |  **83.19**   | **62.45** |  86.10   | 50.87 |
> > | SFM (ours) |  **89.16** | 48.44 | **91.88** | **36.46** | **93.24** | **34.87** | **95.47** | **23.49** | 82.27 | 70.35 | **90.41** | **42.72** |
> >
> > **Results for ImageNet-1k-LT in-distribution dataset**
> > | Method | Texture (AUROC) | Texture (FPR95) | Places (AUROC) | Places (FPR95) | SUN (AUROC) | SUN (FPR95) | iNaturalist (AUROC) | iNaturalist (FPR95) | ImageNet-O (AUROC) | ImageNet-O (FPR95) | Average (AUROC) | Average (FPR95) |
> > | ---- | ---- | ---- | ---- | ---- | ---- | ---- | ---- |---- | ---- | ---- | ---- | ---- |
> > | KL-matching   | 88.72  | 38.71  |  87.41  | 50.03 |  89.14   | 45.83  |  98.44   | 6.19 |  87.24   | 47.50 |  90.19   | 37.65 |
> > | SFM (ours)  |  **96.92** | **11.79** | **89.82** | **45.36** | **92.18** | **36.16** | **99.33** | **2.51** | **93.46** | **31.10** | **94.34** | **25.38** |
> > | ---- | ---- | ---- | ---- | ---- | ---- | ---- | ---- |---- | ---- | ---- | ---- | ---- |
> > | KL-matching    | 85.35  | 51.56  |  82.84  | 57.00  |  82.51   | 57.56  |  94.54   | 23.36 |  82.52  | 64.00 |  85.55   | 50.70 |
> > | SFM (ours)  |  **90.95** | **43.10** | **92.43** | **34.80** | **92.62** | **39.07** | **94.59** | **28.54** | **83.62** | **68.90** | **90.84** | **42.88** |

---

> ### Author Response · Authors · 2024-11-16
>
> **[W4] Why chose to fine-tune a network pre-trained on ImageNet-21k?**
>
> **[A4]** The main reason is that our downstream tasks are ImageNet-1k and CIFAR-100. We aim to **avoid overfitting of the model to the downstream datasets**, as an overly fitted model would render the tasks overly simplistic and might not provide a comprehensive and accurate assessment of the method's performance and generalization ability. To ensure a fair comparison, we have chosen to follow the practices of existing works such as NeCo and ViM. Moreover, to further guarantee that the pre-training dataset and the downstream task datasets do not have excessive overlap, we have also conducted experiments using CLIP. This multi-faceted approach allows us to better evaluate the effectiveness and robustness of our proposed method across different datasets and pre-training setups, and we believe it contributes to a more reliable and meaningful research outcome.

---

> ### Author Response · Authors · 2024-11-16
>
> **[W5] Whether effectiveness relies on the pre-trained models and test with smaller pre-trained models.**
>
> **[A5]** Thank you so much for raising this important point regarding whether the effectiveness of our method relies on the pre-trained models and suggesting to test with smaller pre-trained models. Firstly, we wish to convey that our Out-of-Distribution (OOD) score's effectiveness is not contingent upon pre-trained models. We employ pre-trained models primarily due to their practical benefits. It's significantly simpler and requires fewer resources to fine-tune existing pre-trained models rather than developing a new model from scratch. Moreover, many contemporary state-of-the-art methods also rely on fine-tuning pre-trained models. This prevalent approach not only streamlines the experimental process but also facilitates more straightforward and fair comparisons across different approaches, which is essential for an objective assessment of our method's performance and contributions.
> Additionally, in light of your insightful suggestion, we have indeed verified the validity of our OOD score with smaller pre-trained models. As depicted in the table below, we have included models like vit_tiny_patch16_224 and vit_small_patch16_224, shown in the upper and lower sections of each table. The outcomes from these smaller models provide further confirmation that our OOD score remains robust and effective across various model scales, thereby enhancing the generalizability and reliability of our proposed approach.
>
>
> **Results for CIFAR-100 in-distribution dataset**
> | Method | Texture (AUROC) | Texture (FPR95) | SVHN (AUROC) | SVHN (FPR95) | CIFAR10 (AUROC) | CIFAR10 (FPR95) | Tiny ImageNet (AUROC) | Tiny ImageNet (FPR95) | LSUN (AUROC) | LSUN (FPR95) | Places365 (AUROC) | Places365 (FPR95) | Average (AUROC) | Average (FPR95) |
> | ---- | ---- | ---- | ---- | ---- | ---- | ---- | ---- |---- | ---- | ---- | ---- | ---- | ---- | ---- |
> | MSP         | 92.09   | 35.34  |  83.28   | 61.28 |  83.30   | 63.73 |  79.89  | 69.73  |  72.86   | 84.07 |  82.13  | 65.55  |  82.26   | 63.28  |
> | MLS         | 98.62   | 6.13   |  92.09   | 35.85 |  87.39   | 54.92 |  87.71  | 52.79  |  88.92   | 57.57 |  94.27  | 30.32  |  91.50   | 39.60  |
> | Energy      | 99.03   | 4.26   |  92.78   | 32.41 |  87.28   | 55.96 |  88.18  | 51.27  |  90.32   | 51.04 |  95.25  | 24.94  |  92.14   | 36.65  |
> | Mahalanobis | 99.90   | 0.35   |  96.28   | 15.78 |  87.78   | 56.67 |  92.48  | 33.81  |  98.20   | 9.10  |  98.77  | 6.27   |  95.57   | 20.33  |
> | residual    | 99.71   | 0.85   |  86.24   | 52.70 |  76.86   | 72.72 |  90.89  | 42.55  |  97.46   | 14.02 |  97.25  | 13.66  |  91.40   | 32.75  |
> | Vim         | 99.19   | 3.62   |  92.99   | 31.18 |  87.49   | 54.90 |  88.70  | 48.96  |  91.14   | 47.63 |  95.67  | 22.63  |  92.53   | 34.82  |
> | NECO        | 99.17   | 3.83   |  92.34   | 34.24 |  **87.85**    | **53.47** |  89.47  | 46.45  |  92.38  | 43.05 |  96.06  | 21.11  |  92.88   | 33.69  |
> | KL-matching | 95.41   | 18.40  |  87.58   | 45.71 |  86.32    | 53.75 |  82.47  | 60.45 |  75.28   | 79.24 |  85.65  | 52.82 |  85.45   | 51.73  |
> | SFM (ours)  | **100.0**  | **0.02**   |  **96.85**   | **14.13** |  86.48    | 60.56 |  **97.00**  | **15.34**  |  **99.98**   | **0.01** |  **99.96**  | **0.10**  |  **96.71**   | **15.03**  |
> | ---- | ---- | ---- | ---- | ---- | ---- | ---- | ---- |---- | ---- | ---- | ---- | ---- | ---- | ---- |
> | MSP         | 95.98   | 19.17  |  92.29   | 38.18 |  90.82   | 39.01 |  85.95   | 52.36  |  82.84   | 68.92 |  89.31  | 47.87  |  89.53   | 44.25  |
> | MLS         | 99.28   | 3.16   |  96.35   | 18.16 |  95.22   | 24.90 |  92.18   | 32.72  |  96.21   | 25.40 |  97.71  | 13.44  |  96.16   | 19.63  |
> | Energy      | 99.48   | 2.29   |  96.54   | 16.55 |  95.42   | 23.24 |  92.59   | 29.99  |  97.12   | 18.57 |  98.25  | 10.09  |  96.57   | 16.79  |
> | Mahalanobis | 99.91   | 0.59   |  99.05   | 4.72  |  94.65   | 28.65 |  97.53   | 11.36  |  99.60   | 1.78  |  99.54  | 2.52   |  98.38   | 8.27   |
> | residual    | 99.96   | 0.11   |  98.60   | 7.06  |  88.66   | 52.27 |  98.09   | 9.68   |  99.65   | 0.75  |  99.67  | 1.14   |  97.44   | 11.83  |
> | Vim         | 99.56   | 1.99   |  96.88   | 14.63 |  **95.46**   | **23.06** |  93.17   | 27.68  |  97.52   | 16.28 |  98.47  | 8.82   |  96.84   | 15.41  |
> | NECO        | 99.50   | 2.16   |  96.76   | 15.91 |  95.33   | 24.33 |  93.49   | 26.62  |  97.26   | 17.04 |  98.29  | 9.75   |  96.77   | 15.97  |
> | KL-matching | 97.66   | 9.24   |  94.75   | 22.21 |  93.18   | 28.02 |  88.04   | 40.26  |  85.27   | 55.47 |  91.73  | 33.41  |  91.77   | 31.43  |
> | SFM (ours)  | **100.0**   | **0.00**   |  **99.36**   | **3.16** |  94.09    | 31.35 |  **99.47**   | **2.69**  |  **99.99**   | **0.01** |  **99.99**  | **0.01**  |  **98.82**   | **6.20**  |

---

> ### Author Response · Authors · 2024-11-16
> **The second part of reply to [W5]**
>
> **Results for CIFAR-100-LT in-distribution dataset**
> | Method | Texture (AUROC) | Texture (FPR95) | SVHN (AUROC) | SVHN (FPR95) | CIFAR10 (AUROC) | CIFAR10 (FPR95) | Tiny ImageNet (AUROC) | Tiny ImageNet (FPR95) | LSUN (AUROC) | LSUN (FPR95) | Places365 (AUROC) | Places365 (FPR95) | Average (AUROC) | Average (FPR95) |
> | ---- | ---- | ---- | ---- | ---- | ---- | ---- | ---- |---- | ---- | ---- | ---- | ---- | ---- | ---- |
> | MSP         | 90.08 | 43.37 |  81.70   | 67.65 |  79.48  | 71.62  |  75.96  | 75.24  |  71.36 | 84.26  |  79.41  | 71.40  |  79.66   | 68.92  |
> | MLS         | 99.12 | 3.60  |  93.33   | 33.93 |  79.87  | 74.86  |  85.81  | 56.68  |  93.82 | 33.03  |  96.58  | 18.97  |  91.42   | 36.85  |
> | Energy      | 99.38 | 2.13  |  93.83   | 31.25 |  78.33  | 78.86  |  86.24  | 56.10  |  95.32 | 24.89  |  97.50  | 13.26  |  91.77   | 34.42  |
> | Mahalanobis | 99.85 | 0.53  |  97.29   | 12.80 |  **85.08**  | **63.78**  |  91.26  | 35.45  |  98.44 | 8.10  |  98.67 | 6.67  |  95.10   | 21.22  |
> | residual    | 99.36 | 2.70  |  85.09   | 63.99 |  63.92  | 86.08  |  86.92  | 56.50  |  95.55 | 23.38  |  95.84  | 23.68  |  88.68   | 42.72  |
> | Vim         | 99.48 | 1.86  |  94.02   | 30.06 |  78.59  | 78.29  |  86.75  | 53.95  |  95.67 | 22.96  |  97.70  | 12.17 |  92.03   | 33.21  |
> | NECO        | 99.43 | 2.16  |  93.71   | 31.78 |  80.42  | 73.09  |  87.26  | 50.52  |  95.19 | 24.22  |  97.42  | 14.01  |  92.24   | 32.63  |
> | KL-matching | 94.50 | 23.48 |  86.54   | 55.21 |  82.74  | 62.81  |  79.16  | 66.66  |  74.56 | 80.30  |  83.53  | 60.59  |  83.51   | 58.18  |
> | SFM (ours)  | **99.99** | **0.04**  |  **97.77**   | **10.81** |  83.67  | 66.99  |  **96.31**  | **16.67**  |  **99.97** | **0.01**  |  **99.95**  | **0.14**  |  **96.28**   | **15.78**  |
> | ---- | ---- | ---- | ---- | ---- | ---- | ---- | ---- |---- | ---- | ---- | ---- | ---- | ---- | ---- |
> | MSP         | 96.39  | 16.72  |  92.72   | 37.39  |  87.58  | 49.60  |  82.39  | 57.10  |  80.54  | 68.09  |  87.52   | 49.64  |  87.85 | 46.42  |
> | MLS         | 99.69  | 1.44   |  95.97   | 21.84  |  91.96  | 41.67  |  92.62  | 29.45  |  97.66  | 14.61  |  98.77   | 6.74   |  96.11 | 19.29  |
> | Energy      | 99.80  | 1.13   |  95.29   | 27.28  |  91.55  | 45.20  |  93.37  | 25.75  |  98.58  | 8.86   |  99.28   | 3.78   |  96.31 | 18.67  |
> | Mahalanobis | 99.91  | 0.53   |  99.43   | 2.35   |  **93.02**  | **35.98**  |  97.15  | 12.93  |  99.59  | 2.35   |  99.65   | 1.67   |  98.12 | 9.30   |
> | residual    | 99.93  | 0.25   |  96.22   | 24.49  |  83.28  | 64.13  |  95.96  | 21.20  |  99.26  | 3.78   |  99.37   | 2.78   |  95.67 | 19.44  |
> | Vim         | 99.84  | 0.96   |  95.74   | 25.12  |  91.69  | 44.68  |  93.78  | 24.60  |  98.77  | 7.65  |   99.38   | 3.29   |  96.53 | 17.72  |
> | NECO        | 99.77  | 1.13   |  96.30   | 20.92  |  91.64  | 41.49  |  92.86  | 27.03  |  97.67  | 14.01  |  98.92   | 5.90   |  96.19 | 18.41  |
> | KL-matching | 98.12  | 7.73   |  95.63   | 20.27  |  90.16  | 37.62  |  85.25  | 46.40  |  83.94  | 56.00  |  90.77   | 36.06  |  90.65 | 34.01  |
> | SFM (ours)  | **100.0**   | **0.00** |  **99.68**   | **1.05**  |  92.24  | 39.11  |  **99.48**  | **2.44** |  **100.0**  | **0.00**  |  **100.0**   | **0.00**  |  **98.57** | **7.10**  |

---

> ### Author Response · Authors · 2024-11-16
> **The third part of reply to [W5]**
>
> **Results for ImageNet-1k-LT in-distribution dataset**
> | Method | Texture (AUROC) | Texture (FPR95) | Places (AUROC) | Places (FPR95) | SUN (AUROC) | SUN (FPR95) | iNaturalist (AUROC) | iNaturalist (FPR95) | ImageNet-O (AUROC) | ImageNet-O (FPR95) | Average (AUROC) | Average (FPR95) |
> | ----| ---- | ---- | ---- | ---- | ---- | ---- | ---- |---- | ---- | ---- | ---- | ---- |
> | MSP         | 78.01  | 73.35 |  75.50 | 78.45 |  75.30   | 79.07  |  87.21  | 54.52 |  67.95  | 87.70 |  76.80   | 74.62 |
> | MLS         | 84.44  | 65.73 |  78.50 | 75.84 |  79.44   | 75.44  |  91.83  | 46.49 |  76.83  | 84.55 |  82.21   | 69.61 |
> | Energy      | 85.85  | 60.04 |  78.76 | 74.90 |  80.02   | 74.11  |  92.72  | 42.10 |  78.73  | 81.55 |  83.22   | 66.54 |
> | Mahalanobis | 89.61  | 41.86 |  **79.27** | **67.26** |  **82.44**   | **63.88**  |  97.62  | 11.83 |  80.09  | 77.30 |  85.81   | 52.43 |
> | residual    | 84.86  | 56.21 |  68.21 | 86.03 |  69.96   | 84.55  |  88.63   | 49.79 |  73.94  | 77.95 |  77.12   | 70.91 |
> | Vim         | 86.49  | 57.06 |  78.97 | 74.35 |  80.27   | 73.02  |  93.25   | 38.66 |  79.22  | 80.85 |  83.64   | 64.79 |
> | NECO        | 86.84  | 56.44 |  79.03 | 73.47 |  80.61   | 72.26  |  94.78   | 30.01 |  79.54  | 79.75 |  84.16   | 52.39 |
> | KL-matching | 81.97  | 67.22 |  77.80 | 74.27 |  78.06   | 73.72  |  91.59   | 41.18 |  72.78  | 84.35 |  80.44   | 68.15 |
> | SFM (ours)  | **92.21**  | **29.84** |  78.40 | 68.87 |  81.25   | 66.86  |  **97.72**   | **11.30** |  **82.43**  | **69.90** |  **86.40**   | **49.35** |
> | ----| ---- | ---- | ---- | ---- | ---- | ---- | ---- |---- | ---- | ---- | ---- | ---- |
> | MSP         | 82.60  | 60.11 |  81.41 | 66.59 |  81.97   | 63.84  |  94.31   | 25.69 |  77.74  | 73.60 |  83.61   | 57.97 |
> | MLS         | 87.94  | 50.51 |  84.97 | 60.73 |  86.46   | 56.44  |  96.63   | 17.04 |  84.64  | 65.40 |  88.13   | 50.02 |
> | Energy      | 88.96  | 46.03 |  85.45 | 58.10 |  87.20   | 53.27  |  97.09   | 14.06 |  85.91  | 60.85 |  88.92   | 46.46 |
> | Mahalanobis | 91.13  | 36.06 |  **86.30** | **54.87** |  **89.54**   | 46.81  |  99.03   | 4.49  |  87.74  | 55.45 |  90.75   | 39.54 |
> | residual    | 88.66  | 45.11 |  79.43 | 70.30 |  84.29   | 60.89  |  96.18   | 20.30 |  82.07  | 65.85 |  86.12   | 52.49 |
> | Vim         | 89.38  | 44.08 |  85.72 | 56.97 |  87.57   | 52.06  |  97.39   | 12.38 |  86.29  | 59.25 |  89.27   | 44.95 |
> | NECO        | 89.72  | 43.40 |  85.86 | 56.63 |  88.18   | 51.50  |  98.07   | 9.36  |  87.00  | 58.40 |  89.77   | 43.86 |
> | KL-matching | 86.01  | 50.67 |  83.63 | 60.68 |  84.80   | 56.71  |  96.68   | 14.62 |  81.90  | 65.35 |  86.60   | 49.61 |
> | SFM (ours)  | **93.35**  | **25.94** |  86.18 | 55.21 |  89.52   | **46.74**  |  **99.13**   | **3.99**  |  **89.13**  | **50.20** |  **91.46**   | **36.42** |

---

> ### Author Response · Authors · 2024-11-16
>
> **[W6] May not be fair to compare with MCM.**
>
> **[A6]** The MCM method is naturally better suited for zero-shot OOD tasks compared to fine-tuning tasks. The prevalent fine-tuning approach, which mainly targets the visual encoder, tends to disrupt the initial alignment between the visual and text components after fine-tuning, resulting in less effective outcomes. Our goal in including the MCM method in our experiment was not to make a direct comparison but to empirically showcase that our proposed method enhances OOD detection performance. Conversely, methods like ViM and NECO are methodologically and conceptually more similar to our approach and, therefore, require a more thorough comparison. We will elaborate on this in the revised manuscript to provide readers with a clearer understanding of the comparative framework and the uniqueness of our contributions. Moreover, we present the results of MCM on the fine-tuned model (i.e. MCM-tuned) in the table below for comparison.
>
> **Results for CIFAR-100 in-distribution dataset**
> | Method | Texture (AUROC) | Texture (FPR95) | SVHN (AUROC) | SVHN (FPR95) | CIFAR10 (AUROC) | CIFAR10 (FPR95) | Tiny ImageNet (AUROC) | Tiny ImageNet (FPR95) | LSUN (AUROC) | LSUN (FPR95) | Places365 (AUROC) | Places365 (FPR95) | Average (AUROC) | Average (FPR95) |
> | ---- | ---- | ---- | ---- | ---- | ---- | ---- | ---- |---- | ---- | ---- | ---- | ---- | ---- | ---- |
> | MCM-tuned   | 75.33  | 91.38 |  91.55  | 60.96  |  75.60  | 91.03 |  64.07   | 95.40 |  55.14  | 98.93 |  63.71   | 97.67 |  70.90   | 89.23  |
> | SFM   | **99.95**  | **0.02**  |  **98.31**  | **8.62**  |  **88.56**   | **53.97** |  **97.54**   | **12.91** |  **99.93**  | **0.06**  |  **99.95**   | **0.02**  |  **97.37**   | **12.60**  |
>
> **Results for CIFAR-100-LT in-distribution dataset**
> | Method | Texture (AUROC) | Texture (FPR95) | SVHN (AUROC) | SVHN (FPR95) | CIFAR10 (AUROC) | CIFAR10 (FPR95) | Tiny ImageNet (AUROC) | Tiny ImageNet (FPR95) | LSUN (AUROC) | LSUN (FPR95) | Places365 (AUROC) | Places365 (FPR95) | Average (AUROC) | Average (FPR95) |
> | ---- | ---- | ---- | ---- | ---- | ---- | ---- | ---- |---- | ---- | ---- | ---- | ---- | ---- | ---- |
> | MCM-tuned   | 75.33  | 91.38  |  91.55  | 60.96  |  75.60  | 91.03 |  64.07   | 95.40 |  55.14   | 98.93 |  63.71   | 97.67 |  70.90   | 89.23  |
> | SFM   | **99.92**  | **0.02**   |  **97.50**  | **15.65**  |  **85.20**  | **60.41** |  **95.05**   | **26.05** |  **99.92**   | **0.04**  |  **99.93**   | **0.01**  |  **96.25**   | **17.03** |
>
> **Results for ImageNet-1k-LT in-distribution dataset**
> | Method | Texture (AUROC) | Texture (FPR95) | Places (AUROC) | Places (FPR95) | SUN (AUROC) | SUN (FPR95) | iNaturalist (AUROC) | iNaturalist (FPR95) | ImageNet-O (AUROC) | ImageNet-O (FPR95) | Average (AUROC) | Average (FPR95) |
> | ----| ---- | ---- | ---- | ---- | ---- | ---- | ---- |---- | ---- | ---- | ---- | ---- |
> | MCM-tuned  | 85.64  | 60.11  |  89.82  | 44.32  |  **92.92**   | **36.25**  |  94.26   | 32.01 |  79.26   | 76.10 |  88.38   | 49.76 |
> | SFM  | **90.95**  | **43.10**  |  **92.43**  | **34.80**  |  92.62   | 39.07  |  **94.59**   | 2**8.54** |  **83.62**   | **68.90**|  **90.84**   | **42.88** |

---

> ### Author Response · Authors · 2024-11-26
>
> Dear Reviewer cJWd,
>
> Thank you for your follow-up questions! We address your concerns below.
>
> **[Q1] Performance comparision with ViM based on pre-trained checkpoints as the backbone.**
>
> **[A1]** Thanks for the suggestion! We compare the performance of our method with ViM in the table below. The results are based on Imagenet-1K-LT as the ID dataset. Given that running experiments on the ImageNet-1k dataset is time-consuming, we'll update the results as promptly as possible once the testing phase finishes.
>
> From the results, we can see that our approach still outperforms ViM by over 3\% w.r.t. FPR95 averaged over 5 OOD datasets, also showing its effectiveness.
>
> *Table 1: Performance comparison between ViM and our approach using ImageNet-1k-LT in-distribution dataset. In this experiment, we utilize the pre-trained model as the backbone without fine-tuning.*
> | Method | Texture (AUROC) | Texture (FPR95) | Places (AUROC) | Places (FPR95) | SUN (AUROC) | SUN (FPR95) | iNaturalist (AUROC) | iNaturalist (FPR95) | ImageNet-O (AUROC) | ImageNet-O (FPR95) | Average (AUROC) | Average (FPR95) |
> | ---- | ---- | ---- | ---- | ---- | ---- | ---- | ---- |---- | ---- | ---- | ---- | ---- |
> | ViM   | 90.78  | 34.88  |  87.56  | 52.80 |  89.03   | 48.09  |  98.61   | 5.62 |  89.36   | 47.50 |  91.07   | 37.78 |
> | Ours  | 97.15 | 10.20 | 81.51 | 67.20 | 85.49 | 56.49 | 98.69 | 5.13 | 93.15 | 32.15 | **91.20** | **34.23** |
>
> **[Q2] Clearification on the results in Table 7 of the paper**
>
> **[A2]** Sorry for the confusion! `Table 7 in the paper` reports results for both pre-trained models (denoted by `zero-shot`) and fine-tuned models (denoted by `long-tailed` or `balanced`). We will revise the description to improve the clearity.

---

> > ### Comment · Reviewer_cJWd · 2024-11-26
> >
> > Thank you for the prompt responses. Since all my concerns have been thoroughly addressed, I will be increasing my score.

---

> > > ### Author Response · Authors · 2024-11-26
> > >
> > > Thank you for all your suggestions which help a lot to improve our work!

---

> ### Author Response · Authors · 2024-11-27
>
> As mentioned in the previous response, we present the comparison results for ViM and SFM (ours) on the ImageNet-1k dataset when utilizing the pre-trained checkpoints as the backbone. From the results, we draw a similar conclusion that our proposed SFM achieves superior performance than ViM, showing its robustness to different backbones.
>
>
> *Results for ViM and SFM on ImageNet-1k in-distribution dataset based on pre-trained model without fine-tuning.*
> | Method | Texture (AUROC) | Texture (FPR95) | Places (AUROC) | Places (FPR95) | SUN (AUROC) | SUN (FPR95) | iNaturalist (AUROC) | iNaturalist (FPR95) | ImageNet-O (AUROC) | ImageNet-O (FPR95) | Average (AUROC) | Average (FPR95) |
> | ---- | ---- | ---- | ---- | ---- | ---- | ---- | ---- |---- | ---- | ---- | ---- | ---- |
> | ViM   | 90.57  | 35.16  |  86.13  | 56.54 |  87.67   | 53.82  |  97.90   | 9.16 |  89.90   | 44.95 |  90.43   | 39.93 |
> | SFM (ours)  | 97.19 | 9.88 | 82.72 | 65.14 | 86.41 | 54.64 | 98.80 | 4.75 | 93.40 | 31.45 | **91.70** | **33.17** |

---

### Official Review · Reviewer_5FtG · 2024-11-03

**Soundness:** 3
**Presentation:** 3
**Contribution:** 2
**Rating:** 5
**Confidence:** 4

**Summary:**

This paper works on out-of-distribution detection (OOD) questions and proposes to use the rich information contained in the shallow features to improve the detection of the OOD samples.  Experiments on both class-balanced and long-tailed datasets show the effectiveness of using such shallow features and the proposed method.

**Strengths:**

1. The writing is good and the idea is easy to follow.
2. The observation of existing methods neglecting the shallow features makes sense to me and Fig.1 well illustrates the comparison of using the fused features from all layers than only using the last layer.
3. The achieved results are promising showing the proposed method outperforms other competitors.

**Weaknesses:**

1.  The contribution of using all features for OOD looks over-claimed given the fact that there are already two related works (Mahalanobis and TRUSTED) explored that for OOD.  Emphasis on the novel multiple-feature fuse method might be more suitable.
2. The novelty part is not that new to me. Though the adaptive fuse for multiple features is not explored for OOD, what I understand of the key idea is to improve the discriminates of different image features, from this perspective,  the idea of using multilayer features simply or adaptively is quite explored as in [ref1-4] for example.
3.  The proposed fusion method is not comprehensively studied. For example, what about the comparison results with some simple fuse method e.g., average or fixed ratios, and other proposed fusion methods e.g. [ref1-4]?
4. Open questions about the task motivation:  is it definitely better to detect all out-of-distribution data for the model training, or will the OOD data help with the model's generalization ability for the cross-domain testing which we can't ensure is always in the same domain with the training.
5. Minor: the bold font is recommended to apply for tables to make it easier to read the results.

[ref1] Attentional Feature Fusion
[ref2] Enhanced Blind Face Restoration with Multi-Exemplar Images and Adaptive Spatial Feature Fusion
[ref3] Shallow Feature Matters for Weakly Supervised Object Localization
[ref4] Multilayer Feature Fusion Network for Scene Classification in Remote Sensing

**Questions:**

please see weakness

---

> ### Author Response · Authors · 2024-11-16
>
> Dear Reviewer 5FtG,
>
> We are truly and deeply grateful for the substantial amount of time and painstaking effort you have committed to meticulously reviewing our paper and furnishing such highly detailed and exceptionally valuable feedback.
>
> **[W1] Emphasis on the novel multiple-feature fuse method might be more suitable.**
>
> **[A1]** We truly value your valuable feedback and suggestions regarding the focus of our paper. We will modify our statement in the contribution section to "proposed a new adaptive feature fusion strategy".  We think better showcases the unique characteristics and advantages of our research and appreciate your help in improving and strengthening our manuscript.
>
> **[W2,W3] Innovation declaration and the comparison results with some simple fuse method.**
>
> **[A2,A3]** We appreciate your suggestions greatly. After carefully reviewing [ref1-4], we agree that these articles are pioneering in the use of multi-layer feature fusion. Consequently, contingent upon the acceptance of our paper, we shall ensure an accurate citation and comprehensive analysis of these works within the related work section. Regarding the novelty of our paper, we wish to clarify that we have proposed a novel adaptive feature fusion algorithm applicable to OOD detection. Unlike [ref1-3], which modify network architecture by utilizing residuals or attention mechanisms to combine multi-layer features, [ref4] uses different enhancements of the same sample. Our approach of adaptive feature fusion distinctly differs from these methodologies. To thoroughly investigate our algorithm's application in OOD detection, we have also examined other weight configurations, such as uniform or fixed ratio weights, and conducted ablation experiments detailed in the following table.
> Due to constraints on rebuttal time, we assessed CIFAR-100, CIFAR-100-LT, and ImageNet-1k-LT utilizing the pre-trained ImageNet21k model alongside CLIP. The findings demonstrate that the SFM approach consistently achieves either the top (bold) or the second-best (italic) position across all contexts. Remarkably, when ranked second, our method's performance closely rivals that of the first-place finisher. Specifically, **our approach attains 96.17 and 17.82, whereas W_0.5 achieves 95.82 and 19.04**, w.r.t. AUROC and FPR95 respectively, which shows the effectiveness of our SFM approach.
>
> *Table 1: Results on CIFAR-100 in-distribution dataset based on ImageNet-21k pre-trained ViT and CLIP, respectively.*
> | Method | Texture (AUROC) | Texture (FPR95) | SVHN (AUROC) | SVHN (FPR95) | CIFAR10 (AUROC) | CIFAR10 (FPR95) | Tiny ImageNet (AUROC) | Tiny ImageNet (FPR95) | LSUN (AUROC) | LSUN (FPR95) | Places365 (AUROC) | Places365 (FPR95) | Average (AUROC) | Average (FPR95) |
> | ---- | ---- | ---- | ---- | ---- | ---- | ---- | ---- |---- | ---- | ---- | ---- | ---- | ---- | ---- |
> | In21k W_0.5   | **100.0**  | **0.00**  |  99.43   | 2.36  |  96.76   | 18.26  |  99.62   | 1.81  |  **100.0**   | **0.00**  |  99.99  | 0.01  |  **99.30**  | **3.74**  |
> | In21k W_0.75   | 99.99   | 0.04  |  99.27   | 3.33  |  96.99   | 16.82  |  98.83   | 5.22  |  99.89   | 0.06  |  99.89   | 0.26  |  99.14   | 4.29  |
> | In21k W_1.0   | 99.97   | 0.12  |  99.16   | 3.92  |  **97.09**  | **16.49**  |  97.99   | 8.96  |  99.61   | 1.07  |  99.67   | 1.33  |  98.92   | 5.32  |
> | In21k SFM   | **100.0**   | **0.00**  |  **99.50**   | **1.91**  |  96.47   | 19.52  |  **99.80**   | **1.11**  |  **100.0**   | **0.00**  |  **100.0**   | **0.00**  |  *99.29*   | *3.76*  |
> | ---- | ---- | ---- | ---- | ---- | ---- | ---- | ---- |---- | ---- | ---- | ---- | ---- | ---- | ---- |
> | CLIP W_0.5   | 99.92  | 0.07  |  98.00  | 10.27  |  89.15  | 51.09 |  96.49   | 17.36  |  99.88   | 0.21 |  99.91   | 0.12  |  97.22   | 13.19  |
> | CLIP W_0.75   | 99.61  | 0.85  |  97.84  | 12.34  |  **89.36**  | **50.66** |  94.38   | 26.56 |  99.24   | 3.85 |  99.55   | 1.68  |  96.66   | 15.99  |
> | CLIP W_1.0   | 99.23  | 1.68  |  96.89  | 23.27  |  89.01   | 52.26  |  93.75   | 32.28  |  98.81   | 6.44  |  99.29   | 3.13  |  96.16   | 19.84  |
> | CLIP SFM   | **99.95**  | **0.02**  |  **98.31**  | **8.62**  |  88.56   | 53.97  |  **97.54**   | **12.91** |  **99.93**   | **0.06**  |  **99.95**   | **0.02**  |  **97.37**   | **12.60** |

---

> ### Author Response · Authors · 2024-11-16
> **The second part of reply to [W2, W3]**
>
> *Table 2: Results on CIFAR-100-LT in-distribution dataset based on ImageNet-21k pre-trained ViT and CLIP, respectively.*
>
> | Method | Texture (AUROC) | Texture (FPR95) | SVHN (AUROC) | SVHN (FPR95) | CIFAR10 (AUROC) | CIFAR10 (FPR95) | Tiny ImageNet (AUROC) | Tiny ImageNet (FPR95) | LSUN (AUROC) | LSUN (FPR95) | Places365 (AUROC) | Places365 (FPR95) | Average (AUROC) | Average (FPR95) |
> | ---- | ---- | ---- | ---- | ---- | ---- | ---- | ---- |---- | ---- | ---- | ---- | ---- | ---- | ---- |
> | In21k W_0.5   | 100.0   | 0.00  |  99.68   | 0.68  |  94.56   | 27.90  |  99.58   | 1.68  |  **99.99**   | **0.00**  |  99.98 | 0.01  |  **98.97**  | **5.05**  |
> | In21k W_0.75   | 99.99   | 0.05  |  99.49   | 1.66  |  94.93   | 26.42  |  98.67   | 5.57  |  99.81   | 0.39  |  99.83   | 0.53  |  98.79   | 5.77  |
> | In21k W_1.0   | 99.96   | 0.20  |  99.33   | 2.51  |  **95.09**   | **25.98**  |  97.63   | 9.26  |  99.48   | 2.26  |  99.57   | 1.71  |  98.51   | 6.99  |
> | In21k SFM   | **100.0**   | **0.00**  |  **99.75**  | **0.43**  |  94.22   | 29.86  |  **99.75**   | **1.12**  |  **99.99**   | **0.00**  |  **99.99**   | **0.01**  |  98.95   | 5.24  |
> | ---- | ---- | ---- | ---- | ---- | ---- | ---- | ---- |---- | ---- | ---- | ---- | ---- | ---- | ---- |
> | CLIP W_0.5   | 99.88   | 0.04  |  97.37   | 16.90  |  85.52   | **58.97**  |  94.36   | 28.65  |  99.87   | 0.11  |  99.89 | 0.05  |  96.15  | 17.45  |
> | CLIP W_0.75   | 99.52   | 0.55  |  97.15  | 19.66  |  **85.73**  | 58.99|  91.58   | 39.22  |  99.36   | 2.63 |  99.52  | 1.20 |  95.48   | 20.38  |
> | CLIP W_1.0   | 99.11  | 1.03  |  95.92  | 29.87  |  84.76   | 60.58  |  90.97   | 43.83 |  99.08   | 4.07  |  99.28   | 1.99  |  94.85   | 23.56 |
> | CLIP SFM   | **99.92**  | **0.02**  |  **97.50**  | **15.65**  |  85.20   | 60.41  |  **95.05**   | **26.05** | **99.92**   | **0.04**  |  **99.93**   | **0.01** |  **96.25**   | **17.03** |

---

> ### Author Response · Authors · 2024-11-16
>
> **[W4] Open questions about the task motivation.**
>
> **[A4]** Regarding the initial point about whether detecting every out-of-distribution (OOD) sample during model training is always advantageous, the answer isn't straightforward. Many semi-supervised or label noise learning techniques recommend using the most accurate data possible during training. However, there are instances where including a small number of errors or OOD samples can be beneficial to the learning process. This inclusion might mimic the effects seen in self-supervised or contrastive learning approaches. Hence, we assert that identifying all OOD samples may not always be the best strategy.
>
> For the second part of your inquiry, this paper primarily focuses on semantic OOD. It proposes that in-distribution (ID) and OOD classes are distinct, and training is conducted solely with ID data, yet OOD detection is necessary during testing. As pertains to your question, a covariate shift is expected in OOD detection. In such cases, cross-domain OOD becomes relevant during training, particularly when cross-domain test samples are involved. This is due to its potential to improve the model's ability to generalize and withstand varied data distributions.
>
> We hope this provides a clearer and more satisfactory explanation. We appreciate your insightful feedback once more.
>
>
> **[W5] Minor: the bold font**
>
> **[A4]** We're sincerely thankful for your careful review and the suggestion about bold font. We'll use it appropriately in the table to enhance result readability. Appreciate your help in improving our work.

---

> ### Author Response · Authors · 2024-11-18
> **The third part of reply to [W2,W3]**
>
> *Table 3: Results on ImageNet-1k-LT in-distribution dataset based on ImageNet-21k pre-trained ViT and CLIP, respectively.*
>
> | Method | Texture (AUROC) | Texture (FPR95) | Places (AUROC) | Places (FPR95) | SUN (AUROC) | SUN (FPR95) | iNaturalist (AUROC) | iNaturalist (FPR95) | ImageNet-O (AUROC) | ImageNet-O (FPR95) | Average (AUROC) | Average (FPR95) |
> | ----| ---- | ---- | ---- | ---- | ---- | ---- | ---- |---- | ---- | ---- | ---- | ---- |
> | In21k W_0.5   | 95.02  | 17.96  |  89.76  | **44.79** |  92.08   | 36.60  |  **99.36**   | 2.63 |  92.68  | 34.30 |  93.78   | 27.26 |
> | In21k W_0.75   | 93.78  | 23.39  |  89.62  | 45.54 |  91.88   | 37.22  |  99.32   | 2.74 |  92.06   | 36.75 |  93.33   | 29.13 |
> | In21k W_1.0   | 92.99  | 26.95  |  89.48  | 46.34  |  91.71   | 38.35  |  99.28   | 2.84 |  91.66   | 38.85 |  93.02   | 30.67 |
> | In21k SFM   | **96.92**  | **11.79**  |  **89.82**  | 45.36 |  **92.18**  | **36.16**  |  99.33   | **2.51** |  **93.46**   | **31.10** |  **94.34**   | **25.38** |
> | ---- | ---- | ---- | ---- | ---- | ---- | ---- | ---- |---- | ---- | ---- | ---- | ---- | ---- | ---- |
> | CLIP W_0.5   | 88.52  | 52.23  |  89.87  | 45.47  |  92.21   | 40.58  |  94.54   | **28.45** |  82.56   | 71.00 |  89.54   | 47.55 |
> | CLIP W_0.75   | 87.68  | 55.39  |  89.69  | 46.37  |  92.08   | 41.23  |  94.41  | 29.30 |  82.23   | 72.40 |  89.22   | 48.94 |
> | CLIP W_1.0   | 83.81  | 67.64  |  84.44  | 66.85 |  85.50   | 69.58  |  87.49   | 72.57 |  78.82   | 80.20  |  84.01   | 71.37 |
> | CLIP SFM   | **90.95**  | **43.10**  |  **92.43**  | **34.80**  |  **92.62**   | **39.07**  | **94.59**   | 28.54 |  **83.62**   | **68.90**  |  **90.84**   | **42.88** |

---

> > ### Comment · Reviewer_5FtG · 2024-11-26
> > **Respond to the Rebuttal**
> >
> > Dear authors,
> >
> > thanks for the responses, I have read them. However, my first and second concerns still hold, and I will keep my score unchanged.

---

> > > ### Author Response · Authors · 2024-11-26
> > > **Further response by the Authors**
> > >
> > > We sincerely apologize for any confusion caused by the original writing. We acknowledge that several existing works explore multi-layer or all-layer feature integration for OOD detection, and we do not claim to be the first to address this. Instead, as emphasized in our revised manuscript, our primary contribution lies in the development of an adaptive weighting module that dynamically adjusts based on the data. We believe this approach provides a novel perspective in addressing the challenges of OOD detection.
> > >
> > > Additionally, as the reviewer has correctly pointed out, the adaptive fusion of multiple features remains underexplored in the OOD detection domain. After carefully reviewing the papers referenced by the reviewer, we found that our key idea is substantially different from those works. In our understanding, many existing methods focus on improving the discrimination of image features, while our work proposes a distinct mechanism for adaptive feature fusion, specifically tailored to OOD detection.
> > >
> > > Therefore, we respectfully disagree with the assertion that the originality of our method is significantly diminished. We sincerely hope the reviewer can reconsider the evaluation of our work. Thank you.

---

### Official Review · Reviewer_GG5e · 2024-11-05

**Soundness:** 3
**Presentation:** 3
**Contribution:** 2
**Rating:** 5
**Confidence:** 4

**Summary:**

This paper presents a weighted feature fusion method from the top to bottom layers of a network for OOD detection. It argues that, instead of relying solely on the features of the penultimate layer in a deep network for OOD detection, incorporating features from shallow layers provides diversity in feature spaces, aiding in the separation of OOD examples from ID examples. The method uses the total variance of features across different neural network layers to estimate feature weights for OOD detection. Experiments are conducted on CIFAR-100 and Imagenet, two popular benchmarks for OOD detection, including a setup with a long-tail distribution in in-domain training examples. The experimental results show that the proposed method outperforms the compared method.

**Strengths:**

The paper is well-written and easy to follow.

Extensive experiments are presented to support the claims.

There are sufficient ablation studies.

**Weaknesses:**

I find the novelty of the method modest. There is a remark section in the paper highlighting that the idea of estimating layerwise importance is the contribution compared to the inspiring methods, Mahalanobis and TRUSTED. While I can see a performance comparison with Mahalanobis in the paper, I could not find one for TRUSTED. Moreover, based on Figures 4 and 5 from the ablation studies, it hardly supports any contribution of layerwise feature weighting in OOD detection. The last layer has significant weight, while the other layers have more or less similar weights. A baseline with uniform weighting across all feature layers is also important.

Although not directly related, a paper on the variance of gradients for estimating the difficulty of images for OOD detection [ref1] is relevant and would support the argument.

[ref1] Agarwal, C., D'souza, D., & Hooker, S. (2022). Estimating example difficulty using variance of gradients. In Proceedings of the IEEE/CVF Conference on Computer Vision and Pattern Recognition (pp. 10368-10378).

**Questions:**

I am not fully convinced that there is a significant role of layerwise weightage in OOD detection. Giving more weights for the penultimate layer and the same weight to other layers can still perform well.    A clarification with experiments may change my rating.

---

> ### Author Response · Authors · 2024-11-16
>
> Dear Reviewer GG5e,
>
> We sincerely appreciate the time and effort you have dedicated to reviewing our paper. We address your concerns below.
>
> **[W1] Experimental results for TRUSTED.**
>
> **[A1]** Thank you for your reminder. We previously overlooked including the comparison results with the Trusted method, but these `have now been added to the revised manuscript in Table 1, 2, 3, 4`. For your convenience, we have summarized the comparison between the Trusted method and our approach on CIFAR-100 dataset in tables below. As shown in the results, our method consistently outperforms Trusted.
>
> *Table 1: Results on CIFAR-100 in-distribution dataset for Trusted and SFM. The top two rows of the table present results based on the ImageNet-21k pre-trained ViT, while the bottom two rows show results based on CLIP.*
> | Method | Texture (AUROC) | Texture (FPR95) | SVHN (AUROC) | SVHN (FPR95) | CIFAR10 (AUROC) | CIFAR10 (FPR95) | Tiny ImageNet (AUROC) | Tiny ImageNet (FPR95) | LSUN (AUROC) | LSUN (FPR95) | Places365 (AUROC) | Places365 (FPR95) | Average (AUROC) | Average (FPR95) |
> | ---- | ---- | ---- | ---- | ---- | ---- | ---- | ---- |---- | ---- | ---- | ---- | ---- | ---- | ---- |
> | Trusted  | **100.0**   | **0.00**  |  98.78   | 5.77  |  93.35   | 33.51  |  98.09   | 9.76  |  **100.0**   | 0.01  |  100.0   | 0.00  |  98.37   | 8.17  |
> |SFM (ours)  |  **100.0** | **0.00** | **99.50**    | **1.91** | **96.47**  | **19.52** | **99.80** | **1.11** | **100.0** | **0.00** | **100.0** | **0.00** | **99.29** | **3.76** |
> | ---- | ---- | ---- | ---- | ---- | ---- | ---- | ---- |---- | ---- | ---- | ---- | ---- | ---- | ---- |
> | Trusted | **99.98**   | 0.04  |  97.21   | 17.80  |  86.32   | 61.45  |  97.13   | 15.68  |  99.95   | 0.03  |  99.96   | 0.08  |  96.76   | 15.85  |
> | SFM (ours)  |  99.95 | **0.02** | **98.31**  | **8.62** | **88.56**  | **53.97** | **97.54** | **12.91** | **99.93** | **0.06** | **99.95** | **0.02** | **97.37** | **12.60** |
>
> ---
> ---
>
> *Table 2: Results on CIFAR-100-LT in-distribution dataset for Trusted and SFM. The top two rows of the table present results based on the ImageNet-21k pre-trained ViT, while the bottom two rows show results based on CLIP.*
> | Method | Texture (AUROC) | Texture (FPR95) | SVHN (AUROC) | SVHN (FPR95) | CIFAR10 (AUROC) | CIFAR10 (FPR95) | Tiny ImageNet (AUROC) | Tiny ImageNet (FPR95) | LSUN (AUROC) | LSUN (FPR95) | Places365 (AUROC) | Places365 (FPR95) | Average (AUROC) | Average (FPR95) |
> | ---- | ---- | ---- | ---- | ---- | ---- | ---- | ---- |---- | ---- | ---- | ---- | ---- | ---- | ---- |
> | Trusted | **100.0**   | **0.00**  |  99.12   | 3.60  |  87.34   | 52.84  |  97.67   | 10.37  |  99.97   | **0.00**  |  99.98   | **0.00**  |  97.35   | 11.13  |
> | SFM (ours)  |  **100.0** | **0.00** | **99.75**  | **0.43** | **94.22**  | **29.86** | **99.75** | **1.12** | **99.99** | **0.00** | **99.99** | 0.01 | **98.95** | **5.24** |
> | ---- | ---- | ---- | ---- | ---- | ---- | ---- | ---- |---- | ---- | ---- | ---- | ---- | ---- | ---- |
> | Trusted | **99.97**   | 0.11  |  93.57   | 43.80  |  80.76   | 70.36  |  **95.46**   | **25.58**  |  **99.95**   | 0.10  |  99.95   | 0.08  |  94.94   | 23.34 |
> | SFM (ours)  |  99.92 | **0.02** | **97.50**  | **15.65** | **85.20**  | **60.41** | 95.05 | 26.05 | 99.92 | **0.04** | **99.93** | **0.01** | **96.25** | **17.03** |

---

> ### Author Response · Authors · 2024-11-16
>
> **[W2,Q1] Contribution of layerwise feature weighting in OOD detection.**
>
> **[A2]** Thank you very much for your valuable suggestion! We agree with the reviewer that the penultimate layer plays a crucial role in OOD detection. To further investigate this, we conducted comparative experiments by assigning different weights to the penultimate layer. For example, in the case of W_0.5, the penultimate layer was given a weight of 0.5, while the remaining layers equally shared the remaining 0.5 weight.
>
> The results are summarized in the tables below. From these results, it is evident that our method achieves superior performance on most datasets compared to assigning fixed weights across all layers. Notably, the improvement is particularly significant on the ImageNet dataset.
>
> Thank you again for your insightful comment, which helped us further validate the effectiveness of our approach.
>
>
> *Table 1: Results on CIFAR-100 in-distribution dataset based on ImageNet-21k pre-trained ViT and CLIP, respectively.*
> | Method | Texture (AUROC) | Texture (FPR95) | SVHN (AUROC) | SVHN (FPR95) | CIFAR10 (AUROC) | CIFAR10 (FPR95) | Tiny ImageNet (AUROC) | Tiny ImageNet (FPR95) | LSUN (AUROC) | LSUN (FPR95) | Places365 (AUROC) | Places365 (FPR95) | Average (AUROC) | Average (FPR95) |
> | ---- | ---- | ---- | ---- | ---- | ---- | ---- | ---- |---- | ---- | ---- | ---- | ---- | ---- | ---- |
> | In21k W_0.5   | **100.0**  | **0.00**  |  99.43   | 2.36  |  96.76   | 18.26  |  99.62   | 1.81  |  **100.0**   | **0.00**  |  99.99  | 0.01  |  **99.30**  | **3.74**  |
> | In21k W_0.75   | 99.99   | 0.04  |  99.27   | 3.33  |  96.99   | 16.82  |  98.83   | 5.22  |  99.89   | 0.06  |  99.89   | 0.26  |  99.14   | 4.29  |
> | In21k W_1.0   | 99.97   | 0.12  |  99.16   | 3.92  |  **97.09**  | **16.49**  |  97.99   | 8.96  |  99.61   | 1.07  |  99.67   | 1.33  |  98.92   | 5.32  |
> | In21k SFM   | **100.0**   | **0.00**  |  **99.50**   | **1.91**  |  96.47   | 19.52  |  **99.80**   | **1.11**  |  **100.0**   | **0.00**  |  **100.0**   | **0.00**  |  *99.29*   | *3.76*  |
> | ---- | ---- | ---- | ---- | ---- | ---- | ---- | ---- |---- | ---- | ---- | ---- | ---- | ---- | ---- |
> | CLIP W_0.5   | 99.92  | 0.07  |  98.00  | 10.27  |  89.15  | 51.09 |  96.49   | 17.36  |  99.88   | 0.21 |  99.91   | 0.12  |  97.22   | 13.19  |
> | CLIP W_0.75   | 99.61  | 0.85  |  97.84  | 12.34  |  **89.36**  | **50.66** |  94.38   | 26.56 |  99.24   | 3.85 |  99.55   | 1.68  |  96.66   | 15.99  |
> | CLIP W_1.0   | 99.23  | 1.68  |  96.89  | 23.27  |  89.01   | 52.26  |  93.75   | 32.28  |  98.81   | 6.44  |  99.29   | 3.13  |  96.16   | 19.84  |
> | CLIP SFM   | **99.95**  | **0.02**  |  **98.31**  | **8.62**  |  88.56   | 53.97  |  **97.54**   | **12.91** |  **99.93**   | **0.06**  |  **99.95**   | **0.02**  |  **97.37**   | **12.60** |

---

> ### Author Response · Authors · 2024-11-16
> **The second part of reply to [W2, Q1]**
>
> *Table 2: Results on CIFAR-100-LT in-distribution dataset based on ImageNet-21k pre-trained ViT and CLIP, respectively.*
>
> | Method | Texture (AUROC) | Texture (FPR95) | SVHN (AUROC) | SVHN (FPR95) | CIFAR10 (AUROC) | CIFAR10 (FPR95) | Tiny ImageNet (AUROC) | Tiny ImageNet (FPR95) | LSUN (AUROC) | LSUN (FPR95) | Places365 (AUROC) | Places365 (FPR95) | Average (AUROC) | Average (FPR95) |
> | ---- | ---- | ---- | ---- | ---- | ---- | ---- | ---- |---- | ---- | ---- | ---- | ---- | ---- | ---- |
> | In21k W_0.5   | 100.0   | 0.00  |  99.68   | 0.68  |  94.56   | 27.90  |  99.58   | 1.68  |  **99.99**   | **0.00**  |  99.98 | 0.01  |  **98.97**  | **5.05**  |
> | In21k W_0.75   | 99.99   | 0.05  |  99.49   | 1.66  |  94.93   | 26.42  |  98.67   | 5.57  |  99.81   | 0.39  |  99.83   | 0.53  |  98.79   | 5.77  |
> | In21k W_1.0   | 99.96   | 0.20  |  99.33   | 2.51  |  **95.09**   | **25.98**  |  97.63   | 9.26  |  99.48   | 2.26  |  99.57   | 1.71  |  98.51   | 6.99  |
> | In21k SFM   | **100.0**   | **0.00**  |  **99.75**  | **0.43**  |  94.22   | 29.86  |  **99.75**   | **1.12**  |  **99.99**   | **0.00**  |  **99.99**   | **0.01**  |  98.95   | 5.24  |
> | ---- | ---- | ---- | ---- | ---- | ---- | ---- | ---- |---- | ---- | ---- | ---- | ---- | ---- | ---- |
> | CLIP W_0.5   | 99.88   | 0.04  |  97.37   | 16.90  |  85.52   | **58.97**  |  94.36   | 28.65  |  99.87   | 0.11  |  99.89 | 0.05  |  96.15  | 17.45  |
> | CLIP W_0.75   | 99.52   | 0.55  |  97.15  | 19.66  |  **85.73**  | 58.99|  91.58   | 39.22  |  99.36   | 2.63 |  99.52  | 1.20 |  95.48   | 20.38  |
> | CLIP W_1.0   | 99.11  | 1.03  |  95.92  | 29.87  |  84.76   | 60.58  |  90.97   | 43.83 |  99.08   | 4.07  |  99.28   | 1.99  |  94.85   | 23.56 |
> | CLIP SFM   | **99.92**  | **0.02**  |  **97.50**  | **15.65**  |  85.20   | 60.41  |  **95.05**   | **26.05** | **99.92**   | **0.04**  |  **99.93**   | **0.01** |  **96.25**   | **17.03** |
>
>
> ---
> ---
>
> *Table 3: Results on ImageNet-1k-LT in-distribution dataset based on ImageNet-21k pre-trained ViT and CLIP, respectively.*
>
> | Method | Texture (AUROC) | Texture (FPR95) | Places (AUROC) | Places (FPR95) | SUN (AUROC) | SUN (FPR95) | iNaturalist (AUROC) | iNaturalist (FPR95) | ImageNet-O (AUROC) | ImageNet-O (FPR95) | Average (AUROC) | Average (FPR95) |
> | ----| ---- | ---- | ---- | ---- | ---- | ---- | ---- |---- | ---- | ---- | ---- | ---- |
> | In21k W_0.5   | 95.02  | 17.96  |  89.76  | **44.79** |  92.08   | 36.60  |  **99.36**   | 2.63 |  92.68  | 34.30 |  93.78   | 27.26 |
> | In21k W_0.75   | 93.78  | 23.39  |  89.62  | 45.54 |  91.88   | 37.22  |  99.32   | 2.74 |  92.06   | 36.75 |  93.33   | 29.13 |
> | In21k W_1.0   | 92.99  | 26.95  |  89.48  | 46.34  |  91.71   | 38.35  |  99.28   | 2.84 |  91.66   | 38.85 |  93.02   | 30.67 |
> | In21k SFM   | **96.92**  | **11.79**  |  **89.82**  | 45.36 |  **92.18**  | **36.16**  |  99.33   | **2.51** |  **93.46**   | **31.10** |  **94.34**   | **25.38** |
> | ---- | ---- | ---- | ---- | ---- | ---- | ---- | ---- |---- | ---- | ---- | ---- | ---- | ---- | ---- |
> | CLIP W_0.5   | 88.52  | 52.23  |  89.87  | 45.47  |  92.21   | 40.58  |  94.54   | **28.45** |  82.56   | 71.00 |  89.54   | 47.55 |
> | CLIP W_0.75   | 87.68  | 55.39  |  89.69  | 46.37  |  92.08   | 41.23  |  94.41  | 29.30 |  82.23   | 72.40 |  89.22   | 48.94 |
> | CLIP W_1.0   | 83.81  | 67.64  |  84.44  | 66.85 |  85.50   | 69.58  |  87.49   | 72.57 |  78.82   | 80.20  |  84.01   | 71.37 |
> | CLIP SFM   | **90.95**  | **43.10**  |  **92.43**  | **34.80**  |  **92.62**   | **39.07**  | **94.59**   | 28.54 |  **83.62**   | **68.90**  |  **90.84**   | **42.88** |

---

> > ### Author Response · Authors · 2024-11-16
> >
> > **[W3] Relevant [ref1] to support the argument.**
> >
> > **[A3]** Upon thorough examination of [ref1], we have found it to be highly pertinent to the task of OOD detection. We are truly grateful for this valuable reference. We intend to discuss it comprehensively and accurately cite it within the related work section. This will fortify the theoretical basis and context of our research, while also augmenting the thoroughness and persuasiveness of our arguments concerning OOD detection. We believe that by incorporating the insights from [ref1], our work will make a more significant contribution to the field of OOD detection and related research areas.

---

> ### Author Response · Authors · 2024-11-18
> **The second part of reply to [W1]**
>
> **Results for ImageNet-1k in-distribution dataset on Trusted and SFM**
> | Method | Texture (AUROC) | Texture (FPR95) | Places (AUROC) | Places (FPR95) | SUN (AUROC) | SUN (FPR95) | iNaturalist (AUROC) | iNaturalist (FPR95) | ImageNet-O (AUROC) | ImageNet-O (FPR95) | Average (AUROC) | Average (FPR95) |
> | ---- | ---- | ---- | ---- | ---- | ---- | ---- | ---- |---- | ---- | ---- | ---- | ---- |
> | Trusted  | 43.56   | 86.45 |  46.82  | 96.95  |  50.95 | 94.75  |  49.36  | 91.48 |  39.15   | 95.45   |  45.97   | 93.02 |
> | SFM (ours) |  **96.65** | **11.70** | **89.64** | **46.00** | **92.04** | **37.78** | **99.40** | **2.26** | **93.76** | **29.80** | **94.30** | **25.51** |
> | ---- | ---- | ---- | ---- | ---- | ---- | ---- | ---- |---- | ---- | ---- | ---- | ---- |
> | Trusted | **95.87**    | **19.80** |  74.59  | 78.06  |  76.71 | 76.42  |  84.61  | 72.77 |  **84.12**   | **62.40**   |  83.18   | 61.89 |
> | SFM (ours) |  89.16 | 48.44 | **91.88** | **36.46** | **93.24** | **34.87** | **95.47** | **23.49** | 82.27 | 70.35 | **90.41** | **42.72** |
>
>
> **Results for ImageNet-1k-LT in-distribution dataset on Trusted and SFM**
> | Method | Texture (AUROC) | Texture (FPR95) | Places (AUROC) | Places (FPR95) | SUN (AUROC) | SUN (FPR95) | iNaturalist (AUROC) | iNaturalist (FPR95) | ImageNet-O (AUROC) | ImageNet-O (FPR95) | Average (AUROC) | Average (FPR95) |
> | ---- | ---- | ---- | ---- | ---- | ---- | ---- | ---- |---- | ---- | ---- | ---- | ---- |
> | Trusted | 91.98   | 32.36 |  82.11  | 66.31  |  85.72  | 58.34  |  98.09  | 9.29 |  90.91   | 40.15   |  89.76   | 41.29 |
> | SFM (ours)  |  **96.92** | **11.79** | **89.82** | **45.36** | **92.18** | **36.16** | **99.33** | **2.51** | **93.46** | **31.10** | **94.34** | **25.38** |
> | ---- | ---- | ---- | ---- | ---- | ---- | ---- | ---- |---- | ---- | ---- | ---- | ---- |
> | Trusted | 71.96   | 70.46 |  44.51  | 97.89  |  49.78 | 97.77  |  49.44  | 98.59 |  48.79  | 89.05   |  52.90  | 90.75 |
> | SFM (ours)  |  **90.95** | **43.10** | **92.43** | **34.80** | **92.62** | **39.07** | **94.59** | **28.54** | **83.62** | **68.90** | **90.84** | **42.88** |

---

> ### Author Response · Authors · 2024-11-27
> **Looking Forward to Discussing with the Reviewer**
>
> Dear Reviewer  GG5e,
>
> We sincerely thank you for taking the time to review our paper and providing us with your valuable feedback. Your insights are greatly appreciated and have been invaluable in improving our work.
>
> May we kindly ask if our responses have sufficiently addressed your concerns? If there are any remaining questions or issues, we are fully committed to addressing them.
>
> Once again, we deeply appreciate your time and effort in reviewing our work.
>
> Best regards,
>
> The Authors

---

### Comment · Area_Chair_MSiR · 2024-11-24
**Reminder - Public Discussion Phase Ending Soon**

Dear PC memebers,

Thank you for your valuable comments during the review period, which raised many interesting and insightful questions. Now the discussion period is coming to a close, please take a moment to review the authors’ responses if you haven’t done so already. Even if you decide not to update your evaluation, kindly confirm that you have reviewed the responses and that they do not change your assessment.

Timeline: As a reminder, the review timeline is as follows:

November 26: Last day for reviewers to ask questions to authors.
November 27: Last day for authors to respond to reviewers.
November 28 - December 10: Reviewer and area chair discussion phase.
December 20: Meta reviews and initial decisions are due.


Thank you for your time and effort!

Best regards,
AC

---

> ### Author Response · Authors · 2024-11-24
> **Thanks to the AC for initiating the discussion**
>
> We sincerely thank each reviewer for their valuable comments. We have addressed them in our responses and updated the manuscript accordingly. If the reviewers have any further questions, we are always ready to provide additional clarifications.
>
> The Authors

---

### Meta-Review · Area_Chair_MSiR · 2024-12-19

**Metareview:**

This paper focuses on layer weights and proposes an adaptive fusion method for addressing OOD scenarios. While the idea is straightforward and easy to follow, as the reviewers noted, it lacks novelty. Although there isn’t an identical method in the literature, similar ideas have been explored previously. The proposed approach, while showing some improvement in numerical experiments through careful design, does not offer a fundamental advancement in OOD.

The reviewers' opinions are split, with two negative and one positive score. The positive reviewer did not appreaciate fundamental or technical contributions neither and did not defend his/her score during the internal discussion. Considering the overall feedback, I recommend rejection.

**Additional Comments On Reviewer Discussion:**

The three reviewers generally agreed that the method lacks novelty, noting that similar ideas (of course, not the same) have been explored previously. In the rebuttal, the authors were unable to overturn this impression but provided additional experiments to strengthen the numerical comparison with existing methods. As a result, Reviewer 5FtG increased their score to 6, while the other two reviewers kept their scores unchanged.

Unfortunately, Reviewer GG5e did not respond to reminders and did not participate in the discussion, which is regrettable. However, based on the feedback from the other two reviewers and Reviewer GG5e’s initial comments, it is clear that the paper does not present an exciting contribution, and the numerical improvements is not fully convincing. Therefore, I recommend rejection.

---

### Decision · Program_Chairs · 2025-01-22

Reject